# Using Stochastic Gradient Descent to Smooth Nonconvex Functions: Analysis of Implicit Graduated Optimization with Optimal Noise Scheduling

## Abstract

The graduated optimization approach is a heuristic method for finding globally optimal solutions for nonconvex functions and has been theoretically analyzed in several studies. This paper defines a new family of nonconvex functions for graduated optimization, discusses their sufficient conditions, and provides a convergence analysis of the graduated optimization algorithm for them. It shows that stochastic gradient descent (SGD) with mini-batch stochastic gradients has the effect of smoothing the function, the degree of which is determined by the learning rate and batch size. This finding provides theoretical insights on why large batch sizes fall into sharp local minima, why decaying learning rates and increasing batch sizes are superior to fixed learning rates and batch sizes, and what the optimal learning rate scheduling is. To the best of our knowledge, this is the first paper to provide a theoretical explanation for these aspects. Moreover, a new graduated optimization framework that uses a decaying learning rate and increasing batch size is analyzed and experimental results of image classification that support our theoretical findings are reported.

## 1 Introduction

### 1.1 Background

The amazing success of deep neural networks (DNN) in recent years has been based on optimization by stochastic gradient descent (SGD) (Robbins & Monro, 1951) and its variants, such as Adam (Kingma & Ba, 2015). These methods have been widely studied for their convergence (Moulines & Bach, 2011; Needell et al., 2014) (Fehrman et al., 2020; Bottou et al., 2018; Scaman & Malherbe, 2020; Loizou et al., 2021; Zaheer et al., 2018; Zou et al., 2019; Chen et al., 2019; Zhou et al., 2020; Chen et al., 2021; Iiduka, 2022) and stability (Hardt et al., 2016; Lin et al., 2016; Mou et al., 2018; He et al., 2019) in nonconvex optimization.

SGD updates the parameters as $\boldsymbol{x}_{t+1} := \boldsymbol{x}_t - \eta \nabla f_{\mathcal{S}_t}(\boldsymbol{x}_t)$, where $\eta$ is the learning rate and $\nabla f_{\mathcal{S}_t}$ is the stochastic gradient estimated from the full gradient $\nabla f$ using a mini-batch $\mathcal{S}_t$. Therefore, there is only an $\boldsymbol{\omega}_t := \nabla f_{\mathcal{S}_t}(\boldsymbol{x}_t) - \nabla f(\boldsymbol{x}_t)$ difference between the search direction of SGD and the true steepest descent direction. Some studies claim that it is crucial in nonconvex optimization. For example, it has been proven that noise helps the algorithm to escape local minima (Ge et al., 2015; Jin et al., 2017; Daneshmand et al., 2018; Harshvardhan & Stich, 2021), achieve better generalization (Hardt et al., 2016; Mou et al., 2018), and to find a local minimum with a small loss value in polynomial time under some assumptions (Zhang et al., 2017).

Kleinberg et al. (2018) also suggests that noise smooths the objective function. Here, at time $t$, let $\boldsymbol{y}_t$ be the parameter updated by the gradient descent (GD) and $\boldsymbol{x}_{t+1}$ be the parameter updated by SGD, i.e.,

$$\boldsymbol{y}_t := \boldsymbol{x}_t - \eta \nabla f(\boldsymbol{x}_t),$$
$$\boldsymbol{x}_{t+1} := \boldsymbol{x}_t - \eta \nabla f_{\mathcal{S}_t}(\boldsymbol{x}_t)$$
$$= \boldsymbol{x}_t - \eta(\nabla f(\boldsymbol{x}_t) + \boldsymbol{\omega}_t).$$

Then, we obtain the following update rule for the sequence $\{\boldsymbol{y}_t\}$,

$$\mathbb{E}_{\boldsymbol{\omega}_t}\left[\boldsymbol{y}_{t+1}\right] = \mathbb{E}_{\boldsymbol{\omega}_t}\left[\boldsymbol{y}_t\right] - \eta \nabla \mathbb{E}_{\boldsymbol{\omega}_t}\left[f(\boldsymbol{y}_t - \eta \boldsymbol{\omega}_t)\right], \tag{1}$$

where $f$ is Lipschitz continuous and differentiable. Therefore, if we define a new function $\hat{f}(\boldsymbol{y}_t) := \mathbb{E}_{\boldsymbol{\omega}_t}[f(\boldsymbol{y}_t - \eta \boldsymbol{\omega}_t)]$, $\hat{f}$ can be smoothed by convolving $f$ with noise (see Definition 2.1, also Wu (1996)), and its parameters $\boldsymbol{y}_t$ can approximately be viewed as being updated by using the gradient descent to minimize $\hat{f}$. In other words, simply using SGD with a mini-batch smooths the function to some extent and may enable escapes from local minima. (The derivation of equation (1) is in Section A.)

**Graduated Optimization.** Graduated optimization is one of the global optimization methods, which searches for the global optimal solution of difficult multimodal optimization problems. The method generates a sequence of simplified optimization problems that gradually approach the original problem through different levels of local smoothing operations. It solves the easiest simplified problem first, as it should have nice properties such as convexity or strong convexity; after that, it uses that solution as the initial point for solving the second-simplest problem, then the second solution as the initial point for solving the third-simplest problem and so on, as it attempts to escape from local optimal solutions of the original problem and reach a global optimal solution.

This idea was first established as graduated non-convexity (GNC) by Blake & Zisserman (1987) and has since been studied in the field of computer vision for many years. Similar early approaches can be found in Witkin et al. (1987) and Yuille (1989), and the same concept has appeared in the fields of numerical analysis (Allgower & Georg, 1990) and optimization (Rose et al., 1990; Wu, 1996). Over the past 25 years, graduated optimization has been successfully applied to many tasks in computer vision, such as early vision (Black & Rangarajan, 1996), image denoising (Nikolova et al., 2010), optical flow (Sun et al., 2010; Brox & Malik, 2011), dense correspondence of images (Kim et al., 2013), and robust estimation (Yang et al., 2020; Antonante et al., 2022; Peng et al., 2023). In addition, it has been applied to certain tasks in machine learning, such as semi-supervised learning (Chapelle et al., 2006; Sindhwani et al., 2006; Chapelle et al., 2008), unsupervised learning (Smith & Eisner, 2004), and ranking Chapelle & Wu (2010). Moreover, score-based generative models (Song & Ermon, 2019; Song et al., 2021b) and diffusion models (Sohl-Dickstein et al., 2015; Ho et al., 2020; Song et al., 2021a; Rombach et al., 2022), which are currently state-of-the-art generative models, implicitly use the techniques of graduated optimization. A comprehensive survey on the graduated optimization approach can be found in (Mobahi & Fisher III, 2015b).

While graduated optimization is popular, there is not much theoretical analysis on it. Mobahi & Fisher III (2015a) performed the first theoretical analysis, but they did not provide a practical algorithm. Hazan et al. (2016) defined a family of nonconvex functions satisfying certain conditions, called $\sigma$-nice, and proposed a first-order algorithm based on graduated optimization. In addition, they studied the convergence and convergence rate of their algorithm to a global optimal solution for $\sigma$-nice functions. Iwakiri et al. (2022) proposed a single-loop method that simultaneously updates the variable that defines the noise level and the parameters of the problem and analyzed its convergence. Li et al. (2023) analyzed graduated optimization based on a special smoothing operation. Note that Duchi et al. (2012) pioneered the theoretical analysis of optimizers using Gaussian smoothing operations for nonsmooth convex optimization problems. Their method of optimizing with decreasing noise level is truly a graduated optimization approach.

### 1.2 Motivation

Equation (1) indicates that SGD smooths the function (Kleinberg et al., 2018), but it is not clear to what extent the function is smoothed or what factors are involved in the smoothing. Therefore, we decided to clarify these aspects and identify what parameters contribute to the smoothing.

Although Hazan et al. (2016) proposed a $\sigma$-nice function, it is unclear how special a nonconvex function the $\sigma$-nice function is. In some cases, there may be no function that satisfies the $\sigma$-nice property. Here, we decided to try to define and analyze a new family of functions with clear sufficient conditions as replacements for the $\sigma$-nice function.

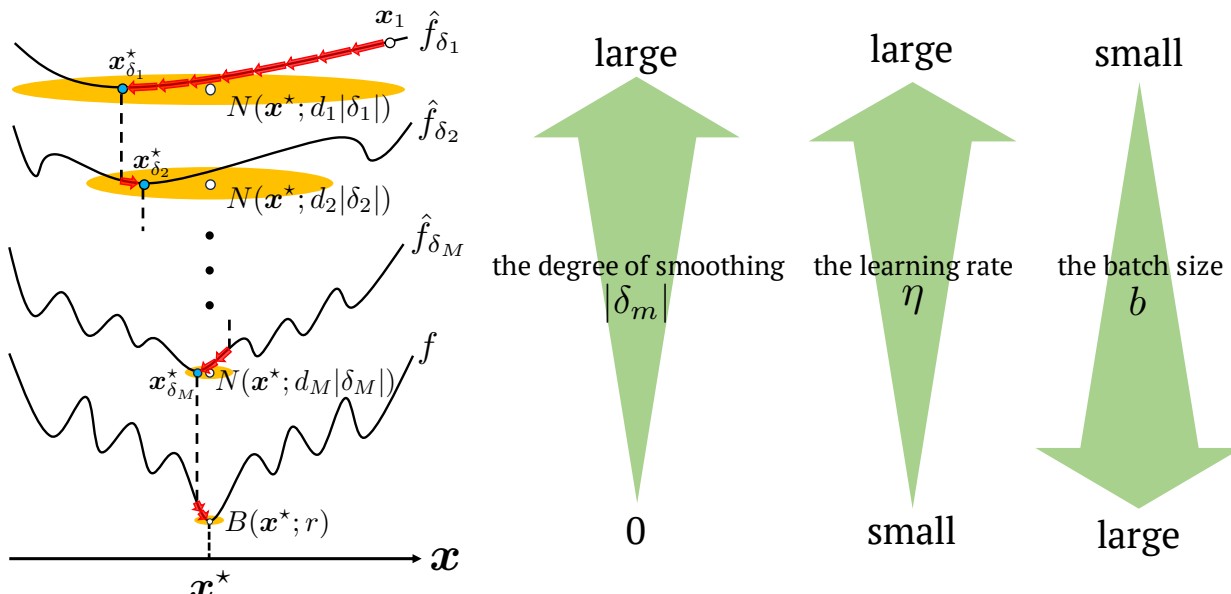

Figure 1: Conceptual diagram of new $\sigma$-nice function and its smoothed versions (see also the Notation 1).

In graduated optimization, the noise level is gradually reduced, eventually arriving at the original function, but there are an infinite number of ways to reduce the noise. For better optimization, the choice of noise scheduling is a very important issue. Therefore, we also aimed to clarify the optimal noise scheduling theoretically.

Once it is known what parameters of SGD contribute to smoothing and the optimal noise scheduling, an implicit graduated optimization can be achieved by varying the parameters so that the noise level is optimally reduced gradually. Our goal was thus to construct an implicit graduated optimization framework using the smoothing properties of SGD to achieve global optimization of deep neural networks.

### 1.3 Contributions

#### 1.3.1 SGD's Smoothing Property

We show that the degree of smoothing by SGD depends on the ratio $\frac{\eta}{\sqrt{b}}$ between the batch size and the learning rate. Accordingly, the smaller the batch size $b$ and the larger the learning rate $\eta$ are, the more smoothed the function becomes (see Figure 1). Also, we can say that halving the learning rate is the same as quadrupling the batch size. (Goyal et al., 2017; Smith et al., 2018; Xie et al., 2021) also studied SGD dynamics and demonstrated how the ratio $\frac{\eta}{b}$ affect training dynamics. Note that our theory includes and does not conflict with their results.

#### 1.3.2 Why the Use of Large Batch Sizes Leads to Solutions Falling into Sharp Local Minima

In other words, from a smoothing perspective, if we use a large batch size and/or a small learning rate, it is easy for the algorithm to fall into a sharp local minimum and experience a drop in generalization performance, since it will optimize a function that is close to the original multimodal function. As is well known, training with a large batch size leads to convergence to sharp local minima and poor generalization performance, as evidenced by the fact that several prior studies (Hoffer et al., 2017; Goyal et al., 2017; You et al., 2020) provided techniques that do not impair generalization performance even with large batch sizes. Keskar et al. (2017) showed this experimentally, and our results provide theoretical support for it.

### 1.3.3 Why Using Decaying Learning Rates and Increasing Batch Sizes is Superior to Using Fixed Ones

Moreover, we can say that decreasing the learning rate and/or increasing the batch size during training is indeed an implicit graduated optimization. Hence, using a decaying learning rate and increasing the batch size makes sense in terms of avoiding local minima. Our results provide theoretical support for the many positive findings on using decaying learning rates (Wu et al., 2014; Ioffe & Szegedy, 2015; Loshchilov & Hutter, 2017; Hundt et al., 2019; You et al., 2019; Hundt et al., 2019; Lewkowycz, 2021) and increasing batch sizes (Byrd et al., 2012; Friedlander & Schmidt, 2012; Balles et al., 2017; De et al., 2017; Bottou et al., 2018; Smith et al., 2018).

### 1.3.4 New $\sigma$-nice Function

We propose a new $\sigma$-nice function that generalizes the $\sigma$-nice function. All smoothed functions of the new $\sigma$-nice function are $\sigma$-strongly convex in a neighborhood $B(\boldsymbol{x}^\star; d_m|\delta_m|)$ of the optimal solution $\boldsymbol{x}^\star$ that is proportional to the noise level $|\delta_m|$ (see Figure 1). In contrast to Hazan et al. (2016), we show sufficient conditions for a certain nonconvex function $f$ to be a new $\sigma$-nice function as follows:

$$\frac{2L_g \max\left\{ \left\| \boldsymbol{x}^\star_{\delta_m} - \boldsymbol{x}^\star \right\|, \left\| \boldsymbol{x}^\star_{\delta_{m+1}} - \boldsymbol{x}^\star \right\| \right\}}{\sigma\left(1 - \gamma_m\right)} \leq |\delta_m| = \left| \delta_m^- \right|.$$

where $|\delta_m^-| > 0, d_{m+1} > 1, \gamma_m \in \left( \frac{1}{d_{m+1}}, 1 \right)$, and $m \in [M] \subset \mathbb{N}$. $\delta_m$ is the noise level of $\hat{f}_{\delta_m}$, which is a smoothed version of $f$, and $\boldsymbol{x}^\star$ is the global optimal solution of the original function $f$. Furthermore, we show that the graduated optimization algorithm for the $L_f$-Lipschitz new $\sigma$-nice function converges to an $\epsilon$-neighborhood of the globally optimal solution in $\mathcal{O}\left( 1/\epsilon^{\frac{1}{p}+2} \right) (p \in (0,1])$ rounds.

### 1.3.5 Optimal Noise Scheduling

Let $|\delta_m|$ be the current noise level, and let the next noise level be determined by $|\delta_{m+1}| := \gamma_m |\delta_m|$, where $\gamma_m$ is the decay rate of noise. We show theoretically that $\gamma_m$ should decay slowly from a value close to 1 for convergence to the globally optimal solution. To the best of our knowledge, ours is the first paper to provide theoretical results on optimal scheduling, both in terms of how to reduce the noise in graduated optimization and how to decay the learning rate and increase the batch size in general optimization. Noise scheduling also has an important role in score-based models (Song & Ermon, 2020), diffusion models (Chen, 2023), panoptic segmentation (Chen et al., 2023), etc., so our theoretical findings will contribute to these methodologies as well.

Furthermore, since the decay rate of noise in graduated optimization is equivalent to the decay rate of the learning rate and rate of increase in batch size, we can say that it is desirable to vary them gradually from a value close to 1. As for the schedule for decaying the learning rate, many previous studies have tried cosine annealing (without restart) (Loshchilov & Hutter, 2017), cosine power annealing (Hundt et al., 2019), or polynomial decay (Liu et al., 2015; Chen et al., 2018; Zhao et al., 2017; Chen et al., 2017), but it has remained unclear why they are superior to fixed rates. We provide theoretical support showing why they are experimentally superior. In particular, we show that a polynomial decay with a power less than or equal to 1 is the optimal learning rate schedule and demonstrate this in Section 4.

### 1.3.6 Implicit Graduated Optimization

We propose a new implicit graduated optimization algorithm. The algorithm decreases the learning rate of SGD and increases the batch size during training. We show that the algorithm for the $L_f$-Lipschitz new $\sigma$-nice function converges to an $\epsilon$-neighborhood of the globally optimal solution in $\mathcal{O}\left( 1/\epsilon^{\frac{1}{p}} \right) (p \in (0,1])$ rounds. In Section 4, we show experimentally that methods that reduce noise outperform methods that use a constant learning rate and constant batch size. We also find that methods which increase the batch size outperform those which decrease the learning rate when the decay rate of the noise is set at $1/\sqrt{2}$.

## 2 Preliminaries

### 2.1 Definitions and Notation

The notation used in this paper is summarized in Table 1.

Table 1: Notation List

| Notation | Description |
|---|---|
| $\mathbb{N}$ | The set of all nonnegative integers |
| $[N]$ | $[N] := \{1, 2, \ldots, N\}$ ($N \in \mathbb{N}\setminus\{0\}$) |
| $\mathbb{R}^d$ | A $d$-dimensional Euclidean space with inner product $\langle \cdot, \cdot \rangle$, which induces the norm $\|\cdot\|$ |
| $\mathbb{E}_\xi[X]$ | The expectation with respect to $\xi$ of a random variable $X$ |
| $\mathcal{S}_t$ | Mini-batch of $b$ samples $\boldsymbol{z}_i$ at time $t$ |
| $N(\boldsymbol{x}^\star; \epsilon)$ | $\epsilon$-neighborhood of a vector $\boldsymbol{x}^\star$, i.e., $N(\boldsymbol{x}^\star; \epsilon) := \left\{ \boldsymbol{x} \in \mathbb{R}^d \colon \|\boldsymbol{x} - \boldsymbol{x}^\star\| < \epsilon \right\}$ |
| $B(\boldsymbol{x}^\star; r)$ | The Euclidian closed ball of radius $r$ centered at $\boldsymbol{x}^\star$, i.e., $B(\boldsymbol{x}^\star; r) := \left\{ \boldsymbol{x} \in \mathbb{R}^d \colon \|\boldsymbol{x} - \boldsymbol{x}^\star\| \leq r \right\}$ |
| $\boldsymbol{u} \sim B(\boldsymbol{x}^\star; r)$ | A random variable distributed uniformly over $B(\boldsymbol{x}^\star; r)$ |
| $M$ | The number of smoothed functions, i.e., $M \in \mathbb{N}$ |
| $m$ | Counts from the smoothest function, i.e., $m \in [M]$ |
| $\delta$ | The degree of smoothing of the smoothed function, i.e., $\delta \in \mathbb{R}$ |
| $\delta_m$ | The degree of smoothing of the $m$-th smoothed function, i.e., $\delta_m \in \mathbb{R}$ |
| $\hat{f}_\delta$ | The function obtained by smoothing $f$ with a noise level $\delta$ |
| $\hat{f}_{\delta_m}$ | The $m$-th smoothed function obtained by smoothing $f$ with a noise level $\delta_m$ |
| $\boldsymbol{x}_{m+1}$ | $\boldsymbol{x}_{m+1}$ is defined by $\hat{f}_{\delta_m}(\boldsymbol{x}_{m+1}) \leq \hat{f}_{\delta_m}(\hat{\boldsymbol{x}}_t)$, where $(\hat{\boldsymbol{x}}_t)_{t=1}^{T_F+1}$ is generated by GD |
| $f_i(\boldsymbol{x})$ | A loss function for $\boldsymbol{x} \in \mathbb{R}^d$ and $\boldsymbol{z}_i$ |
| $f(\boldsymbol{x})$ | The total loss function for $\boldsymbol{x} \in \mathbb{R}^d$, i.e., $f(\boldsymbol{x}) := |\mathcal{S}|^{-1} \sum_{i \in \mathcal{S}} f_i(\boldsymbol{x})$ |
| $\xi$ | A random variable supported on $\Xi$ that does not depend on $\boldsymbol{x} \in \mathbb{R}^d$ |
| $\xi_t$ | $\xi_0, \xi_1, \ldots$ are independent samples and $\xi_t$ is independent of $(\boldsymbol{x}_k)_{k=0}^t \subset \mathbb{R}^d$ |
| $\xi_{t,i}$ | A random variable generated from the $i$-th sampling at time $t$ |
| $\mathsf{G}_{\xi_t}(\boldsymbol{x})$ | The stochastic gradient of $f(\cdot)$ at $\boldsymbol{x} \in \mathbb{R}^d$ |
| $\nabla f_{\mathcal{S}_t}(\boldsymbol{x}_t)$ | The mini-batch stochastic gradient of $f(\boldsymbol{x}_t)$ for $\mathcal{S}_t$, i.e., $\nabla f_{\mathcal{S}_t}(\boldsymbol{x}_t) := b^{-1} \sum_{i \in [b]} \mathsf{G}_{\xi_{t,i}}(\boldsymbol{x}_t)$ |

**Definition 2.1** (Smoothed function)**.** *Given an $L_f$-Lipschitz function $f$, define $\hat{f}_\delta$ to be the function obtained by smoothing $f$ as*

$$\hat{f}_\delta(\boldsymbol{x}) := \mathbb{E}_{\boldsymbol{u} \sim B(\boldsymbol{0};1)}\left[f(\boldsymbol{x} - \delta\boldsymbol{u})\right], \tag{2}$$

*where $\delta \in \mathbb{R}$ represents the degree of smoothing and $\boldsymbol{u}$ is a random variable distributed uniformly over $B(\boldsymbol{0}; 1)$. Also,*

$$\boldsymbol{x}^\star := \underset{\boldsymbol{x} \in \mathbb{R}^d}{\arg\min} f(\boldsymbol{x}) \quad and \quad \boldsymbol{x}_\delta^\star := \underset{\boldsymbol{x} \in \mathbb{R}^d}{\arg\min} \hat{f}_\delta(\boldsymbol{x}).$$

**Remark:** For a general smoothing as in Definition 2.1, the distribution followed by the random variable $\boldsymbol{u}$ need not necessarily be uniform; it can be a normal distribution. In fact, several previous studies (Wu, 1996; Iwakiri et al., 2022) assumed that $\boldsymbol{u}$ follows a normal distribution. Here, we assume that it follows a uniform distribution because this is necessary for the analysis of the new $\sigma$-nice function. This is also true for the analysis of the $\sigma$-nice function (Hazan et al., 2016).

There are a total of $M$ smoothed functions in this paper. The largest noise level is $\delta_1$ and the smallest noise level is $\delta_{M+1} = 0$. Thus, $\hat{f}_{\delta_{M+1}} = f$.

**Definition 2.2** ($\sigma$-nice function (Hazan et al., 2016))**.** *A function $f \colon \mathbb{R}^d \to \mathbb{R}$ is said to be $\sigma$-nice if the following two conditions hold:*

*(i)* *For every $\delta > 0$ and every $\boldsymbol{x}_\delta^\star$, there exists $\boldsymbol{x}_{\delta/2}^\star$ such that:*

$$\left\|\boldsymbol{x}_\delta^\star - \boldsymbol{x}_{\delta/2}^\star\right\| \leq \frac{\delta}{2}.$$

*(ii) For every $\delta > 0$, let $r_\delta = 3\delta$; then, the function $\hat{f}_\delta(\boldsymbol{x})$ over $N(\boldsymbol{x}_\delta^\star; r_\delta)$ is $\sigma$-strongly convex.*

The $\sigma$-nice property implies that optimizing the smoothed function $\hat{f}_\delta$ is a good start for optimizing the next smoothed function $\hat{f}_{\delta/2}$, which has been shown to be sufficient for graduated optimization (Hazan et al., 2016).

## 2.2 Assumptions and Lemmas

We make the following assumptions:

**Assumption 2.1.** *(A1) $f\colon \mathbb{R}^d \to \mathbb{R}$ is continuously differentiable and $L_g$-smooth, i.e., for all $\boldsymbol{x}, \boldsymbol{y} \in \mathbb{R}^d$,*

$$\|\nabla f(\boldsymbol{x}) - \nabla f(\boldsymbol{y})\| \le L_g \|\boldsymbol{x} - \boldsymbol{y}\|.$$

*(A2) $f\colon \mathbb{R}^d \to \mathbb{R}$ is $L_f$-Lipschitz function, i.e., for all $\boldsymbol{x}, \boldsymbol{y} \in \mathbb{R}^d$,*

$$|f(\boldsymbol{x}) - f(\boldsymbol{y})| \le L_f \|\boldsymbol{x} - \boldsymbol{y}\|.$$

*(A3) Let $(\boldsymbol{x}_t)_{t\in\mathbb{N}} \subset \mathbb{R}^d$ be the sequence generated by SGD.*

*(i) For each iteration $t$,*

$$\mathbb{E}_{\xi_t}\left[\mathsf{G}_{\xi_t}(\boldsymbol{x}_t)\right] = \nabla f(\boldsymbol{x}_t).$$

*(ii) There exists a nonnegative constant $C^2$ such that*

$$\mathbb{E}_{\xi_t}\left[\|\mathsf{G}_{\xi_t}(\boldsymbol{x}_t) - \nabla f(\boldsymbol{x}_t)\|^2\right] \le C^2.$$

*(A4) For each iteration $t$, SGD samples a mini-batch $\mathcal{S}_t \subset \mathcal{S}$ and estimates the full gradient $\nabla f$ as*

$$\nabla f_{\mathcal{S}_t}(\boldsymbol{x}_t) := \frac{1}{b} \sum_{i\in[b]} \mathsf{G}_{\xi_{t,i}}(\boldsymbol{x}_t) = \frac{1}{b} \sum_{\{i\colon \boldsymbol{z}_i \in \mathcal{S}_t\}} \nabla f_i(\boldsymbol{x}_t).$$

**Lemma 2.1.** *Suppose that (A3)(ii) and (A4) hold for all $t \in \mathbb{N}$; then,*

$$\mathbb{E}_{\xi_t}\left[\|\nabla f_{\mathcal{S}_t}(\boldsymbol{x}_t) - \nabla f(\boldsymbol{x}_t)\|^2\right] \le \frac{C^2}{b}.$$

The proof of Lemma 2.1 can be found in Appendix B.1.

The following Lemmas concern the properties of smoothed functions $\hat{f}_\delta$. See Appendix B for their proofs.

**Lemma 2.2.** *Suppose that (A1) holds; then, $\hat{f}_\delta$ defined by (2) is also $L_g$-smooth; i.e., for all $\boldsymbol{x}, \boldsymbol{y} \in \mathbb{R}^d$,*

$$\left\|\nabla \hat{f}_\delta(\boldsymbol{x}) - \nabla \hat{f}_\delta(\boldsymbol{y})\right\| \le L_g \|\boldsymbol{x} - \boldsymbol{y}\|.$$

**Lemma 2.3.** *Suppose that (A2) holds; then $\hat{f}_\delta$ is also an $L_f$-Lipschitz function; i.e., for all $\boldsymbol{x}, \boldsymbol{y} \in \mathbb{R}^d$,*

$$\left|\hat{f}_\delta(\boldsymbol{x}) - \hat{f}_\delta(\boldsymbol{y})\right| \le L_f \|\boldsymbol{x} - \boldsymbol{y}\|.$$

Lemmas 2.2 and 2.3 imply that the Lipschitz constants $L_f$ of the original function $f$ and $L_g$ of $\nabla f$ are taken over by the smoothed function $\hat{f}_\delta$ and its gradient $\nabla \hat{f}_\delta$ for all $\delta \in \mathbb{R}$.

**Lemma 2.4.** *Let $\hat{f}_\delta$ be the smoothed version of $f$; then, for all $\boldsymbol{x} \in \mathbb{R}^d$,*

$$\left|\hat{f}_\delta(\boldsymbol{x}) - f(\boldsymbol{x})\right| \le |\delta| L_f.$$

Lemma 2.4 implies that the larger the degree of smoothing is, the further away the smoothed function is from the original function. Since the degree of smoothing is determined by the learning rate and batch size (see Section 3.3), we can say that the optimal value obtained by using a large learning rate and/or small batch size may be larger than the optimal value obtained by using a small learning rate and/or large batch size. When decreasing the learning rate or increasing the batch size, the sharp decrease in function values at that time depends on the change in the objective function (see also Figure 1), and this phenomenon is especially noticeable in schedules that use the same noise level for multiple epochs, such as the step decay learning rate (see Figures 7-9).

## 3 Main Results

### 3.1 New $\sigma$-nice function

Since the definition of the $\sigma$-nice function is inappropriate for large noise levels (see Section 3.2), we generalize the $\sigma$-nice function and define a new $\sigma$-nice function that can be defined even when the noise level is large.

**Definition 3.1.** *Let $\delta_1 \in \mathbb{R}$. A function $f \colon \mathbb{R}^d \to \mathbb{R}$ is said to be "new $\sigma$-nice" if the following two conditions hold :*

*(i) For all $m \in [M]$ and all $\gamma_m \in (0,1)$, there exist $\delta_m \in \mathbb{R}$ with $|\delta_{m+1}| := \gamma_m |\delta_m|$ and $\boldsymbol{x}^\star_{\delta_m}$ such that*

$$\left\| \boldsymbol{x}^\star_{\delta_m} - \boldsymbol{x}^\star_{\delta_{m+1}} \right\| \leq |\delta_m| - |\delta_{m+1}|.$$

*(ii) For all $m \in [M]$ and all $\gamma_m \in (0,1)$, there exist $\delta_m \in \mathbb{R}$ with $|\delta_{m+1}| := \gamma_m |\delta_m|$ and $d_m > 1$ such that the function $\hat{f}_{\delta_m}(\boldsymbol{x})$ is $\sigma$-strongly convex on $N(\boldsymbol{x}^\star; d_m \delta_m)$.*

The value $\delta_m \in \mathbb{R}$ in Definition 3.1 is the degree of smoothing or noise level (see Definition 2.1) and $\gamma_m \in (0,1)$ is the decay rate of the noise, i.e., $\gamma_m := |\delta_{m+1}|/|\delta_m|$. In the definition of the $\sigma$-nice function (Definition 2.2), $\gamma_m$ is always 0.5. We have extended this condition to $\gamma_m \in (0,1)$. We can show that, for the graduated optimization algorithm to be successful, $\gamma_m$ requires a certain lower bound, which provides important insights into the optimal noise scheduling (see Section 3.2).

The next propositions provide a sufficient condition for the function $f$ to be a new $\sigma$-nice function. The proofs of Propositions 3.1 and 3.2 are in Section D.5 and D.6, respectively.

**Proposition 3.1.** *Let $a_m > \sqrt{2}$ for all $m \in [M]$. Suppose that the function $f \colon \mathbb{R}^d \to \mathbb{R}$ is $\sigma$-strongly convex on $B(\boldsymbol{x}^\star; r)$ for sufficiently small $r > 0$ and the noise level $|\delta_m|$ satisfies $|\delta_m| = |\delta_m^-|$; then, the smoothed function $\hat{f}_{\delta_m}$ is $\sigma$-strongly convex on $N(\boldsymbol{x}^\star; a_m r)$, where*

$$|\delta_m^-| := \sup_{\boldsymbol{x} \in N(\boldsymbol{x}^\star; a_m r) \setminus \{\boldsymbol{x}^\star\}} \mathbb{E}_{\boldsymbol{u}_m \sim B(\boldsymbol{0}; 1)} \left[ \left| \|\boldsymbol{x}^\star - \boldsymbol{x}\| \|\boldsymbol{u}_m\| \cos\theta - \sqrt{\|\boldsymbol{x}^\star - \boldsymbol{x}\|^2 \|\boldsymbol{u}_m\|^2 \cos^2\theta - r^2(a_m^2 - 1)} \right| \right],$$

*and $\theta$ is the angle between $\boldsymbol{u}_m$ and $\boldsymbol{x}^\star - \boldsymbol{x}$.*

*Also, if we define $d_m$ as $d_m := a_m r / |\delta_m^-|$, then the smoothed function $\hat{f}_{\delta_m}$ is also $\sigma$-strongly convex on $N(\boldsymbol{x}^\star; d_m |\delta_m|)$.*

Note that $\boldsymbol{u}_m \in \mathbb{R}^d$ is a random variable used to define the smoothed function $\hat{f}_{\delta_m}$, which we assume follows a uniform distribution (see Definition 2.1). In addition, $a_m \in \mathbb{R}$ is only required for the analysis and $(a_m)_{m \in [M]}$ is monotone decreasing. Proposition 3.1 implies that the radius of the strongly convex region of the function $\hat{f}_{\delta_m}$ extends to $a_m r$ if the sequence of noise $(\delta_m)_{m \in [M]}$ added to the function $f$ satisfies $|\delta_m| = |\delta_m^-|$ for all $m \in [M]$. Thus, if $a_m r \geq d_m |\delta_m|$ holds, then the smoothed function $\hat{f}_{\delta_m}$ is also strongly convex in the neighborhood $N(\boldsymbol{x}^\star; d_m |\delta_m|)$. Therefore, we define $d_m \in \mathbb{R}$ as $d_m := a_m r / |\delta_m^-|$.

Now, let us discuss $d_m$. From the definition of $d_m$, the lower and upper bounds of $d_m$ can be expressed as

$$1 < d_m \leq \frac{a_m}{\sqrt{a_m^2 - 1} - 1}. \tag{3}$$

Thus, the upper bound of $d_m$ gradually increases as $a_m$ decreases. The $\sigma$-nice function (Hazan et al., 2016) always assumes $d_m = 3$, but we see that this does not hold when $a_m$ is large (see Figure 2 in Section 3.2).

**Proposition 3.2.** *Let $d_m > 1$ for all $m \in [M]$. Suppose that the function $f : \mathbb{R}^d \to \mathbb{R}$ is $\sigma$-strongly convex and $L_g$-smooth on $B(\boldsymbol{x}^\star; r)$ for sufficiently small $r > 0$; a sufficient condition for $f$ to be a new $\sigma$-nice function is that the noise level $|\delta_m|$ satisfies the following condition :*

*For all $m \in [M]$, suppose that $\boldsymbol{x}^\star_{\delta_{m-1}} \in N(\boldsymbol{x}^\star; d_m|\delta_m|)$,*

$$\frac{2L_g \max\left\{\left\|\boldsymbol{x}^\star_{\delta_m} - \boldsymbol{x}^\star\right\|, \left\|\boldsymbol{x}^\star_{\delta_{m+1}} - \boldsymbol{x}^\star\right\|\right\}}{\sigma\left(1 - \gamma_m\right)} \leq |\delta_m| = \left|\delta_m^-\right|. \tag{4}$$

Proposition 3.2 shows that any function is a new $\sigma$-nice function if $|\delta_m|$ satisfies equations (4). Note that $\delta_m^-$ does not always exist. The probability $p(a_m)$ that $\delta_m^-$ exists depends on the direction of the random variable vector $\boldsymbol{u}_m$ and can be expressed as

$$0 < p(a_m) := \frac{\arccos\left(\dfrac{r\sqrt{a_m^2 - 1}}{\|\boldsymbol{x}^\star - \boldsymbol{x}\|\|\boldsymbol{u}_m\|}\right)}{\pi} < \frac{\arccos\left(\dfrac{\sqrt{a_m^2 - 1}}{a_m}\right)}{\pi} < 1,$$

where $r > 0, a_m > \sqrt{2}, \boldsymbol{x} \in N(\boldsymbol{x}^\star; a_m r) \backslash \{\boldsymbol{x}^\star\}$. Since the upper bound of $p(a_m)$ approaches 0 when $a_m$ is large, the probability $p(a_m)$ approaches 0 as $a_m$ gets larger, but it never reaches 0. Therefore, the success of Algorithm 1 depends on the random variable $\boldsymbol{u}_m$, especially when $a_m$ is large, i.e., when $\delta_m$ is large.

The framework of graduated optimization for the new $\sigma$-nice function is shown in Algorithm 1. Algorithm 2 is used to optimize each smoothed function.

---

**Algorithm 1** Graduated Optimization

**Require:** $\epsilon > 0, r \in (0,1), p \in (0,1], \bar{d} > 0, \boldsymbol{x}_1, B_2 > 0$
    $\delta_1 := \frac{2L_g}{\sigma r}$
    $\alpha_0 := \min\left\{\frac{\sigma r}{8L_f^2(1+\bar{d})}, \frac{\sqrt{\sigma} r}{2\sqrt{2}L_g}\right\}, M^p := \frac{1}{\alpha_0 \epsilon}$
    **for** $m = 1$ to $M + 1$ **do**
      **if** $m \neq M + 1$ **then**
        $\epsilon_m := \sigma \delta_m^2, T_F := 2B_2/\sigma \epsilon_m$
        $\gamma_m := \frac{(M-m)^p}{\{M-(m-1)\}^p}$
      **end if**
      $\boldsymbol{x}_{m+1} := \text{GD}(T_F, \boldsymbol{x}_m, \hat{f}_{\delta_m})$
      $\delta_{m+1} := \gamma_m \delta_m$
    **end for**
    **return** $\boldsymbol{x}_{M+2}$

---

**Algorithm 2** GD with decaying learning rate

**Require:** $T_F, \hat{\boldsymbol{x}}_1, F$
    **for** $t = 1$ to $T_F$ **do**
      $\eta_t := 2/\sigma t$
      $\hat{\boldsymbol{x}}_{t+1} := \hat{\boldsymbol{x}}_t - \eta_t \nabla F(\hat{\boldsymbol{x}}_t)$
    **end for**
    **return** $\hat{\boldsymbol{x}}_{T_F+1} = \text{GD}(T_F, \hat{\boldsymbol{x}}_1, F)$

---

The smoothed function $\hat{f}_{\delta_m}$ is $\sigma$-strongly convex in the neighborhood $N(\boldsymbol{x}^\star; d_m|\delta_m|)$. Thus, we should now consider the convergence of GD for a $\sigma$-strongly convex function $F = \hat{f}_{\delta_m}$. Theorem 3.1 is a convergence analysis for when a decaying learning rate is used (The proof of Theorem 3.1 is in Section D.1).

**Theorem 3.1** (Convergence analysis of Algorithm 2)**.** *Suppose that* $F \colon \mathbb{R}^d \to \mathbb{R}$ *is a $\sigma$-strongly convex and $L_g$-smooth function and $\eta_t := \frac{2}{\sigma t}$. Then, the sequence $(\hat{\boldsymbol{x}}_t)_{t \in \mathbb{N}}$ generated by Algorithm 2 satisfies*

$$\min_{t \in [T]} \left( F\left(\hat{\boldsymbol{x}}_t\right) - F(\boldsymbol{x}^\star) \right) \leq \frac{2B_2}{\sigma T} = \mathcal{O}\left(\frac{1}{T}\right), \tag{5}$$

*where $\boldsymbol{x}^\star$ is the global minimizer of $F$, and $B_2 > 0$ is a nonnegative constant.*

Theorem 3.1 is the convergence analysis of Algorithm 2 for any $\sigma$-strongly convex function $F$. It shows that the algorithm can reach an $\epsilon_m$-neighborhood of the optimal solution $\boldsymbol{x}_{\delta_m}^\star$ of $\hat{f}_{\delta_m}$ in approximately $T_F := 2B_2/\sigma\epsilon_m$ iterations.

**Remark:** Algorithms 1 and 2 represent explicit graduated optimization algorithms. Function smoothing is accomplished explicitly by convolving random variables as in Definition 2.1, and the smoothed strongly convex function is optimized by the gradient descent (Algorithm 2). However, in general, the integral operation of the function $f$ is not possible, so optimization by Algorithms 1 and 2 is not feasible. If smoothing of the function $f$ by Definition 2.1 is possible and $\hat{f}_\delta$ is accessible, then Algorithm 2 may be SGD-type optimizer. For example, Algorithm 2 can be the projected SGD generated by the sequence $(\hat{\boldsymbol{x}}_t)$ with $\hat{\boldsymbol{x}}_{t+1} = P_m(\hat{\boldsymbol{x}}_t - \eta_t \nabla F_{S_t}(\hat{\boldsymbol{x}}_t))$, where $P_m$ is the projection onto $B(\boldsymbol{x}^\star; d_m \delta_m)$. Since $\hat{\boldsymbol{x}}_0 = \boldsymbol{x}_{\delta_{m-1}}^\star \in B(\boldsymbol{x}^\star; d_m \delta_m)$ is guaranteed by Proposition 3.3(ii), the sequence $(\hat{\boldsymbol{x}}_t)$ generated by the projected SGD is always in $B(\boldsymbol{x}^\star; d_m \delta_m)$. Using the proof of Theorem 3.1 and the nonexpansivity of $P_m$ (i.e., $\|P_m(\boldsymbol{x}) - P_m(\boldsymbol{y})\| \leq \|\boldsymbol{x} - \boldsymbol{y}\|$), we can show that the projected SGD satisfies

$$\min_{t \in [T]} \mathbb{E}\left[F(\hat{\boldsymbol{x}}_t) - F(\boldsymbol{x}^\star)\right] \leq \frac{2D_2}{\sigma T},$$

where $D_1 := \sup_{t \in \mathbb{N}} \mathbb{E}\left[\|\nabla F(\hat{\boldsymbol{x}}_t)\|^2\right]$ and $D_2 := C^2 + D_1$.

The next theorem guarantees the convergence of Algorithm 1 for the new $\sigma$-nice function (The proof of Theorem 3.2 is in Section D.2).

**Theorem 3.2** (Convergence analysis of Algorithm 1)**.** *Let $\epsilon \in (0, 1)$ and $f$ be an $L_f$-Lipschitz new $\sigma$-nice function. Suppose that we apply Algorithm 1; then, after $\mathcal{O}\left(1/\epsilon^{\frac{1}{p}+2}\right)$ rounds, the algorithm reaches an $\epsilon$-neighborhood of the global optimal solution $\boldsymbol{x}^\star$.*

**Remark:** In Algorithm 1 and Theorem 3.2, we assume that we can access the full gradient of the smoothed function $\nabla \hat{f}_{\delta_m}$. Thus, our explicit graduated optimization by Algorithms 1 and 2 is only valid for functions $f$ for which the computation of $\hat{f}_{\delta_m}$ by Definition 2.1 and the access to its full gradient $\nabla \hat{f}_{\delta_m}$ are possible. Hence, Algorithm 1 and 2 are not applicable to DNN.

Note that Theorem 3.2 provides a total complexity that integrates Algorithm 1 and Algorithm 2 because Algorithm 1 uses Algorithm 2 at each $m \in [M]$. Theorem 3.2 implies that convergence is faster when the power of the polynomial decay $p$ is large, and when $p = 1$, it takes at least $\mathcal{O}\left(1/\epsilon^3\right)$ rounds for new $\sigma$-nice functions. Hazan et al. (2016) showed that their graduated optimization algorithm converges to a globally optimal solution in $\mathcal{O}\left(1/\epsilon^2\right)$ iterations for a $\sigma$-nice function. However, explicit graduated optimization, such as with our Algorithm 1 and Algorithm 1 in Hazan et al. (2016), is not applicable to DNN due to the impossibility of computing a smoothed function $\hat{f}_{\delta_m}$.

## 3.2 Optimal Noise Scheduling

The next proposition is crucial to the success of Algorithm 1 (The proof is in Section D.7).

**Proposition 3.3.** *Let $d_m > 1$ for all $m \in [M]$ and suppose that $f \colon \mathbb{R}^d \to \mathbb{R}$ is a new $\sigma$-nice function.*

*(i) Then, the following always holds for all $m \in [M]$,*

$$\left\|\boldsymbol{x}_{\delta_m}^\star - \boldsymbol{x}^\star\right\| < d_m |\delta_m|,$$

*(ii) If, $\gamma_m$ satisfies $\gamma_m \in \left(\frac{1}{d_{m+1}}, 1\right)$ for all $m \in [M]$, then the following holds for all $m \in \{2, 3, \cdots, M\}$,*

$$\left\| \boldsymbol{x}^\star_{\delta_{m-1}} - \boldsymbol{x}^\star \right\| < d_m |\delta_m|.$$

Proposition 3.3 (i) implies that if the objective function $f$ is the new $\sigma$-nice function, the optimal solution $\boldsymbol{x}^\star_{\delta_m}$ of the smoothed function $\hat{f}_{\delta_m}$ is always contained in the $\sigma$-strongly convex region $N(\boldsymbol{x}^\star; d_m|\delta_m|)$ of the function $\hat{f}_{\delta_m}$. Therefore, if the initial point of the optimization of the function $\hat{f}_{\delta_m}$ is contained in the $\sigma$-strongly convex region $N(\boldsymbol{x}^\star; d_m|\delta_m|)$, the sequence generated by Algorithm 2 never leaves the $\sigma$-strongly convex region. Also, assuming that Algorithm 2 comes sufficiently close to the optimal solution $\boldsymbol{x}^\star_{\delta_{m-1}}$ after more than $T_F$ iterations in the optimization of the $\hat{f}_{\delta_{m-1}}$, $\boldsymbol{x}^\star_{\delta_{m-1}}$ is the initial point of the optimization of the next function $\hat{f}_{\delta_m}$. Proposition 3.3 (ii) therefore implies that the initial point of optimization of the function $\hat{f}_{\delta_m}$ is contained in the $\sigma$-strongly convex region of the function $\hat{f}_{\delta_m}$. Hence, Proposition 3.3 guarantees that if the initial point $\bar{\boldsymbol{x}}_1$ of Algorithm 1 is contained in the $\sigma$-strongly convex region $N(\boldsymbol{x}^\star; d_1|\delta_1|)$ of the smoothest function $\hat{f}_{\delta_1}$, then the algorithm will always reach the globally optimal solution $\boldsymbol{x}^\star$ of the original function $f$. Note that the decay rate $\gamma_m$ used in Algorithm 1 satisfies $\gamma_m \in (1/d_{m+1}, 1)$. See the following discussion.

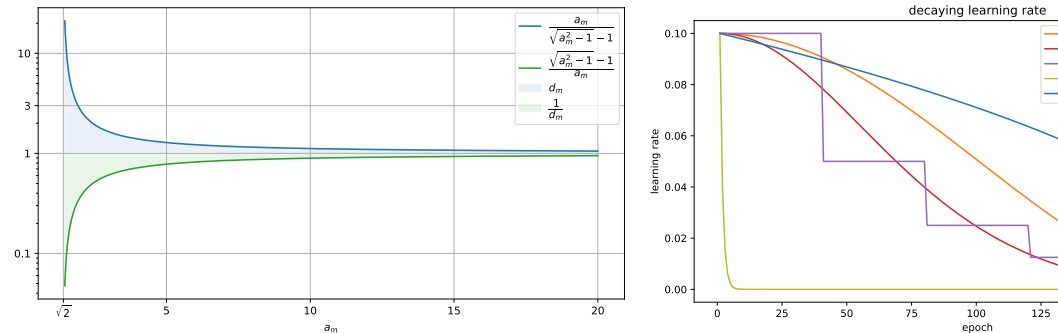

Figure 2: The ranges of possible values for $d_m$ and $1/d_m$ are colored blue and green, respectively. Note that the vertical axis is logarithmic.

Figure 3: Existing decay learning rate versus number of epochs. The schedule definitions are included in equations (6) through (7).

According to Proposition 3.1, for a function to be a new $\sigma$-nice function, $d_m$ must satisfy equation (3). Thus, there is a range of possible values for $1/d_m$:

$$1 < d_m \leq \frac{a_m}{\sqrt{a_m^2 - 1} - 1} \text{ and } \frac{\sqrt{a_m^2 - 1} - 1}{a_m} \leq \frac{1}{d_m} < 1.$$

The range is plotted in Figure 2. Recall that $a_m$ is a value that appears only in the theoretical analysis and it becomes smaller as $m$ increases and $\delta_m$ decreases, since it satisfies $d_m|\delta_m| = a_m r$.

$d_m$ is involved in the radius of the strongly convex region $N(\boldsymbol{x}^\star; d_m|\delta_m|)$ of the smoothed function $\hat{f}_{\delta_m}$. According to Figure 2, when $a_m$ is large, i.e., when $m$ is small and $|\delta_m|$ is large, $d_m$ can only take almost 1. From the definition of a $\sigma$-nice function (Hazan et al., 2016) (see Definition 2.2), a smoothed function $\hat{f}_{\delta_m}$ is strongly convex in a neighborhood $N(\boldsymbol{x}^\star_{\delta_m}; 3\delta_m)$. Then, since $\boldsymbol{x}^\star_{\delta_m}$ is always contained in $N(\boldsymbol{x}^\star; d_m|\delta_m|)$ (see Proposition 3.3), we see that $d_m = 3$ does not always hold. That is, a $\sigma$-nice function cannot be defined when the noise level is large.

From Proposition 3.3 and its proof, for Algorithm 1 to be successful, $1/d_{m+1}$ is required as a lower bound for $\gamma_m$, i.e., $\gamma_m \in \left(\frac{1}{d_{m+1}}, 1\right)$. Recall that $\gamma_m$ is the decay rate of the noise level $|\delta_m|$, i.e., $|\delta_{m+1}| := \gamma_m |\delta_m|$. According to Figure 2, when $a_m$ is large, i.e., when $m$ is small and $|\delta_m|$ is large, $1/d_m$ and $\gamma_m$ can only

take almost 1. Therefore, $\gamma_m$ should vary very gradually from a value close to 1. From the definition of a $\sigma$-nice function (see Definition 2.2), $\gamma_m$ is always 0.5. When the noise level is large, a small decay rate such as 0.5 cannot be used, so the definition of the $\sigma$-nice function is still not appropriate when the noise level is large. Even when the noise level is large, our new $\sigma$-nice function can satisfy the conditions (Proposition 3.3) necessary for the success of Algorithm 1 because the radius $d_m$ of the strongly convex region and the decay rate $\gamma_m$ vary with the noise level $\delta_m$.

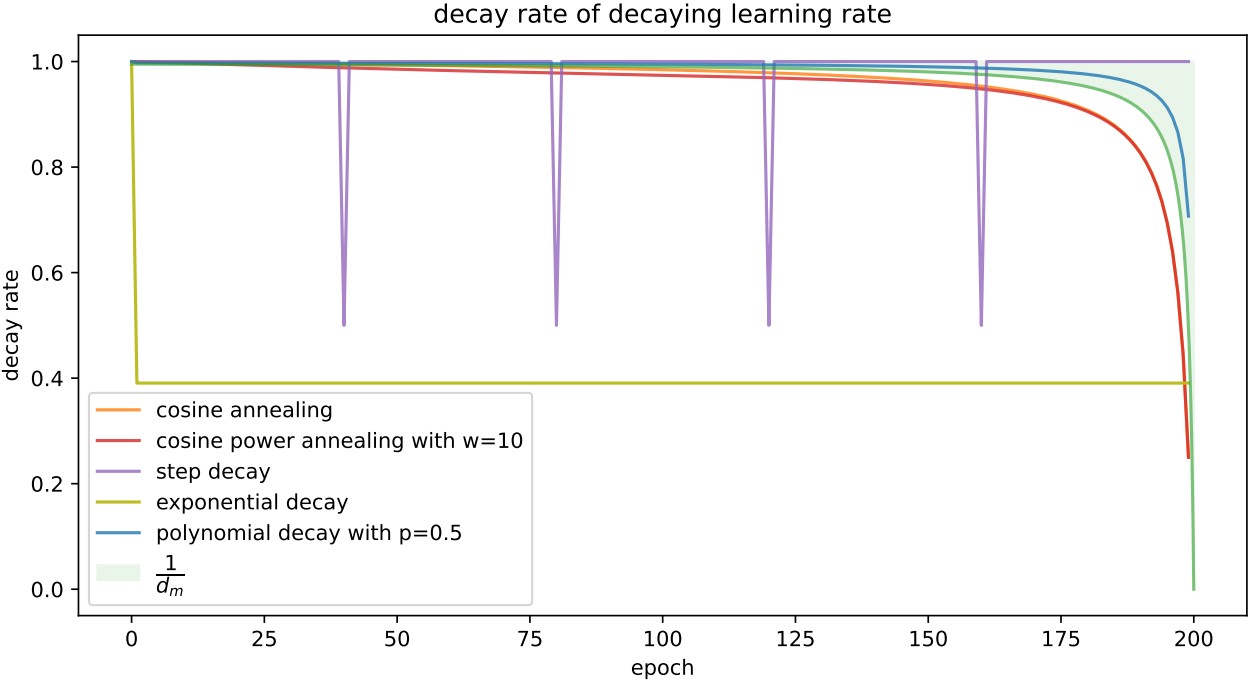

Figure 4: Decay rate of the existing decaying learning rate schedule. The area colored green represents the value that the decay rate $\gamma_m$ must satisfy for the graduated optimization approach to succeed. Here, the green curve in this figure is a symmetric shift and parallel shift of the green curve in Figure 2 to zero at epoch 200.

Now let us see if there is a decaying learning rate schedule that satisfies the decay rate $\gamma_m$ condition. The existing decaying learning rate schedule is shown in Figure 3 (Methods that include an increase in the learning rate, even partially, such as warm-up, are omitted). The following defines the update rules for all decaying learning rates $(\eta_t)_{t \in \{0,1,\cdots,T-1\}}$, where $T$ means the number of epochs.

$$\text{cosine annealing (Loshchilov \& Hutter, 2017): } \eta_t := \eta_{\min} + \frac{1}{2}\left(\eta_{\max} - \eta_{\min}\right)\left(1 + \cos\left(\frac{t}{T}\pi\right)\right) \tag{6}$$

$$\text{cosine power annealing (Hundt et al., 2019): } \eta_t := \eta_{\min} + \left(\eta_{\max} - \eta_{\min}\right)\frac{w^{\frac{1}{2}\left(1+\cos\left(\frac{t}{T}\pi\right)\right)+1} - w}{w^2 - w} \quad (w > 0)$$

$$\text{step decay (Lu, 2022): } \eta_t := \eta_{\max}d^{\lfloor \frac{t}{n} \rfloor} \ (0 < d < 1, n < T)$$

$$\text{exponential decay (Wu et al., 2014): } \eta_t := \eta_{\max}\exp\left(-kt\right) \ (k > 0)$$

$$\text{polynomial decay (Chen et al., 2018): } \eta_t := \left(\eta_{\max} - \eta_{\min}\right)\left(1 - \frac{t}{T}\right)^p + \eta_{\min} \ (p > 0) \tag{7}$$

The curves in Figure 3 are plotted for $T = 200, \eta_{\min} = 0, \eta_{\max} = 0.1, d = 0.5, n = 40, k = 0.94, w = 10$ and $p = 0.5$. The decay rates of these schedules are plotted in Figure 4. Figure 5 and Figure 6 are for polynomial decays with different parameters $p$. Note that $\eta_{\min}$ is 0, but since $t \in [0, 1, \cdots, T-1]$, $\eta_t$ will never be 0 under any update rule. In Figure 4 and 6, only the first one is set to 1 artificially. Also, the value shown at epoch $t$ represents the rate of decay from the learning rate used in epoch $t$ to the learning rate used in epoch $t + 1$. Therefore, the graphs stop at 199 epochs.

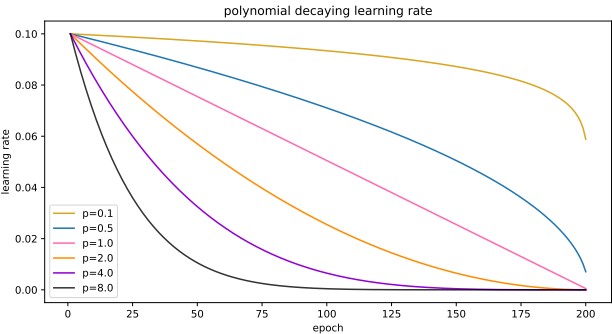
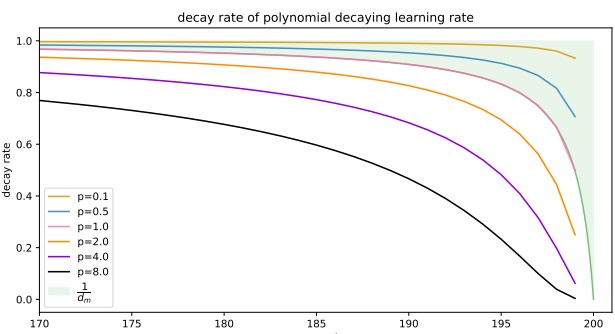

Figure 5: Polynomial decay learning rate versus epoch. The update rule for polynomial decay is defined by (7).

Figure 6: Decay rate of polynomial decay learning rate schedule. The area colored green represents the value that the decay rate $\gamma_m$ must satisfy for the graduated optimization approach to succeed.

According to Figure 4 and Figure 6, only a polynomial decay with small power $p$ satisfies the conditions that $\gamma_m$ must satisfy.

Finally, we would like to mention something about warm-up techniques (Radford et al., 2018; Liu et al., 2018; Gotmare et al., 2019). Although warm-up techniques that increase the learning rate in the early stages of learning are very popular, they are a distraction from the discussion of the decay rates shown in Figures 4 and 6; hence, we have focused on monotonically decreasing learning rates in this paper. Since the learning rate determines the smoothing level of the function, increasing the learning rate in the early learning phase, with a fixed batch size, means temporarily smoothing the function significantly and exploring that function with a large learning rate. Therefore, we can say that the warm-up technique is a reasonable scheduling that, as conventionally understood, defines the best starting point. However, we should also note that, since Algorithm 3 assumes that the learning rate is monotonically decreasing, Theorem 3.4 may not hold if the warm-up technique is used.

### 3.3 SGD's smoothing property

This section discusses the smoothing effect of using stochastic gradients. From Lemma 2.1, we have

$$\mathbb{E}_{\xi_t}\left[\|\boldsymbol{\omega}_t\|\right] \leq \frac{C}{\sqrt{b}},$$

due to $\boldsymbol{\omega}_t := \nabla f_{\mathcal{S}_t}(\boldsymbol{x}_t) - \nabla f(\boldsymbol{x}_t)$. The $\boldsymbol{\omega}_t$ for which this equation is satisfied can be expressed as $\boldsymbol{\omega}_t = \frac{C}{\sqrt{b}}\boldsymbol{u}_t,$ where $\boldsymbol{u}_t \sim \mathcal{N}\left(\mathbf{0}; \frac{1}{\sqrt{d}}I_d\right), \mathcal{N}\left(\mathbf{0}; \frac{1}{\sqrt{d}}I_d\right)$ is a normal distribution with mean $\mathbf{0}$ and variance-covariance matrix $\frac{1}{\sqrt{d}}I_d$, and $I_d$ denotes the identity matrix in $\mathbb{R}^d$. Note that, according to (Zhang et al., 2020), for some deep learning models and datasets, the stochastic noise follows a normal distribution. Based on this, we assume that the stochastic noise follows a normal distribution. Then, using Definition 2.1, we further transform equation (1) as follows:

$$\begin{aligned} \mathbb{E}_{\boldsymbol{\omega}_t}\left[\boldsymbol{y}_{t+1}\right] &= \mathbb{E}_{\boldsymbol{\omega}_t}\left[\boldsymbol{y}_t\right] - \eta\nabla\mathbb{E}_{\boldsymbol{\omega}_t}\left[f(\boldsymbol{y}_t - \eta\boldsymbol{\omega}_t)\right] \\ &= \mathbb{E}_{\boldsymbol{\omega}_t}\left[\boldsymbol{y}_t\right] - \eta\nabla\mathbb{E}_{\boldsymbol{u}_t\sim\mathcal{N}\left(\mathbf{0};\frac{1}{\sqrt{d}}I_d\right)}\left[f\left(\boldsymbol{y}_t - \frac{\eta C}{\sqrt{b}}\boldsymbol{u}_t\right)\right] \\ &\approx \mathbb{E}_{\boldsymbol{\omega}_t}\left[\boldsymbol{y}_t\right] - \eta\nabla\mathbb{E}_{\boldsymbol{u}_t\sim B(\mathbf{0};1)}\left[f\left(\boldsymbol{y}_t - \frac{\eta C}{\sqrt{b}}\boldsymbol{u}_t\right)\right] \\ &= \mathbb{E}_{\boldsymbol{\omega}_t}\left[\boldsymbol{y}_t\right] - \eta\nabla\hat{f}_{\frac{\eta C}{\sqrt{b}}}(\boldsymbol{y}_t), \end{aligned} \tag{8}$$

where we have used the fact that the standard normal distribution in high dimensions $d$ is close to a uniform distribution on a sphere of radius $\sqrt{d}$ (Vershynin, 2018, Section 3.3.3). This shows that $\mathbb{E}_{\boldsymbol{\omega}_t}\left[f(\boldsymbol{y}_t - \eta\boldsymbol{\omega}_t)\right]$

is a smoothed version of $f$ with a noise level $\eta C/\sqrt{b}$ and its parameter $\boldsymbol{y}_t$ can be approximately updated by using the gradient descent to minimize $\hat{f}_{\frac{\eta C}{\sqrt{b}}}$. Therefore, we can say that the degree of smoothing by the stochastic gradients in SGD is determined by the learning rate $\eta$ and batch size $b$. The findings gained from this insight are immeasurable.

### 3.3.1 Why the Use of Large Batch Sizes Leads to Solutions Falling into Sharp Local Minima

It is known that training with large batch sizes leads to a persistent degradation of model generalization performance. In particular, Keskar et al. (2017) showed experimentally that learning with large batch sizes leads to sharp local minima and worsens generalization performance. According to equation (8), using a large learning rate and/or a small batch size will make the function smoother. Thus, in using a small batch size, the sharp local minima will disappear through extensive smoothing, and SGD can reach a flat local minimum. Conversely, when using a large batch size, the smoothing is weak and the function is close to the original multimodal function, so it is easy for the solution to fall into a sharp local minimum. Thus, we have theoretical support for what Keskar et al. (2017) showed experimentally. In addition, equation (8) implies that halving the learning rate is the same as quadrupling the batch size. Note that Smith et al. (2018) argues that reducing the learning rate by half is equivalent to doubling the batch size.

**Remark:** Note that our argument is based on the somewhat non-theoretical finding that flat local solutions have better generalizability than sharp local solutions (Hochreiter & Schmidhuber, 1997; Keskar et al., 2017; Izmailov et al., 2018; Li et al., 2018). Since the function optimized by the optimizer is constructed from a limited training sample, there should be some deviation from the function constructed with unknown data, including the test data. Therefore, the intuitive explanation is that the flatness around the local solution prevents the deviation from degrading the generalizability.

### 3.3.2 Why Decaying Learning Rates and Increasing Batch Sizes are Superior to Fixed Learning Rates and Batch Sizes

From equation (8), the use of a decaying learning rate or increasing batch size during training is equivalent to decreasing the noise level of the smoothed function, so using a decaying learning rate or increasing the batch size is an implicit graduated optimization. Thus, we can say that using a decaying learning rate (Loshchilov & Hutter, 2017; Hundt et al., 2019; You et al., 2019; Lewkowycz, 2021) or increasing batch size (Byrd et al., 2012; Friedlander & Schmidt, 2012; Balles et al., 2017; De et al., 2017; Bottou et al., 2018; Smith et al., 2018) makes sense in terms of avoiding local minima and provides theoretical support for their experimental superiority.

### 3.3.3 Optimal Decay Rate of Learning Rate

As indicated in Section 3.2, gradually decreasing the noise from a value close to 1 is an optimal noise scheduling for graduated optimization. Therefore, we can say that the optimal update rule for a decaying learning rate and increasing batch size is varying slowly from a value close to 1, as in cosine annealing (without restart) (Loshchilov & Hutter, 2017), cosine power annealing (Hundt et al., 2019), and polynomial decay (Liu et al., 2015; Chen et al., 2018; Zhao et al., 2017; Chen et al., 2017). Thus, we have a theoretical explanation for why these schedules are superior. In particular, a polynomial decay with small powers from 0 to 1 satisfies the conditions that the decay rate must satisfy (see also Figures 4 and 6 in Section 3.2). Therefore, we argue that polynomial decays with powers less than equal to 1 are the optimal decaying learning rate schedule.

### 3.4 Implicit graduated optimization algorithm

Algorithm 3 embodies the framework of implicit graduated optimization for the new $\sigma$-nice function. Algorithm 4 is used to optimize each smoothed function. The $\gamma_m$ used in Algorithms 1 and 3 is a polynomial decay rate with powers from 0 to 1 which satisfies the condition that $\gamma_m$ must satisfy for the Algorithm to succeed 3 (see Proposition 3.3). The smoothed function $\hat{f}_{\delta_m}$ is $\sigma$-strongly convex in the neighborhood $N(\boldsymbol{x}^\star; d_m|\delta_m|)$. Also, the learning rate used by Algorithm 4 to optimize $\hat{f}_{\delta_m}$ is always constant. Therefore,

let us now consider the convergence of GD with a constant learning rate for a $\sigma$-strongly convex function $F = \hat{f}_{\delta_m}$. The proof of Theorem 3.3 is in Section D.3.

**Theorem 3.3** (Convergence analysis of Algorithm 4). *Suppose that $F\colon \mathbb{R}^d \to \mathbb{R}$ is a $\sigma$-strongly convex and $L_g$-smooth function and $\eta < \min\left\{\frac{1}{\sigma}, \frac{2}{L_g}\right\}$. Then, the sequence $(\hat{\boldsymbol{x}}_t)_{t\in\mathbb{N}}$ generated by Algorithm 4 satisfies*

$$\min_{t\in[T]}\left(F\left(\hat{\boldsymbol{x}}_t\right) - F(\boldsymbol{x}^\star)\right) \leq \frac{H_3}{T} = \mathcal{O}\left(\frac{1}{T}\right), \tag{9}$$

*where $\boldsymbol{x}^\star$ is the global minimizer of $F$, and $H_3$ is a nonnegative constant.*

Theorem 3.3 is the convergence analysis of Algorithm 4 for any $\sigma$-strongly convex and $L_g$-smooth function $F$. It shows that Algorithm 4 can reach an $\epsilon_m$-neighborhood of the optimal solution $\boldsymbol{x}^\star_{\delta_m}$ of $\hat{f}_{\delta_m}$ in approximately $T_F := H_3/\epsilon_m$ iterations. Proposition 3.3 also holds for Algorithm 3. Therefore, if the initial point $\bar{\boldsymbol{x}}_1$ is contained in the $\sigma$-strongly convex region $N(\boldsymbol{x}^\star; d_1|\delta_1|)$ of the smoothest function $\hat{f}_{\delta_1}$, then the algorithm will always reach the globally optimal solution $\boldsymbol{x}^\star$ of the original function $f$.

**Remark:** Algorithms 3 and 4 represent implicit graduated optimization algorithms. Function smoothing is accomplished implicitly by the stochastic noise in SGD. From Section 3.3, SGD is running for the objective function $f$, but behind the scenes, GD can be regarded as running for the function $\hat{f}_{\frac{\eta C}{\sqrt{b}}}$, which is smoothed version of $f$, where $\eta$ and $b$ are hyperparameters of SGD. That is why, our Algorithm 4 can be GD. The convergence analysis for this case is Theorem 3.3. On the other hand, another way of looking at it is possible. The experiments in Section 4 simply run SGD, which uses a decaying learning rate or increasing batch size, and GD is not used explicitly. In this case, since $b$ data are handled in each step, it may be viewed as $\hat{f}_{\frac{\eta C}{\sqrt{b}}} \approx \frac{1}{b}\sum_{i=1}^{b} f_{\xi_i}$. Then Algorithm 4 can be SGD since $\hat{f}_{\frac{\eta C}{\sqrt{b}}}$ varies depending on the data chosen. If the projection is computable to ensure that the SGD sequence does not leave the strongly convex region of the function $\hat{f}_{\frac{\eta C}{\sqrt{b}}}$, then convergence can be guaranteed as with Remark for Theorem 3.1. The relationship between the loss function $f_i$ for the $i$-th data and the smoothed function $\hat{f}_{\frac{\eta C}{\sqrt{b}}}$ is still unknown. Then, there may be some differences between theory and practice.

---

**Algorithm 3** Implicit Graduated Optimization with SGD

**Require:** $\epsilon > 0, p \in (0,1], \bar{d} > 0, \boldsymbol{x}_1, \eta_1, b_1$

$\quad \delta_1 := \frac{\eta_1 C}{\sqrt{b_1}}$

$\quad \alpha_0 := \min\left\{\frac{\sqrt{b_1}}{4L_f\eta_1 C(1+\bar{d})}, \frac{\sqrt{b_1}}{\sqrt{2\sigma}\eta_1 C}\right\}, M^p := \frac{1}{\alpha_0 \epsilon}$

$\quad$ **for** $m = 1$ to $M + 1$ **do**

$\quad\quad$ **if** $m \neq M + 1$ **then**

$\quad\quad\quad \epsilon_m := \sigma^2\delta_m^2, \ T_F := H_3/\epsilon_m$

$\quad\quad\quad \gamma_m := \frac{(M-m)^p}{\{M-(m-1)\}^p}$

$\quad\quad\quad \kappa_m/\sqrt{\lambda_m} = \gamma_m \ (\kappa_m \in (0,1], \lambda_m \geq 1)$

$\quad\quad$ **end if**

$\quad\quad \boldsymbol{x}_{m+1} := \mathrm{GD}(T_F, \boldsymbol{x}_m, \hat{f}_{\delta_m}, \eta_m)$

$\quad\quad \eta_{m+1} := \kappa_m \eta_m, b_{m+1} := \lambda_m b_m$

$\quad\quad \delta_{m+1} := \frac{\eta_{m+1}C}{\sqrt{b_{m+1}}}$

$\quad$ **end for**

$\quad$ **return** $\boldsymbol{x}_{M+2}$

---

The next theorem guarantees the convergence of Algorithm 3 with the new $\sigma$-nice function (The proof of Theorem 3.4 is in Section D.4).

**Theorem 3.4** (Convergence analysis of Algorithm 3). *Let $\epsilon \in (0,1)$ and $f$ be an $L_f$-Lipschitz new $\sigma$-nice function. Suppose that we run Algorithm 3; then after $\mathcal{O}\left(1/\epsilon^{\frac{1}{p}}\right)$ rounds, the algorithm reaches an $\epsilon$-neighborhood of the global optimal solution $\boldsymbol{x}^\star$.*

---

**Algorithm 4** GD with a constant learning rate

---

**Require:** $T_F, \hat{\boldsymbol{x}}_1, F, \eta$
   **for** $t = 1$ to $T_F$ **do**
      $\hat{\boldsymbol{x}}_{t+1} := \hat{\boldsymbol{x}}_t - \eta \nabla F(\boldsymbol{x}_t)$
   **end for**
   **return** $\hat{\boldsymbol{x}}_{T_F+1} = \text{GD}(T_F, \hat{\boldsymbol{x}}_1, F, \eta)$

---

Note that Theorem 3.4 provides a total complexity including those of Algorithm 3 and Algorithm 4, because Algorithm 3 uses Algorithm 4 at each $m \in [M]$. Theorem 3.4 implies that convergence is faster when the power of the polynomial decay $p$ is high, and when $p = 1$, it takes at least $\mathcal{O}(1/\epsilon)$ rounds for new $\sigma$-nice functions.

## 4  Numerical Results

The experimental environment was as follows: NVIDIA GeForce RTX 4090×2GPU and Intel Core i9 13900KF CPU. The software environment was Python 3.10.12, PyTorch 2.1.0 and CUDA 12.2. The code is available at `https://anonymous.4open.science/r/new-sigma-nice`.

### 4.1  Implicit Graduated Optimization of DNN

We compared four types of SGD for image classification:

1. constant learning rate and constant batch size,

2. decaying learning rate and constant batch size,

3. constant learning rate and increasing batch size,

4. decaying learning rate and increasing batch size.

We evaluated the performance of the four SGDs in training ResNet18 (He et al., 2016) on the CIFAR100 dataset (Krizhevsky, 2009) (Figure 7), WideResNet-28-10 (Zagoruyko & Komodakis, 2016) on the CIFAR100 dataset (Figure 8), and ResNet34 (He et al., 2016) on the ImageNet dataset (Deng et al., 2009) (Figure 9). All experiments were run for 200 epochs. In methods 2, 3, and 4, the noise decreased every 40 epochs, with a common decay rate of $1/\sqrt{2}$. That is, every 40 epochs, the learning rate of method 2 was multiplied by $1/\sqrt{2}$, the batch size of method 3 was doubled, and the learning rate and batch size of method 4 were respectively multiplied by $\sqrt{3}/2$ and 1.5. The initial learning rate was 0.1 for all methods, which was determined by performing a grid search among $[0.01, 0.1, 1.0, 10]$. The noise reduction interval was every 40 epochs, which was determined by performing a grid search among $[10, 20, 25, 40, 50, 100]$. A history of the learning rate or batch size for each method is provided in the caption of each figure.

For methods 2, 3, and 4, the decay rates are all $1/\sqrt{2}$, and the decay intervals are all 40 epochs, so throughout the training, the three methods should theoretically be optimizing the exact same five smoothed functions in sequence. Nevertheless, the local solutions reached by each of the three methods are not exactly the same. All results indicate that method 3 is superior to method 2 and that method 4 is superior to method 3 in both test accuracy and training loss function values. This difference can be attributed to the different learning rates used to optimize each smoothing function. Among methods 2, 3, and 4, method 3, which does not decay the learning rate, maintains the highest learning rate 0.1, followed by method 4 and method 2. In all graphs, the loss function values are always small in this order; i.e., the larger the learning rate is, the lower loss function values become. Therefore, we can say that the noise level $|\delta|$, expressed as $\frac{\eta C}{\sqrt{b}}$, needs to be reduced, while the learning rate $\eta$ needs to remain as large as possible. Alternatively, if the learning rate is small, then a large number of iterations are required. Thus, for the same rate of change and the same number of epochs, an increasing batch size is superior to a decreasing learning rate because it can maintain a large learning rate and can be made to iterate a lot when the batch size is small.

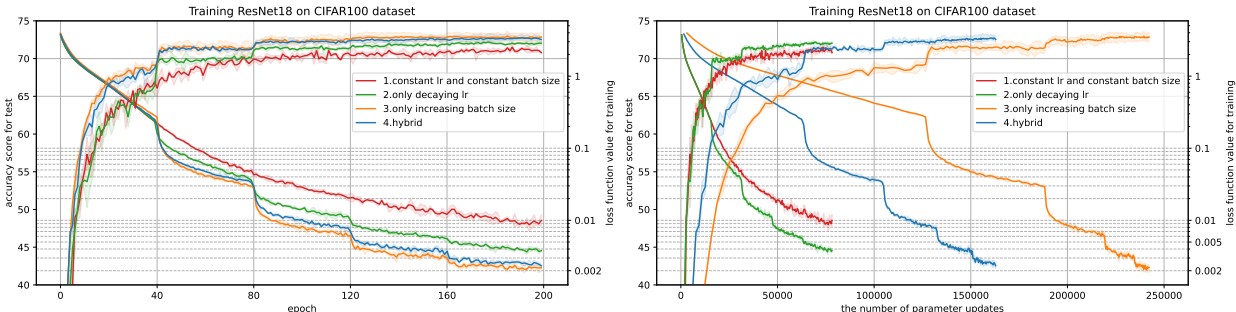

Figure 7: Accuracy score for testing and loss function value for training versus the number of epochs (**left**) and the number of parameter updates (**right**) in training ResNet18 on the CIFAR100 dataset. The solid line represents the mean value, and the shaded area represents the maximum and minimum over three runs. In method 1, the learning rate and the batch size were fixed at 0.1 and 128, respectively. In method 2, the learning rate decreased every 40 epochs as $\left[0.1, \frac{1}{10\sqrt{2}}, 0.05, \frac{1}{20\sqrt{2}}, 0.025\right]$ and the batch size was fixed at 128. In method 3, the learning rate was fixed at 0.1, and the batch size was increased as $[16, 32, 64, 128, 256]$. In method 4, the learning rate was decreased as $\left[0.1, \frac{\sqrt{3}}{20}, 0.075, \frac{3\sqrt{3}}{80}, 0.05625\right]$ and the batch size was increased as $[32, 48, 72, 108, 162]$.

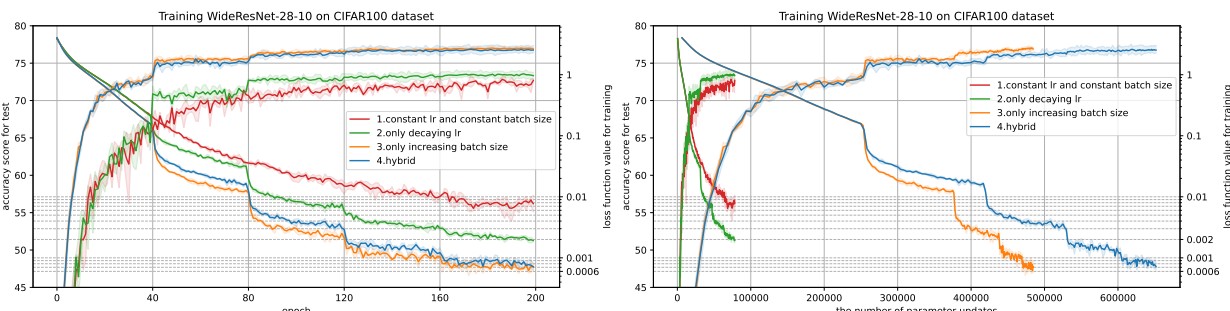

Figure 8: Accuracy score for testing and loss function value for training versus the number of epochs (**left**) and the number of parameter updates (**right**) in training WideResNet-28-10 on the CIFAR100 dataset. The solid line represents the mean value, and the shaded area represents the maximum and minimum over three runs. In method 1, the learning rate and batch size were fixed at 0.1 and 128, respectively. In method 2, the learning rate was decreased every 40 epochs as $\left[0.1, \frac{1}{10\sqrt{2}}, 0.05, \frac{1}{20\sqrt{2}}, 0.025\right]$ and the batch size was fixed at 128. In method 3, the learning rate was fixed at 0.1, and the batch size increased as $[8, 16, 32, 64, 128]$. In method 4, the learning rate decreased as $\left[0.1, \frac{\sqrt{3}}{20}, 0.075, \frac{3\sqrt{3}}{80}, 0.05625\right]$ and the batch size increased as $[8, 12, 18, 27, 40]$.

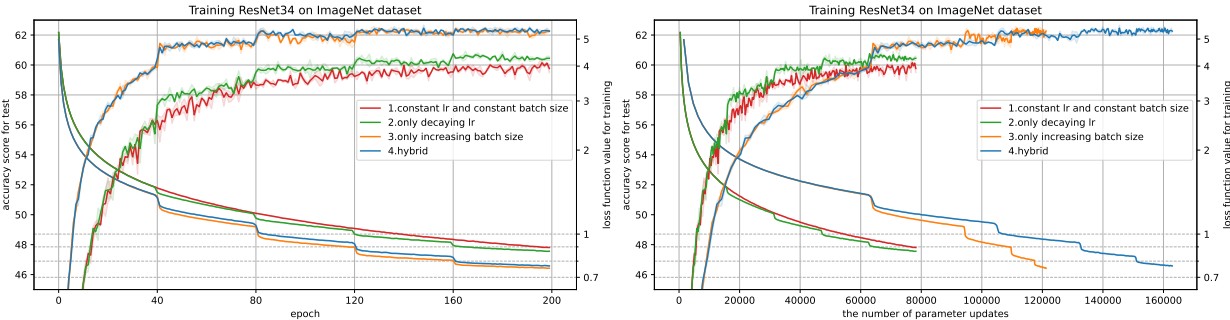

Figure 9: Accuracy score for testing and loss function value for training versus the number of epochs (**left**) and the number of parameter updates (**right**) in training ResNet34 on the ImageNet dataset. The solid line represents the mean value, and the shaded area represents the maximum and minimum over three runs. In method 1, the learning rate and batch size were fixed at 0.1 and 256, respectively. In method 2, the learning rate was decreased every 40 epochs as $\left[0.1, \frac{1}{10\sqrt{2}}, 0.05, \frac{1}{20\sqrt{2}}, 0.025\right]$ and the batch size was fixed at 256. In method 3, the learning rate was fixed at 0.1, and the batch size was increased as $[32, 64, 128, 256, 512]$. In method 4, the learning rate was decreased as $\left[0.1, \frac{\sqrt{3}}{20}, 0.075, \frac{3\sqrt{3}}{80}, 0.05625\right]$ and the batch size was increased as $[32, 48, 72, 108, 162]$.

Theoretically, the noise level $|\delta_m|$ should gradually decrease and become zero at the end, so in our algorithm 3, the learning rate $\eta_m$ should be zero at the end or the batch size $b_m$ should match the number of data sets at the end. However, if the learning rate is 0, training cannot proceed, and if the batch size is close to a full batch, it is not feasible from a computational point of view. For this reason, the experiments described in this paper are not fully graduated optimizations; i.e., full global optimization is not achieved. In fact, the last batch size used by method 2 is around 128 to 512, which is far from a full batch. Therefore, the solution reached in this experiment is the optimal solution for a function that has been smoothed to some extent, and to achieve a global optimization of the DNN, it is necessary to increase only the batch size to eventually reach a full batch, or increase the number of iterations accordingly while increasing the batch size and decaying the learning rate.

Finally, we should note that graduated optimization with Algorithm 1 is not applicable to DNN. Our approach, Algorithm 3, allows implicit graduated optimization by exploiting the smoothness of SGD; the experimental results provided in this section imply its success.

## 4.2 Experiments on Optimal Noise Scheduling

Section 3.3.3 shows that the optimal decaying learning rate is in theory a polynomial decay with small powers from 0 to 1. To demonstrate this, we evaluated the performance of SGDs with several decaying learning rate schedules in training ResNet18 and WideResNet-28-10 on CIFAR100 dataset. Figures 10 and 11 plot the accuracy in testing and the loss function value in training versus number of epochs. All experiments were run for 200 epochs and the batch size was fixed at 128. The learning rate was decreased per epoch; see Section 3.2 for the respective update rules.

Both results show that a polynomial decay with a power less than or equal to 1, which is the schedule that satisfies the condition that $\gamma_m$ must satisfy, is superior in both test accuracy and training loss function value. Furthermore, the loss function values and test accuracy worsen the further away from the green region that the decay rate curve must satisfy (see Figure 4 and Figure 6), and the order is in excellent agreement with the order in which lower loss function values are achieved. According to Theorem 3.4, Algorithm 3 reaches an $\epsilon$-neighborhood of the globally optimal solution after $\mathcal{O}\left(1/\epsilon^{\frac{1}{p}}\right)$ iterations. Thus, theoretically, the closer $p$ is to 1, the fewer iterations are required. This explains why $p = 0.1$ is not initially superior in both test accuracy and loss function value for training in Figures 10 and 11.

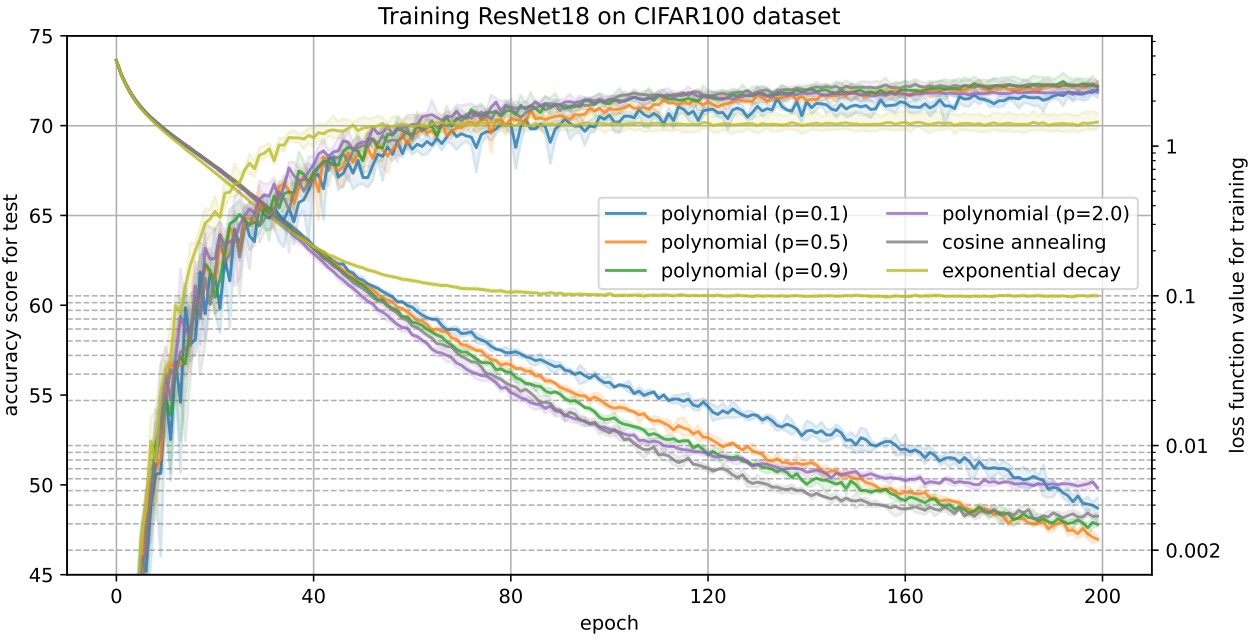

Figure 10: Accuracy score for testing and loss function value for training versus epochs in training of ResNet18 on the CIFAR100 dataset. The solid line represents the mean value, and the shaded area represents the maximum and minimum over three runs. See Figure 15 in Appendix F for full results.

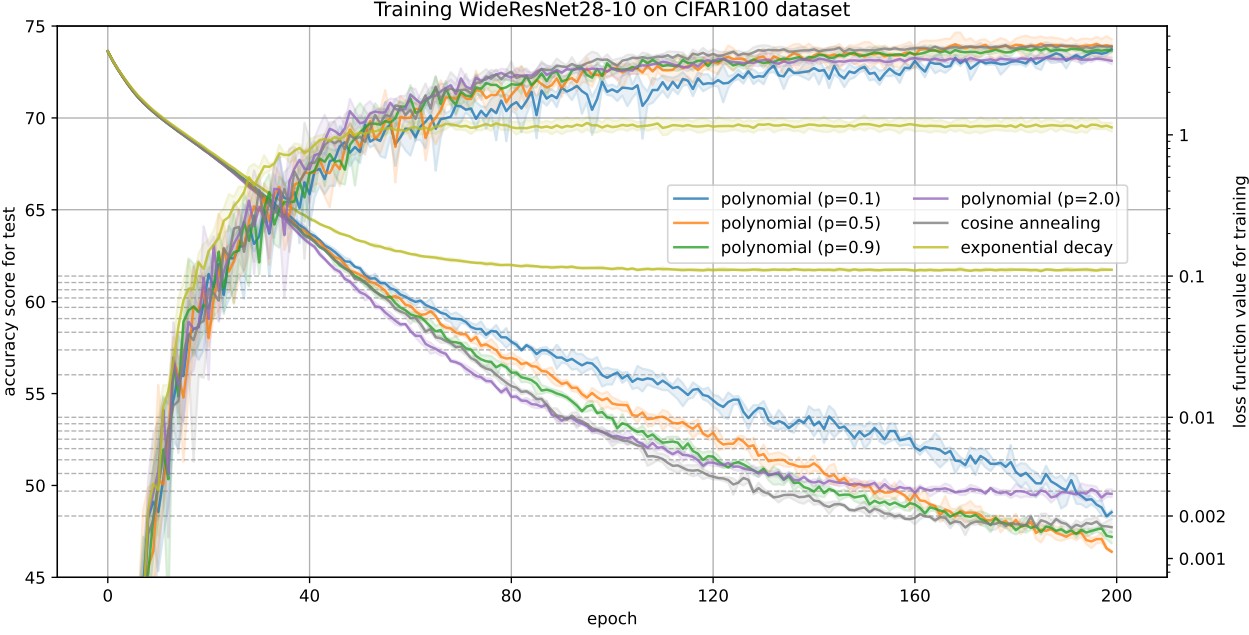

Figure 11: Accuracy score for testing and loss function value for training versus epochs in training of WideResNet-28-10 on the CIFAR100 dataset. The solid line represents the mean value, and the shaded area represents the maximum and minimum over three runs. See Figure 16 in Appendix F for full results.

## 5  Conclusion

We defined a family of nonconvex functions: new $\sigma$-nice functions that prove that the graduated optimization approach converges to a globally optimal solution. We also provided sufficient conditions for any nonconvex function to be a new $\sigma$-nice function and performed a convergence analysis of the graduated optimization algorithm for the new $\sigma$-nice functions. We proved that SGD with a mini-batch stochastic gradient has the effect of smoothing the function, and the degree of smoothing is greater with larger learning rates and smaller batch sizes. This shows theoretically that smoothing with large batch sizes is makes it easy to fall into sharp local minima, that using a decaying learning rate and/or increasing batch size is implicitly graduated optimization, which makes sense in the sense that it avoids local solutions, and that the optimal learning rate scheduling rule is a gradual scheduling with a decreasing rate, such as a polynomial decay with small powers. Based on these findings, we proposed a new graduated optimization algorithm that uses a decaying learning rate and increasing batch size and analyzed it. Finally, we conducted experiments whose results showed the superiority of our recommended framework for image classification tasks on CIFAR100 and ImageNet and that polynomial decay with small powers is an optimal decaying learning rate schedule.

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

# A   Derivation of equation (1)

Let $\boldsymbol{y}_t$ be the parameter updated by gradient descent (GD) and $\boldsymbol{x}_{t+1}$ be the parameter updated by SGD at time $t$, i.e.,

$$\boldsymbol{y}_t := \boldsymbol{x}_t - \eta \nabla f(\boldsymbol{x}_t),$$
$$\boldsymbol{x}_{t+1} := \boldsymbol{x}_t - \eta \nabla f_{\mathcal{S}_t}(\boldsymbol{x}_t)$$
$$= \boldsymbol{x}_t - \eta (\nabla f(\boldsymbol{x}_t) + \boldsymbol{\omega}_t).$$

Then, we have

$$
\begin{aligned}
\boldsymbol{x}_{t+1} &:= \boldsymbol{x}_t - \eta \nabla f_{\mathcal{S}_t}(\boldsymbol{x}_t) \\
&= (\boldsymbol{y}_t + \eta \nabla f(\boldsymbol{x}_t)) - \eta \nabla f_{\mathcal{S}_t}(\boldsymbol{x}_t) \\
&= \boldsymbol{y}_t - \eta \boldsymbol{\omega}_t,
\end{aligned}
\tag{10}
$$

from $\boldsymbol{\omega}_t := \nabla f_{\mathcal{S}_t}(\boldsymbol{x}_t) - \nabla f(\boldsymbol{x}_t)$. Hence,

$$
\begin{aligned}
\boldsymbol{y}_{t+1} &= \boldsymbol{x}_{t+1} - \eta \nabla f(\boldsymbol{x}_{t+1}) \\
&= \boldsymbol{y}_t - \eta \boldsymbol{\omega}_t - \eta \nabla f(\boldsymbol{y}_t - \eta \boldsymbol{\omega}_t).
\end{aligned}
$$

By taking the expectation with respect to $\boldsymbol{\omega}_t$ on both sides, we have, from $\mathbb{E}_{\boldsymbol{\omega}_t}[\boldsymbol{\omega}_t] = \boldsymbol{0}$,

$$\mathbb{E}_{\boldsymbol{\omega}_t}[\boldsymbol{y}_{t+1}] = \mathbb{E}_{\boldsymbol{\omega}_t}[\boldsymbol{y}_t] - \eta \nabla \mathbb{E}_{\boldsymbol{\omega}_t}[f(\boldsymbol{y}_t - \eta \boldsymbol{\omega}_t)],$$

where we have used $\mathbb{E}_{\boldsymbol{\omega}_t}[\nabla f(\boldsymbol{y}_t - \eta \boldsymbol{\omega}_t)] = \nabla \mathbb{E}_{\boldsymbol{\omega}_t}[f(\boldsymbol{y}_t - \eta \boldsymbol{\omega}_t)]$, which holds for the Lipschitz-continuous and differentiable of $f$ (Shapiro et al., 2009, Theorem 7.49). In addition, from (10) and $\mathbb{E}_{\boldsymbol{\omega}_t}[\boldsymbol{\omega}_t] = \boldsymbol{0}$, we obtain

$$\mathbb{E}_{\boldsymbol{\omega}_t}[\boldsymbol{x}_{t+1}] = \boldsymbol{y}_t.$$

Therefore, on average, the parameter $\boldsymbol{x}_{t+1}$ of the function $f$ arrived at by SGD coincides with the parameter $\boldsymbol{y}_t$ of the smoothed function $\hat{f}(\boldsymbol{y}_t) := \mathbb{E}_{\boldsymbol{\omega}_t}[f(\boldsymbol{y}_t - \eta \boldsymbol{\omega}_t)]$ arrived at by GD.

# B   Proofs of the Lemmas in Section 2.2

## B.1   Proof of Lemma 2.1

*Proof.* (A3)(ii) and (A4) guarantee that

$$
\begin{aligned}
\mathbb{E}_{\xi_t}\left[\|\nabla f_{\mathcal{S}_t}(\boldsymbol{x}_t) - \nabla f(\boldsymbol{x}_t)\|^2\right] &= \mathbb{E}_{\xi_t}\left[\left\|\frac{1}{b}\sum_{i=1}^{b} \mathsf{G}_{\xi_{t,i}}(\boldsymbol{x}_t) - \nabla f(\boldsymbol{x}_t)\right\|^2\right] \\
&= \mathbb{E}_{\xi_t}\left[\left\|\frac{1}{b}\sum_{i=1}^{b} \mathsf{G}_{\xi_{t,i}}(\boldsymbol{x}_t) - \frac{1}{b}\sum_{i=1}^{b}\nabla f(\boldsymbol{x}_t)\right\|^2\right] \\
&= \mathbb{E}_{\xi_t}\left[\left\|\frac{1}{b}\sum_{i=1}^{b} \left(\mathsf{G}_{\xi_{t,i}}(\boldsymbol{x}_t) - \nabla f(\boldsymbol{x}_t)\right)\right\|^2\right] \\
&= \frac{1}{b^2}\mathbb{E}_{\xi_t}\left[\left\|\sum_{i=1}^{b} \left(\mathsf{G}_{\xi_{t,i}}(\boldsymbol{x}_t) - \nabla f(\boldsymbol{x}_t)\right)\right\|^2\right] \\
&= \frac{1}{b^2}\mathbb{E}_{\xi_t}\left[\sum_{i=1}^{b} \left\|\mathsf{G}_{\xi_{t,i}}(\boldsymbol{x}_t) - \nabla f(\boldsymbol{x}_t)\right\|^2\right] \\
&\leq \frac{C^2}{b}.
\end{aligned}
$$

This completes the proof. □

## B.2 Proof of Lemma 2.2

*Proof.* From Definition 2.1 and (A1), we have, for all $\boldsymbol{x}, \boldsymbol{y} \in \mathbb{R}^d$,

$$
\begin{aligned}
\left\| \nabla \hat{f}_\delta(\boldsymbol{x}) - \nabla \hat{f}_\delta(\boldsymbol{y}) \right\| &= \left\| \nabla \mathbb{E}_{\boldsymbol{u}} \left[ f(\boldsymbol{x} - \delta \boldsymbol{u}) \right] - \nabla \mathbb{E}_{\boldsymbol{u}} \left[ f(\boldsymbol{y} - \delta \boldsymbol{u}) \right] \right\| \\
&= \left\| \mathbb{E}_{\boldsymbol{u}} \left[ \nabla f(\boldsymbol{x} - \delta \boldsymbol{u}) \right] - \mathbb{E}_{\boldsymbol{u}} \left[ \nabla f(\boldsymbol{y} - \delta \boldsymbol{u}) \right] \right\| \\
&= \left\| \mathbb{E}_{\boldsymbol{u}} \left[ \nabla f(\boldsymbol{x} - \delta \boldsymbol{u}) - \nabla f(\boldsymbol{y} - \delta \boldsymbol{u}) \right] \right\| \\
&\leq \mathbb{E}_{\boldsymbol{u}} \left[ \left\| \nabla f(\boldsymbol{x} - \delta \boldsymbol{u}) - \nabla f(\boldsymbol{y} - \delta \boldsymbol{u}) \right\| \right] \\
&\leq \mathbb{E}_{\boldsymbol{u}} \left[ L_g \left\| (\boldsymbol{x} - \delta \boldsymbol{u}) - (\boldsymbol{y} - \delta \boldsymbol{u}) \right\| \right] \\
&= \mathbb{E}_{\boldsymbol{u}} \left[ L_g \left\| \boldsymbol{x} - \boldsymbol{y} \right\| \right] \\
&= L_g \| \boldsymbol{x} - \boldsymbol{y} \|.
\end{aligned}
$$

This completes the proof. □

## B.3 Proof of Lemma 2.3

*Proof.* From Definition 2.1 and (A2), we have, for all $\boldsymbol{x}, \boldsymbol{y} \in \mathbb{R}^d$,

$$
\begin{aligned}
\left| \hat{f}_\delta(\boldsymbol{x}) - \hat{f}_\delta(\boldsymbol{y}) \right| &= \left| \mathbb{E}_{\boldsymbol{u}} \left[ f(\boldsymbol{x} - \delta \boldsymbol{u}) \right] - \mathbb{E}_{\boldsymbol{u}} \left[ f(\boldsymbol{y} - \delta \boldsymbol{u}) \right] \right| \\
&= \left| \mathbb{E}_{\boldsymbol{u}} \left[ f(\boldsymbol{x} - \delta \boldsymbol{u}) - f(\boldsymbol{y} - \delta \boldsymbol{u}) \right] \right| \\
&\leq \mathbb{E}_{\boldsymbol{u}} \left[ \left| f(\boldsymbol{x} - \delta \boldsymbol{u}) - f(\boldsymbol{y} - \delta \boldsymbol{u}) \right| \right] \\
&\leq \mathbb{E}_{\boldsymbol{u}} \left[ L_f \| (\boldsymbol{x} - \delta \boldsymbol{u}) - (\boldsymbol{y} - \delta \boldsymbol{u}) \| \right] \\
&= \mathbb{E}_{\boldsymbol{u}} \left[ L_f \left\| \boldsymbol{x} - \boldsymbol{y} \right\| \right] \\
&= L_f \| \boldsymbol{x} - \boldsymbol{y} \|.
\end{aligned}
$$

This completes the proof. □

## B.4 Proof of Lemma 2.4

*Proof.* From Definition 2.1 and (A2), we have, for all $\boldsymbol{x}, \boldsymbol{y} \in \mathbb{R}^d$,

$$
\begin{aligned}
\left| \hat{f}_\delta(\boldsymbol{x}) - f(\boldsymbol{x}) \right| &= \left| \mathbb{E}_{\boldsymbol{u}} \left[ f(\boldsymbol{x} - \delta \boldsymbol{u}) \right] - f(\boldsymbol{x}) \right| \\
&= \left| \mathbb{E}_{\boldsymbol{u}} \left[ f(\boldsymbol{x} - \delta \boldsymbol{u}) - f(\boldsymbol{x}) \right] \right| \\
&\leq \mathbb{E}_{\boldsymbol{u}} \left[ \left| f(\boldsymbol{x} - \delta \boldsymbol{u}) - f(\boldsymbol{x}) \right| \right] \\
&\leq \mathbb{E}_{\boldsymbol{u}} \left[ L_f \| (\boldsymbol{x} - \delta \boldsymbol{u}) - \boldsymbol{x} \| \right] \\
&= \mathbb{E}_{\boldsymbol{u}} \left[ L_f |\delta| \| \boldsymbol{u} \| \right] \\
&= |\delta| L_f,
\end{aligned}
$$

where we have used $\| \boldsymbol{u} \| \leq 1$. This completes the proof. □

## C  Lemmas used in the proofs of the theorems

**Lemma C.1.** *Suppose that $F \colon \mathbb{R}^d \to \mathbb{R}$ is $\sigma$-strongly convex and $\hat{\boldsymbol{x}}_{t+1} := \hat{\boldsymbol{x}}_t - \eta_t \boldsymbol{g}_t$. Then, for all $t \in \mathbb{N}$,*

$$
F(\hat{\boldsymbol{x}}_t) - F(\boldsymbol{x}^\star) \leq \frac{1 - \sigma \eta_t}{2 \eta_t} X_t - \frac{1}{2 \eta_t} X_{t+1} + \frac{\eta_t}{2} \| \boldsymbol{g}_t \|^2,
$$

*where $\boldsymbol{g}_t := \nabla F(\hat{\boldsymbol{x}}_t)$, $X_t := \| \hat{\boldsymbol{x}}_t - \boldsymbol{x}^\star \|^2$, and $\boldsymbol{x}^\star$ is the global minimizer of $F$.*

*Proof.* Let $t \in \mathbb{N}$. The definition of $\hat{\boldsymbol{x}}_{t+1}$ guarantees that

$$
\begin{aligned}
\|\hat{\boldsymbol{x}}_{t+1} - \boldsymbol{x}^\star\|^2 &= \|(\hat{\boldsymbol{x}}_t - \eta_t \boldsymbol{g}_t) - \boldsymbol{x}^\star\|^2 \\
&= \|\hat{\boldsymbol{x}}_t - \boldsymbol{x}^\star\|^2 - 2\eta_t \langle \hat{\boldsymbol{x}}_t - \boldsymbol{x}^\star, \boldsymbol{g}_t \rangle + \eta_t^2 \|\boldsymbol{g}_t\|^2.
\end{aligned}
$$

From the $\sigma$-strong convexity of $F$,

$$
\|\hat{\boldsymbol{x}}_{t+1} - \boldsymbol{x}^\star\|^2 \leq \|\hat{\boldsymbol{x}}_t - \boldsymbol{x}^\star\|^2 + 2\eta_t \left( F(\boldsymbol{x}^\star) - F(\hat{\boldsymbol{x}}_t) - \frac{\sigma}{2}\|\hat{\boldsymbol{x}}_t - \boldsymbol{x}^\star\|^2 \right) + \eta_t^2 \|\boldsymbol{g}_t\|^2.
$$

Hence,

$$
F(\hat{\boldsymbol{x}}_t) - F(\boldsymbol{x}^\star) \leq \frac{1 - \sigma \eta_t}{2\eta_t} \|\hat{\boldsymbol{x}}_t - \boldsymbol{x}^\star\|^2 - \frac{1}{2\eta_t} \|\hat{\boldsymbol{x}}_{t+1} - \boldsymbol{x}^\star\|^2 + \frac{\eta_t}{2}\|\boldsymbol{g}_t\|^2.
$$

This completes the proof. $\qquad\square$

**Lemma C.2.** *Suppose that $F \colon \mathbb{R}^d \to \mathbb{R}$ is $L_g$-smooth and $\hat{\boldsymbol{x}}_{t+1} := \hat{\boldsymbol{x}}_t - \eta_t \boldsymbol{g}_t$. Then, for all $t \in \mathbb{N}$,*

$$
\eta_t \left( 1 - \frac{L_g \eta_t}{2} \right) \|\nabla F(\hat{\boldsymbol{x}}_t)\|^2 \leq F(\hat{\boldsymbol{x}}_t) - F(\hat{\boldsymbol{x}}_{t+1}).
$$

*where $\boldsymbol{g}_t := \nabla F(\hat{\boldsymbol{x}}_t)$ and $\boldsymbol{x}^\star$ is the global minimizer of $F$.*

*Proof.* From the $L_g$-smoothness of the $F$ and the definition of $\hat{\boldsymbol{x}}_{t+1}$, we have, for all $t \in \mathbb{N}$,

$$
\begin{aligned}
F(\hat{\boldsymbol{x}}_{t+1}) &\leq F(\hat{\boldsymbol{x}}_t) + \langle \nabla F(\hat{\boldsymbol{x}}_t), \hat{\boldsymbol{x}}_{t+1} - \hat{\boldsymbol{x}}_t \rangle + \frac{L_g}{2}\|\hat{\boldsymbol{x}}_{t+1} - \hat{\boldsymbol{x}}_t\|^2 \\
&= F(\hat{\boldsymbol{x}}_t) - \eta_t \langle \nabla F(\hat{\boldsymbol{x}}_t), \boldsymbol{g}_t \rangle + \frac{L_g \eta_t^2}{2}\|\boldsymbol{g}_t\|^2 \\
&\leq F(\hat{\boldsymbol{x}}_t) - \eta_t \left( 1 - \frac{L_g \eta_t}{2} \right) \|\nabla F(\hat{\boldsymbol{x}}_t)\|^2.
\end{aligned}
$$

Therefore, we have

$$
\eta_t \left( 1 - \frac{L_g \eta_t}{2} \right) \|\nabla F(\hat{\boldsymbol{x}}_t)\|^2 \leq F(\hat{\boldsymbol{x}}_t) - F(\hat{\boldsymbol{x}}_{t+1}).
$$

This completes the proof. $\qquad\square$

**Lemma C.3.** *Suppose that $F \colon \mathbb{R}^d \to \mathbb{R}$ is $L_g$-smooth, $\hat{\boldsymbol{x}}_{t+1} := \hat{\boldsymbol{x}}_t - \eta_t \boldsymbol{g}_t$, and $\eta_t := \eta < \frac{2}{L_g}$. Then, for all $t \in \mathbb{N}$,*

$$
\frac{1}{T} \sum_{t=1}^{T} \|\boldsymbol{g}_t\|^2 \leq \frac{2\left( F(\hat{\boldsymbol{x}}_1) - F(\boldsymbol{x}^\star) \right)}{\eta\left( 2 - L_g \eta \right) T},
$$

*where $\boldsymbol{g}_t := \nabla F(\hat{\boldsymbol{x}}_t)$ and $\boldsymbol{x}^\star$ is the global minimizer of $F$.*

*Proof.* According to Lemma C.2, we have

$$
\eta \left( 1 - \frac{L_g \eta}{2} \right) \|\nabla F(\boldsymbol{x}_t)\|^2 \leq F(\hat{\boldsymbol{x}}_t) - F(\hat{\boldsymbol{x}}_{t+1}).
$$

Summing over $t$, we find that

$$
\eta \left( 1 - \frac{L_g \eta}{2} \right) \frac{1}{T} \sum_{t=1}^{T} \|\nabla F(\hat{\boldsymbol{x}}_t)\|^2 \leq \frac{F(\hat{\boldsymbol{x}}_1) - F(\hat{\boldsymbol{x}}_{T+1})}{T}.
$$

Hence, from $\eta < \frac{2}{L_g}$,

$$\frac{1}{T} \sum_{t=1}^{T} \|\boldsymbol{g}_t\|^2 = \frac{2\left(F(\hat{\boldsymbol{x}}_1) - F(\boldsymbol{x}^\star)\right)}{\eta\left(2 - L_g\eta\right)T}.$$

This completes the proof. □

**Lemma C.4.** *Suppose that $F\colon \mathbb{R}^d \to \mathbb{R}$ is $L_g$-smooth, $\hat{\boldsymbol{x}}_{t+1} := \hat{\boldsymbol{x}}_t - \eta_t \boldsymbol{g}_t$, and $\eta_t := \frac{2}{\sigma t}$. Then, for all $t \in \mathbb{N}$,*

$$\|\boldsymbol{g}_t\|^2 \leq B_2,$$

*where $\boldsymbol{g}_t := \nabla F(\hat{\boldsymbol{x}}_t)$ and $B_2 \geq 0$ is a nonnegative constant.*

*Proof.* According to Lemma C.2, we have

$$\eta_t \left(1 - \frac{L_g \eta_t}{2}\right) \|\nabla F(\hat{\boldsymbol{x}}_t)\|^2 \leq F(\hat{\boldsymbol{x}}_t) - F(\hat{\boldsymbol{x}}_{t+1}).$$

Summing over $t$ from $t = t_0$ to $t = T$, we have

$$\sum_{t=t_0}^{T} \eta_t \left(1 - \frac{L_g \eta_t}{2}\right) \|\nabla F(\hat{\boldsymbol{x}}_t)\|^2 \leq F(\hat{\boldsymbol{x}}_{t_0}) - F(\hat{\boldsymbol{x}}_T),$$

where $t_0$ satisfies

$$\forall t \geq t_0 \colon \eta_{t_0} < \frac{2}{L_g}.$$

Hence, we obtain

$$\left(1 - \frac{L_g \eta_{t_0}}{2}\right) \sum_{t=t_0}^{T} \eta_t \|\nabla F(\hat{\boldsymbol{x}}_t)\|^2 \leq \underbrace{F(\hat{\boldsymbol{x}}_{t_0}) - F(\boldsymbol{x}^\star)}_{=:B} < \infty.$$

Then,

$$\sum_{t=t_0}^{T} \eta_t \|\nabla F(\hat{\boldsymbol{x}}_t)\|^2 \leq \frac{2B}{2 - L_g \eta_{t_0}} < \infty.$$

Therefore,

$$\sum_{t=1}^{T} \eta_t \|\nabla F(\hat{\boldsymbol{x}}_t)\|^2 \leq \underbrace{\frac{2B}{2 - L_g \eta_{t_0}} + \sum_{t=1}^{t_0-1} \eta_t \|\nabla F(\hat{\boldsymbol{x}}_t)\|^2}_{=:\hat{B}} < \infty. \tag{11}$$

From $\eta_T \leq \eta_t := \frac{2}{\sigma t}$,

$$\frac{2}{\sigma T} \sum_{t=1}^{T} \|\nabla F(\hat{\boldsymbol{x}}_t)\|^2 = \eta_T \sum_{t=1}^{T} \|\nabla F(\hat{\boldsymbol{x}}_t)\|^2 \leq \sum_{t=1}^{T} \eta_t \|\nabla F(\hat{\boldsymbol{x}}_t)\|^2 \leq \hat{B}. \tag{12}$$

Then, if $\left(\|\nabla F(\hat{\boldsymbol{x}}_t)\|^2\right)$ is unbounded, we have

$$\forall \epsilon > 0, \exists t_1 \in \mathbb{N}, \forall t \in \mathbb{N} \colon t \geq t_1 \Rightarrow \|\nabla F(\hat{\boldsymbol{x}}_t)\|^2 \geq \epsilon.$$

Therefore, from (12),

$$\hat{B} \geq \frac{2}{\sigma T} \sum_{t=1}^{T} \|\nabla F(\hat{\boldsymbol{x}}_t)\|^2$$

$$= \frac{2}{\sigma T} \left( \sum_{t=t_1}^{T} \|\nabla F(\hat{\boldsymbol{x}}_t)\|^2 + \sum_{t=1}^{t_1-1} \|\nabla F(\hat{\boldsymbol{x}}_t)\|^2 \right)$$

$$\geq \frac{2}{\sigma T} (T - t_1 + 1)\epsilon$$

$$= \frac{2}{\sigma} \left( 1 - \frac{t_1 - 1}{T} \right) \epsilon,$$

where we have used $\sum_{t=1}^{t_1-1} \|\nabla F(\hat{\boldsymbol{x}}_t)\|^2 \geq 0$. Hence, letting $\epsilon := \sigma \hat{B}$, we have

$$\exists t_1, \forall T \geq t_1: \ 2 \left( 1 - \frac{t_1 - 1}{T} \right) \hat{B} \leq \hat{B}.$$

Taking the limit of $T \to \infty$, we have $2\hat{B} \leq \hat{B}$. This is a contradiction. Hence, $\left( \|\nabla F(\hat{\boldsymbol{x}}_t)\|^2 \right)$ is bounded. Let its upper boundary be $B_2$. This completes the proof. $\qquad \square$

## D  Proof of the Theorems and Propositions

### D.1  Proof of Theorem 3.1

*Proof.* Lemma C.1, Lemma C.4, and $\eta_t := \frac{2}{\sigma t}$ guarantee that

$$F(\hat{\boldsymbol{x}}_t) - F(\boldsymbol{x}^\star) \leq \frac{1 - \sigma \eta_t}{2\eta_t} X_t - \frac{1}{2\eta_t} X_{t+1} + \frac{\eta_t}{2} \|\boldsymbol{g}_t\|^2$$

$$\leq \frac{1}{2\eta_t} \left\{ (1 - \sigma \eta_t) X_t - X_{t+1} \right\} + \frac{\eta_t B_2}{2}$$

$$= \frac{\sigma(t-2)}{4} X_t - \frac{\sigma t}{4} X_{t+1} + \frac{B_2}{\sigma t}.$$

Therefore, we have

$$(t-1) \left( F(\hat{\boldsymbol{x}}_t) - F(\boldsymbol{x}^\star) \right) \leq \frac{\sigma(t-2)(t-1)}{4} X_t - \frac{\sigma(t-1)t}{4} X_{t+1} + \frac{B_2(t-1)}{\sigma t}.$$

Summing over $t$, we find that

$$\sum_{t=1}^{T} (t-1) \left( F(\hat{\boldsymbol{x}}_t) - F(\boldsymbol{x}^\star) \right) \leq \frac{\sigma \cdot (-1) \cdot 0}{4} X_1 - \frac{\sigma(T-1)T}{4} X_{T+1} + \frac{B_2}{\sigma} \sum_{t=1}^{T} \frac{t-1}{t}$$

$$\leq \frac{B_2(T-1)}{\sigma}.$$

Then, we have

$$\frac{2}{(T-1)T} \sum_{t=1}^{T} (t-1) \left( F(\hat{\boldsymbol{x}}_t) - F(\boldsymbol{x}^\star) \right) \leq \frac{2B_2}{\sigma T}.$$

From the convexity of $F$,

$$F \left( \frac{2}{T(T-1)} \sum_{t=1}^{T} (t-1)\hat{\boldsymbol{x}}_t \right) \leq \frac{2}{T(T-1)} \sum_{t=1}^{T} (t-1) F(\hat{\boldsymbol{x}}_t).$$

Hence,

$$F\left(\frac{2}{T(T-1)}\sum_{t=1}^{T}(t-1)\hat{\boldsymbol{x}}_t\right) - F(\boldsymbol{x}^\star) \leq \frac{2B_2}{\sigma T} = \mathcal{O}\left(\frac{1}{T}\right).$$

In addition, since the minimum value is smaller than the mean, we have

$$\min_{t\in[T]}\left(F\left(\hat{\boldsymbol{x}}_t\right) - F(\boldsymbol{x}^\star)\right) \leq \frac{2B_2}{\sigma T} = \mathcal{O}\left(\frac{1}{T}\right).$$

This completes the proof. $\qquad\square$

## D.2 Proof of Theorem 3.2

The following proof uses the proof technique of Hazan et al. (2016).

*Proof.* From $M^p := \frac{1}{\alpha_0\epsilon}$, $\delta_1 := \frac{2L_g}{\sigma r}$, and $\gamma_{m+1} := \frac{(M-m)^p}{\{M-(m-1)\}^p}$, we have

$$\begin{aligned}
\delta_M &= \delta_1\left(\gamma_1\gamma_2\cdots\gamma_{M-1}\right)\\
&= \delta_1 \cdot \frac{(M-1)^p}{M^p} \cdot \frac{(M-2)^p}{(M-1)^p} \cdot \frac{(M-3)^p}{(M-2)^p}\cdots\frac{1}{2^p}\\
&= \delta_1 \cdot \frac{1}{M^p}\\
&= \delta_1\alpha_0\epsilon\\
&= \frac{2L_g\alpha_0\epsilon}{\sigma r}.
\end{aligned}$$

According to Theorem 3.1,

$$\begin{aligned}
\hat{f}_{\delta_M}(\boldsymbol{x}_{M+1}) - \hat{f}_{\delta_M}(\boldsymbol{x}^\star_{\delta_M}) &\leq \epsilon_M\\
&= \sigma\delta_M^2\\
&= \left(\frac{2L_g\alpha_0\epsilon}{\sqrt{\sigma}r}\right)^2
\end{aligned}$$

From Lemma 2.3 and 2.4,

$$\begin{aligned}
f(\boldsymbol{x}_{M+2}) - f(\boldsymbol{x}^\star) &= \left\{f(\boldsymbol{x}_{M+2}) - \hat{f}_{\delta_M}(\boldsymbol{x}_{M+2})\right\} + \left\{\hat{f}_{\delta_M}(\boldsymbol{x}^\star) - f(\boldsymbol{x}^\star)\right\} + \left\{\hat{f}_{\delta_M}(\boldsymbol{x}_{M+2}) - \hat{f}_{\delta_M}(\boldsymbol{x}^\star)\right\}\\
&\leq \left\{f(\boldsymbol{x}_{M+2}) - \hat{f}_{\delta_M}(\boldsymbol{x}_{M+2})\right\} + \left\{\hat{f}_{\delta_M}(\boldsymbol{x}^\star) - f(\boldsymbol{x}^\star)\right\} + \left\{\hat{f}_{\delta_M}(\boldsymbol{x}_{M+2}) - \hat{f}_{\delta_M}(\boldsymbol{x}^\star_{\delta_M})\right\}\\
&\leq \delta_M L_f + \delta_M L_f + \left\{\hat{f}_{\delta_M}(\boldsymbol{x}_{M+2}) - \hat{f}_{\delta_M}(\boldsymbol{x}^\star_{\delta_M})\right\}\\
&= 2\delta_M L_f + \left\{\hat{f}_{\delta_M}(\boldsymbol{x}_{M+2}) - \hat{f}_{\delta_M}(\boldsymbol{x}_{M+1})\right\} + \left\{\hat{f}_{\delta_M}(\boldsymbol{x}_{M+1}) - \hat{f}_{\delta_M}(\boldsymbol{x}^\star_{\delta_M})\right\}\\
&\leq 2\delta_M L_f + L_f\|\boldsymbol{x}_{M+2} - \boldsymbol{x}_{M+1}\| + \left\{\hat{f}_{\delta_M}(\boldsymbol{x}_{M+1}) - \hat{f}_{\delta_M}(\boldsymbol{x}^\star_{\delta_M})\right\}.
\end{aligned}$$

Then, we have

$$\begin{aligned}
f(\boldsymbol{x}_{M+2}) - f(\boldsymbol{x}^\star) &\leq 2\delta_M L_f + 2L_f d_M\delta_M + \epsilon_M\\
&\leq 2\delta_M L_f + 2L_f\bar{d}\delta_M + \epsilon_M\\
&= 2L_f\delta_M\left(1 + \bar{d}\right) + \epsilon_M,
\end{aligned}$$

where we have used $\|\boldsymbol{x}_{M+2} - \boldsymbol{x}_{M+1}\| \leq 2d_M\delta_M$ since $\boldsymbol{x}_{M+2}, \boldsymbol{x}_{M+1} \in N(\boldsymbol{x}^\star; d_M\delta_M)$, and $d_M \leq \bar{d} < +\infty$. Therefore,

$$f(\boldsymbol{x}_{M+2}) - f(\boldsymbol{x}^\star) \leq 2L_f \left(1 + \bar{d}\right) \frac{2L_f\alpha_0\epsilon}{\sigma r} + \left(\frac{2L_g\alpha_0\epsilon}{\sqrt{\sigma}r}\right)^2$$
$$= \frac{4L_f^2 \left(1 + \bar{d}\right)\alpha_0\epsilon}{\sigma r} + \left(\frac{2L_g\alpha_0\epsilon}{\sqrt{\sigma}r}\right)^2$$
$$\leq \epsilon,$$

where we have used $\alpha_0 := \min\left\{\frac{\sigma r}{8L_f^2\left(1+\bar{d}\right)}, \frac{\sqrt{\sigma}r}{2\sqrt{2}L_g}\right\}$.

Let $T_{\text{total}}$ be the total number of queries made by Algorithm 1; then,

$$T_{\text{total}} = \sum_{m=1}^{M+1} \frac{2B_2}{\sigma\epsilon_m} = \sum_{m=1}^{M+1} \frac{2B_2}{\sigma^2\delta_m^2}$$
$$= \frac{2B_2}{\sigma^2\delta_1^2}\left(1 + \frac{1}{\gamma_1^2} + \frac{1}{\gamma_1^2\gamma_2^2} + \cdots + \frac{2}{\gamma_1^2\gamma_2^2\cdots\gamma_{M-1}^2}\right)$$
$$= \frac{2B_2}{\sigma^2\delta_1^2}\left\{\frac{\left(\gamma_1^2\gamma_2^2\cdots\gamma_{M-1}^2\right) + \left(\gamma_2^2\gamma_3^2\cdots\gamma_{M-1}^2\right) + \cdots + \gamma_{M-1}^2 + 2}{\gamma_1^2\gamma_2^2\cdots\gamma_{M-1}^2}\right\}$$

From $\gamma_1\gamma_2\cdots\gamma_{M-1} = \frac{1}{M^p}$,

$$T_{\text{total}} = \frac{2B_2}{\sigma^2\delta_1^2} \cdot \frac{\frac{1}{M^{2p}} + \frac{1}{(M-1)^{2p}} + \frac{1}{(M-2)^{2p}} + \cdots \frac{1}{2^{2p}} + 2}{\frac{1}{M^{2p}}}$$
$$\leq \frac{2B_2}{\sigma^2\delta_1^2} \cdot M^{2p}(M+1)$$
$$= \frac{2B_2}{\sigma^2\delta_1^2} \cdot M^{2p+1} + \frac{2B_2}{\sigma^2\delta_1^2} \cdot M^{2p}$$
$$= \frac{2B_2}{\sigma^2\delta_1^2}\left(\frac{1}{\alpha_0\epsilon}\right)^{\frac{1}{p}+2} + \frac{2B_2}{\sigma^2\delta_1^2}\left(\frac{1}{\alpha_0\epsilon}\right)^2$$
$$= \mathcal{O}\left(\frac{1}{\epsilon^{\frac{1}{p}+2}}\right),$$

This completes the proof. □

### D.3 Proof of Theorem 3.3

*Proof.* Lemma C.1 guarantees that

$$F(\hat{\boldsymbol{x}}_t) - F(\boldsymbol{x}^\star) \leq \frac{1 - \sigma\eta_t}{2\eta_t}X_t - \frac{1}{2\eta_t}X_{t+1} + \frac{\eta_t}{2}\|\boldsymbol{g}_t\|^2$$
$$= \frac{1 - \sigma\eta}{2\eta}\left(X_t - X_{t+1}\right) - \frac{\sigma}{2}X_{t+1} + \frac{\eta}{2}\|\boldsymbol{g}_t\|^2.$$

From $\eta < \min\left\{\frac{1}{\sigma}, \frac{2}{L_g}\right\}$ and Lemma C.3, by summing over $t$ we find that

$$\frac{1}{T}\sum_{t=1}^{T}\left(F(\hat{\boldsymbol{x}}_t) - F(\boldsymbol{x}^\star)\right) \leq \frac{1-\sigma\eta}{2\eta T}\left(X_1 - X_{T+1}\right) - \frac{\sigma}{2T}\sum_{t=1}^{T}X_{t+1} + \frac{\eta}{2T}\sum_{t=1}^{T}\|\boldsymbol{g}_t\|^2$$

$$\leq \frac{1-\sigma\eta}{2\eta T}X_1 + \frac{\eta}{2T}\sum_{t=1}^{T}\|\boldsymbol{g}_t\|^2$$

$$\leq \underbrace{\frac{(1-\sigma\eta)X_1}{2\eta}}_{=:H_1}\frac{1}{T} + \underbrace{\frac{F(\hat{\boldsymbol{x}}_1) - F(\boldsymbol{x}^\star)}{(2-L_g\eta)}}_{=:H_2}\frac{1}{T}$$

$$= \underbrace{(H_1 + H_2)}_{=:H_3}\frac{1}{T}$$

$$= \frac{H_3}{T},$$

where $H_3 > 0$ is a nonnegative constant. From the convexity of $F$,

$$F\left(\frac{1}{T}\sum_{t=1}^{T}\hat{\boldsymbol{x}}_t\right) \leq \frac{1}{T}\sum_{t=1}^{T}F(\hat{\boldsymbol{x}}_t).$$

Hence,

$$F\left(\frac{1}{T}\sum_{t=1}^{T}\hat{\boldsymbol{x}}_t\right) - F(\boldsymbol{x}^\star) \leq \frac{H_3}{T} = \mathcal{O}\left(\frac{1}{T}\right).$$

In addition, since the minimum value is smaller than the mean, we have

$$\min_{t\in[T]}\left(F\left(\hat{\boldsymbol{x}}_t\right) - F(\boldsymbol{x}^\star)\right) \leq \frac{H_3}{T} = \mathcal{O}\left(\frac{1}{T}\right).$$

This completes the proof. $\qquad\square$

### D.4 Proof of Theorem 3.4

The following proof uses the proof technique of Hazan et al. (2016).

*Proof.* According to $\delta_{m+1} := \frac{\eta_{m+1}C}{\sqrt{b_{m+1}}}$ and $\frac{\kappa_m}{\sqrt{\lambda_m}} = \gamma_m$, we have

$$\delta_{m+1} := \frac{\eta_{m+1}C}{\sqrt{b_{m+1}}}$$

$$= \frac{\kappa_m\eta_mC}{\sqrt{\lambda_m}\sqrt{b_m}}$$

$$= \frac{\kappa_m}{\sqrt{\lambda_m}}\delta_m$$

$$= \gamma_m\delta_m.$$

Therefore, from $M^p := \frac{1}{\alpha_0 \epsilon}$, $\delta_1 := \frac{\eta_1 C}{\sqrt{b_1}}$, and $\gamma_m := \frac{(M-m)^p}{\{M-(m-1)\}^p}$, then

$$
\begin{aligned}
\delta_M &= \delta_1 \left( \gamma_1 \gamma_2 \cdots \gamma_{M-1} \right) \\
&= \delta_1 \cdot \frac{(M-1)^p}{M^p} \cdot \frac{(M-2)^p}{(M-1)^p} \cdot \frac{(M-3)^p}{(M-2)^p} \cdots \frac{1}{2^p} \\
&= \delta_1 \cdot \frac{1}{M^p} \\
&= \delta_1 \alpha_0 \epsilon \\
&= \frac{\eta_1 C \alpha_0 \epsilon}{\sqrt{b_1}}.
\end{aligned}
$$

According to Theorem 3.3,

$$
\begin{aligned}
\hat{f}_{\delta_M}(\boldsymbol{x}_{M+1}) - \hat{f}_{\delta_M}(\boldsymbol{x}_{\delta_M}^\star) &\le \epsilon_M \\
&= \sigma \delta_M^2 \\
&= \left( \frac{\sqrt{\sigma} \eta_1 C \alpha_0 \epsilon}{\sqrt{b_1}} \right)^2
\end{aligned}
$$

As in the proof of Theorem 3.2, we have

$$
f(\boldsymbol{x}_{M+2}) - f(\boldsymbol{x}^\star) \le 2 L_f \delta_M \left( 1 + \bar{d} \right) + \epsilon_M.
$$

Therefore,

$$
\begin{aligned}
f(\boldsymbol{x}_{M+2}) - f(\boldsymbol{x}^\star) &\le 2 L_f \left( 1 + \bar{d} \right) \frac{\eta_1 C \alpha_0 \epsilon}{\sqrt{b_1}} + \left( \frac{\sqrt{\sigma} \eta_1 C \alpha_0 \epsilon}{\sqrt{b_1}} \right)^2 \\
&= \frac{2 L_f \left( 1 + \bar{d} \right) \eta_1 C \alpha_0 \epsilon}{\sqrt{b_1}} + \left( \frac{\sqrt{\sigma} \eta_1 C \alpha_0 \epsilon}{\sqrt{b_1}} \right)^2 \\
&\le \epsilon,
\end{aligned}
$$

where we have used $\alpha_0 := \min \left\{ \frac{\sqrt{b_1}}{4 L_f \eta_1 C \left( 1 + \bar{d} \right)}, \frac{\sqrt{b_1}}{\sqrt{2\sigma} \eta_1 C} \right\}$.

Let $T_{\text{total}}$ be the total number of queries made by Algorithm 3; then,

$$
\begin{aligned}
T_{\text{total}} &= \sum_{m=1}^{M+1} \frac{H_3}{\epsilon_m} = \sum_{m=1}^{M+1} \frac{H_3}{\sigma^2 \delta_m^2} \\
&= \frac{H_3}{\sigma^2 \delta_1^2} + \frac{H_3}{\sigma^2 \delta_2^2} + \cdots + \frac{2 H_3}{\sigma^2 \delta_M^2} \\
&\le \frac{H_3 (M+1)}{\sigma^2 \delta_M^2} \\
&\le \frac{H_3 (M+1)}{\sigma^2 \delta_M^2} \\
&= \frac{H_3 (M+1)}{\sigma^2 \delta_1^2 \frac{1}{M^{2p}}} \\
&= \frac{H_3 M^{2p} (M+1)}{\sigma^2 \delta_1^2} \\
&= \frac{H_3 M^{2p+1}}{\sigma^2 \delta_1^2} + \frac{H_3 M^{2p}}{\sigma^2 \delta_1^2}.
\end{aligned}
$$

From $M^p := \frac{1}{\alpha_0 \epsilon}$,

$$
\begin{aligned}
T_{\text{total}} &= \frac{H_3 \left( \frac{1}{\alpha_0 \epsilon} \right)^{\frac{1}{p}+2}}{\sigma^2 \delta_1^2} + \frac{H_3 \left( \frac{1}{\alpha_0 \epsilon} \right)^2}{\sigma^2 \delta_1^2} \\
&\leq \frac{2H_3 \left( \frac{1}{\alpha_0 \epsilon} \right)^{\frac{1}{p}+2}}{\sigma^2 \delta_1^2} \\
&= \frac{2H_3 \left( \frac{1}{\alpha_0 \epsilon} \right)^{\frac{1}{p}}}{\sigma^2 \delta_1^2 (\alpha_0 \epsilon)^2} \\
&= \frac{2H_3}{\sigma^2 \delta_1^2 (\alpha_0 \epsilon)^{\frac{1}{p}+2}} \\
&\leq \frac{2H_3}{\sigma^2 \delta_1^2 (\alpha_0 \epsilon)^{\frac{1}{p}}} \\
&= \mathcal{O} \left( \frac{1}{\epsilon^{\frac{1}{p}}} \right).
\end{aligned}
$$

This completes the proof. $\qquad\square$

### D.5 Proof of Proposition 3.1

*Proof.* For all $\boldsymbol{x} \in N(\boldsymbol{x}^\star; a_m r) \backslash \{\boldsymbol{x}^\star\}$ ($a_m \geq 0$), and all $\boldsymbol{u}_m \sim B(\boldsymbol{0}; 1)$, the quadratic equation

$$
\delta_m^2 - 2 \langle \boldsymbol{x}^\star - \boldsymbol{x}, \boldsymbol{u}_m \rangle \delta_m + (a_m^2 - 1)r^2 = 0 \tag{13}
$$

for $\delta_m \in \mathbb{R}$ has solutions with probability $p(a_m)$ when $a_m > 1$ and always has solutions when $0 \leq a_m \leq 1$.

Let us derive $p(a_m)$. When $a_m > 1$, the condition for the discriminant equation of (13) to be positive is as follows:

$$
-1 \leq \cos\theta \leq -\frac{r\sqrt{a_m^2 - 1}}{\|\boldsymbol{x}^\star - \boldsymbol{x}\| \|\boldsymbol{u}_m\|}, \quad \text{or} \quad \frac{r\sqrt{a_m^2 - 1}}{\|\boldsymbol{x}^\star - \boldsymbol{x}\| \|\boldsymbol{u}_m\|} \leq \cos\theta \leq 1, \tag{14}
$$

where $\theta$ is the angle between $\boldsymbol{u}_m \sim B(\boldsymbol{0}; 1)$ and $\boldsymbol{x}^\star - \boldsymbol{x}$. Note that $\cos\theta$ can be positive or negative because $\delta_m \in \mathbb{R}$. Since the random variable $\boldsymbol{u}_m$ is sampled uniformly from the $B(\boldsymbol{0}; 1)$, the probability that $\boldsymbol{u}_m$ satisfies (14) is less than

$$
p(a_m) := \frac{\arccos \left( \frac{r\sqrt{a_m^2 - 1}}{\|\boldsymbol{x}^\star - \boldsymbol{x}\| \|\boldsymbol{u}_m\|} \right)}{\pi},
$$

for $\delta_m > 0$ and $\delta_m < 0$, respectively.

Now let us consider the solution of the quadratic inequality,

$$
\|\boldsymbol{u}_m\|^2 \delta_m^2 - 2 \langle \boldsymbol{x}^\star - \boldsymbol{x}, \boldsymbol{u}_m \rangle \delta_m + (a_m^2 - 1)r^2 \leq 0 \tag{15}
$$

for $\delta_m \in \mathbb{R}$.

(i) When $a_m > 1$, (15) has one or two solutions with probability $p(a_m)$ or less. When $\frac{r\sqrt{a_m^2 - 1}}{\|\boldsymbol{x}^\star - \boldsymbol{x}\| \|\boldsymbol{u}_m\|} \leq \cos\theta \leq 1$, let the larger solution be $D_m^+ > 0$ and the smaller one be $D_m^- > 0$; we can express these solutions as follows:

$$
D_m^+(\boldsymbol{x}, \boldsymbol{u}_m) := \|\boldsymbol{x}^\star - \boldsymbol{x}\| \|\boldsymbol{u}_m\| \cos\theta + \sqrt{\|\boldsymbol{x}^\star - \boldsymbol{x}\|^2 \|\boldsymbol{u}_m\|^2 \cos^2\theta - r^2(a_m^2 - 1)},
$$

$$
D_m^-(\boldsymbol{x}, \boldsymbol{u}_m) := \|\boldsymbol{x}^\star - \boldsymbol{x}\| \|\boldsymbol{u}_m\| \cos\theta - \sqrt{\|\boldsymbol{x}^\star - \boldsymbol{x}\|^2 \|\boldsymbol{u}_m\|^2 \cos^2\theta - r^2(a_m^2 - 1)}.
$$

Moreover, we define $\delta_m^+$ and $\delta_m^-$ as follows:

$$\delta_m^+ := \sup_{\boldsymbol{x} \in N(\boldsymbol{x}^\star; a_m r) \setminus \{\boldsymbol{x}^\star\}} \mathbb{E}_{\boldsymbol{u}_m \sim B(\boldsymbol{0}; 1)} \left[ D_m^+(\boldsymbol{x}, \boldsymbol{u}_m) \right],$$

$$\delta_m^- := \sup_{\boldsymbol{x} \in N(\boldsymbol{x}^\star; a_m r) \setminus \{\boldsymbol{x}^\star\}} \mathbb{E}_{\boldsymbol{u}_m \sim B(\boldsymbol{0}; 1)} \left[ D_m^-(\boldsymbol{x}, \boldsymbol{u}_m) \right].$$

Thus, the solution $D_m(\boldsymbol{x}, \boldsymbol{u}_m)$ to (15) is

$$0 < D_m^-(\boldsymbol{x}, \boldsymbol{u}_m) < D_m(\boldsymbol{x}, \boldsymbol{u}_m) < D_m^+(\boldsymbol{x}, \boldsymbol{u}_m)$$

when $\frac{r\sqrt{a_m^2 - 1}}{\|\boldsymbol{x}^\star - \boldsymbol{x}\| \|\boldsymbol{u}_m\|} \le \cos\theta \le 1$, and

$$-D_m^+(\boldsymbol{x}, \boldsymbol{u}_m) < D_m(\boldsymbol{x}, \boldsymbol{u}_m) < -D_m^-(\boldsymbol{x}, \boldsymbol{u}_m) < 0$$

when $-1 \le \cos\theta \le -\frac{r\sqrt{a_m^2 - 1}}{\|\boldsymbol{x}^\star - \boldsymbol{x}\| \|\boldsymbol{u}_m\|}$. Hence, let $\delta_m := \sup_{\boldsymbol{x} \in N(\boldsymbol{x}^\star; a_m r) \setminus \{\boldsymbol{x}^\star\}} \mathbb{E}_{\boldsymbol{u}_m \sim B(\boldsymbol{0}; 1)} \left[ D_m(\boldsymbol{x}, \boldsymbol{u}_m) \right]$, then we have

$$|\delta_m^-| \le |\delta_m| \le |\delta_m^+|.$$

(ii) When $a_m \le 1$, (15) always has one or two solutions. The two solutions are defined as in (i). Then, the solution to (15) is

$$D_m^-(\boldsymbol{x}, \boldsymbol{u}_m) < D_m(\boldsymbol{x}, \boldsymbol{u}_m) < D_m^+(\boldsymbol{x}, \boldsymbol{u}_m)$$

when $\frac{r\sqrt{a_m^2 - 1}}{\|\boldsymbol{x}^\star - \boldsymbol{x}\| \|\boldsymbol{u}_m\|} \le \cos\theta \le 1$, and

$$-D_m^+(\boldsymbol{x}, \boldsymbol{u}_m) < D_m(\boldsymbol{x}, \boldsymbol{u}_m) < -D_m^-(\boldsymbol{x}, \boldsymbol{u}_m) \quad (-D_m^+(\boldsymbol{x}, \boldsymbol{u}_m) < 0, -D_m^-(\boldsymbol{x}, \boldsymbol{u}_m) > 0)$$

when $-1 \le \cos\theta \le -\frac{r\sqrt{a_m^2 - 1}}{\|\boldsymbol{x}^\star - \boldsymbol{x}\| \|\boldsymbol{u}_m\|}$. Hence, we have

$$|\delta_m| \le |\delta_m^-|.$$

From (i) and (ii), (15) may have a solution for all $a_m > 0$ when $|\delta_m| = |\delta_m^-|$. Therefore, suppose $|\delta_m| = |\delta_m^-|$; then,

$$
\begin{aligned}
r^2 &\ge a_m^2 r^2 - 2\delta_m \langle \boldsymbol{x}^\star - \boldsymbol{x}, \boldsymbol{u}_m \rangle + |\delta_m|^2 \\
&\ge a_m^2 r^2 - 2\delta_m \langle \boldsymbol{x}^\star - \boldsymbol{x}, \boldsymbol{u}_m \rangle + |\delta_m|^2 \|\boldsymbol{u}_m\|^2 \\
&> \|\boldsymbol{x} - \boldsymbol{x}^\star\|^2 - 2\delta_m \langle \boldsymbol{x}^\star - \boldsymbol{x}, \boldsymbol{u}_m \rangle + |\delta_m|^2 \|\boldsymbol{u}_m\|^2 \\
&= \|\boldsymbol{x} + \delta_m \boldsymbol{u}_m - \boldsymbol{x}^\star\|^2.
\end{aligned}
$$

This means that $\boldsymbol{x} + \delta_m \boldsymbol{u}_m \in N(\boldsymbol{x}^\star; r)$ $(\delta_m \in \mathbb{R})$, where $\boldsymbol{u}_m \sim B(\boldsymbol{0}; 1)$. Hence, for all $\boldsymbol{x}, \boldsymbol{y} \in N(\boldsymbol{x}^\star; a_m r) \subset \mathbb{R}^d$ $(a_m > \sqrt{2})$,

$$
\begin{aligned}
\left\langle \nabla \hat{f}_{\delta_m}(\boldsymbol{x}) - \nabla \hat{f}_{\delta_m}(\boldsymbol{y}), \boldsymbol{x} - \boldsymbol{y} \right\rangle &= \langle \nabla \mathbb{E}_{\boldsymbol{u}}[f(\boldsymbol{x} + \delta_m \boldsymbol{u})] - \nabla \mathbb{E}_{\boldsymbol{u}}[f(\boldsymbol{y} + \delta_m \boldsymbol{u})], \boldsymbol{x} - \boldsymbol{y} \rangle \\
&= \langle \mathbb{E}_{\boldsymbol{u}}[\nabla f(\boldsymbol{x} + \delta_m \boldsymbol{u})] - \mathbb{E}_{\boldsymbol{u}}[\nabla f(\boldsymbol{y} + \delta_m \boldsymbol{u})], \boldsymbol{x} - \boldsymbol{y} \rangle \\
&= \langle \mathbb{E}_{\boldsymbol{u}}[\nabla f(\boldsymbol{x} + \delta_m \boldsymbol{u}) - \nabla f(\boldsymbol{y} + \delta_m \boldsymbol{u})], \boldsymbol{x} - \boldsymbol{y} \rangle \\
&= \mathbb{E}_{\boldsymbol{u}}[\langle \nabla f(\boldsymbol{x} + \delta_m \boldsymbol{u}) - \nabla f(\boldsymbol{y} + \delta_m \boldsymbol{u}), \boldsymbol{x} - \boldsymbol{y} \rangle] \\
&\ge \mathbb{E}_{\boldsymbol{u}}[\sigma \|(\boldsymbol{x} + \delta_m \boldsymbol{u}) - (\boldsymbol{y} + \delta_m \boldsymbol{u})\|^2] \\
&= \mathbb{E}_{\boldsymbol{u}}[\sigma \|\boldsymbol{x} - \boldsymbol{y}\|^2] \\
&= \sigma \|\boldsymbol{x} - \boldsymbol{y}\|^2.
\end{aligned}
$$

This means that, if $|\delta_m| = |\delta_m^-|$ holds, then $\hat{f}_{\delta_m}$ is $\sigma$-strongly convex on $N(\boldsymbol{x}^\star; a_m r)$ $(a_m > \sqrt{2})$ when $f$ is $\sigma$-strongly convex on $B(\boldsymbol{x}^\star; r)$. Also, if we define $d_m := \frac{a_m r}{|\delta_m^-|}$, then $d_m |\delta_m| \le a_m r$ holds; i.e., $\hat{f}_{\delta_m}$ is $\sigma$-strongly convex on $N(\boldsymbol{x}^\star; d_m |\delta_m|)$. This completes the proof. $\qquad\square$

**Remark:** In the end, $|\delta_m|$ must be equal to $|\delta_m^-|$. We can show that $|\delta_m^-|$ is non-zero. Suppose that (14) holds, then

$$
|\delta_m^-| := \sup_{\boldsymbol{x} \in N(\boldsymbol{x}^\star; a_m r) \setminus \{\boldsymbol{x}^\star\}} \mathbb{E}_{\boldsymbol{u}_m \sim B(\boldsymbol{0};1)} \left[ \left| \|\boldsymbol{x}^\star - \boldsymbol{x}\| \|\boldsymbol{u}_m\| \cos\theta - \sqrt{\|\boldsymbol{x}^\star - \boldsymbol{x}\|^2 \|\boldsymbol{u}_m\|^2 \cos^2\theta - r^2(a_m^2 - 1)} \right| \right]
$$

$$
> \sup_{\boldsymbol{x} \in N(\boldsymbol{x}^\star; a_m r) \setminus \{\boldsymbol{x}^\star\}} \mathbb{E}_{\boldsymbol{u}_m \sim B(\boldsymbol{0};1)} \left[ \left| r\sqrt{a_m^2 - 1} - \sqrt{a_m^2 r^2 - r^2(a_m^2 - 1)} \right| \right]
$$

$$
= |r\sqrt{a_m^2 - 1} - r|
$$

$$
= r\left( \sqrt{a_m^2 - 1} - 1 \right)
$$

$$
> 0,
$$

where we have used $\|\boldsymbol{x}^\star - \boldsymbol{x}\| < a_m r, \|\boldsymbol{u}_m\| \leq 1, \frac{r\sqrt{a_m^2-1}}{\|\boldsymbol{x}^\star - \boldsymbol{x}\| \|\boldsymbol{u}_m\|} \leq |\cos\theta| \leq 1$, and $a_m > \sqrt{2}$.

### D.6 Proof of Proposition 3.2

*Proof.* From Proposition 3.1, for all $|\delta_m| = |\delta_m^-|$, $\hat{f}_{\delta_m}$ is $\sigma$-strongly convex, i.e.,

$$
\sigma \|\boldsymbol{x}^\star - \boldsymbol{x}_{\delta_{m-1}}^\star\|^2 \leq \left\langle \boldsymbol{x}^\star - \boldsymbol{x}_{\delta_{m-1}}^\star, \nabla\hat{f}_{\delta_m}(\boldsymbol{x}^\star) - \nabla\hat{f}_{\delta_m}(\boldsymbol{x}_{\delta_{m-1}}^\star) \right\rangle
$$

$$
\leq \left\| \boldsymbol{x}^\star - \boldsymbol{x}_{\delta_{m-1}}^\star \right\| \left\| \nabla\hat{f}_{\delta_m}(\boldsymbol{x}^\star) - \nabla\hat{f}_{\delta_m}(\boldsymbol{x}_{\delta_{m-1}}^\star) \right\|,
$$

where we have used the Cauchy-Schwarz inequality and $\left\| \nabla\hat{f}_{\delta_m}(\boldsymbol{x}_{\delta_m}^\star) \right\| = \boldsymbol{0}$. Accordingly, we have

$$
\left\| \boldsymbol{x}^\star - \boldsymbol{x}_{\delta_{m-1}}^\star \right\| \left( \sigma \left\| \boldsymbol{x}^\star - \boldsymbol{x}_{\delta_{m-1}}^\star \right\| - \left\| \nabla\hat{f}_{\delta_m}(\boldsymbol{x}^\star) - \nabla\hat{f}_{\delta_m}(\boldsymbol{x}_{\delta_{m-1}}^\star) \right\| \right) \leq 0.
$$

Because $\left\| \boldsymbol{x}^\star - \boldsymbol{x}_{\delta_{m-1}}^\star \right\| \geq 0$ and Lemma 2.2,

$$
\left\| \boldsymbol{x}^\star - \boldsymbol{x}_{\delta_{m-1}}^\star \right\| \leq \frac{\left\| \nabla\hat{f}_{\delta_m}(\boldsymbol{x}^\star) - \nabla\hat{f}_{\delta_m}(\boldsymbol{x}_{\delta_{m-1}}^\star) \right\|}{\sigma}
$$

$$
\leq \frac{L_g \left\| \boldsymbol{x}_{\delta_{m-1}}^\star - \boldsymbol{x}^\star \right\|}{\sigma}.
$$

Hence,

$$
\left\| \boldsymbol{x}_{\delta_m}^\star - \boldsymbol{x}_{\delta_{m+1}}^\star \right\| \leq \left\| \boldsymbol{x}_{\delta_m}^\star - \boldsymbol{x}^\star \right\| + \left\| \boldsymbol{x}_{\delta_{m+1}}^\star - \boldsymbol{x}^\star \right\|
$$

$$
\leq \frac{L_g}{\sigma} \left( \left\| \boldsymbol{x}_{\delta_m}^\star - \boldsymbol{x}^\star \right\| + \left\| \boldsymbol{x}_{\delta_{m+1}}^\star - \boldsymbol{x}^\star \right\| \right)
$$

$$
\leq \frac{2L_g}{\sigma} \max\left\{ \left\| \boldsymbol{x}_{\delta_m}^\star - \boldsymbol{x}^\star \right\|, \left\| \boldsymbol{x}_{\delta_{m+1}}^\star - \boldsymbol{x}^\star \right\| \right\}
$$

$$
\leq |\delta_m|(1 - \gamma_m)
$$

$$
= |\delta_m| - |\delta_{m+1}|
$$

This completes the proof. $\qquad\square$

### D.7 Proof of Proposition 3.3

*Proof.* By using the triangle inequality, we have, for all $m \in [M]$,

$$
\begin{aligned}
\|\boldsymbol{x}_{\delta_m}^\star - \boldsymbol{x}^\star\| &= \|\boldsymbol{x}_{\delta_m}^\star - \boldsymbol{x}_{\delta_{m+1}}^\star + \boldsymbol{x}_{\delta_{m+1}}^\star - \boldsymbol{x}^\star\| \\
&\leq \|\boldsymbol{x}_{\delta_m}^\star - \boldsymbol{x}_{\delta_{m+1}}^\star\| + \|\boldsymbol{x}_{\delta_{m+1}}^\star - \boldsymbol{x}^\star\| \\
&\leq \|\boldsymbol{x}_{\delta_m}^\star - \boldsymbol{x}_{\delta_{m+1}}^\star\| + \|\boldsymbol{x}_{\delta_{m+1}}^\star - \boldsymbol{x}_{\delta_{m+2}}^\star\| + \cdots + \|\boldsymbol{x}_{\delta_M}^\star - \boldsymbol{x}_{\delta_{M+1}}^\star\| + \|\boldsymbol{x}_{\delta_{M+1}}^\star - \boldsymbol{x}^\star\| \\
&\leq (|\delta_m| - |\delta_{m+1}|) + (|\delta_{m+1}| - |\delta_{m+2}|) + \cdots + (|\delta_M| - |\delta_{M+1}|) + 0 \\
&= |\delta_m|,
\end{aligned}
\tag{16}
$$

where we have used $\boldsymbol{x}_{\delta_{M+1}}^\star = \boldsymbol{x}^\star$, $\delta_{M+1} = 0$. Therefore, from $d_m > 1$, we have

$$
\|\boldsymbol{x}_{\delta_m}^\star - \boldsymbol{x}^\star\| < d_m |\delta_m|.
$$

This completes the proof of Proposition 3.3(i). In addition, if $\gamma_m \in (\frac{1}{d_{m+1}}, 1)$ holds, from (16),

$$
\begin{aligned}
\|\boldsymbol{x}_{\delta_{m-1}}^\star - \boldsymbol{x}^\star\| &\leq \frac{|\delta_m|}{\gamma_{m-1}} \\
&< d_m |\delta_m|.
\end{aligned}
$$

This completes the proof of Proposition 3.3(ii). $\qquad\square$

## E   Additional Experimental Results

For the sake of fairness, we provide here a version of Figures 7-9 with the number of gradient queries on the horizontal axis. Since $b$ stochastic gradients are computed per epoch, the number of gradient queries is $Tb$, where $T$ means the number of steps and $b$ means the batch size.

## F   Full Experimental Results for Section 4.2

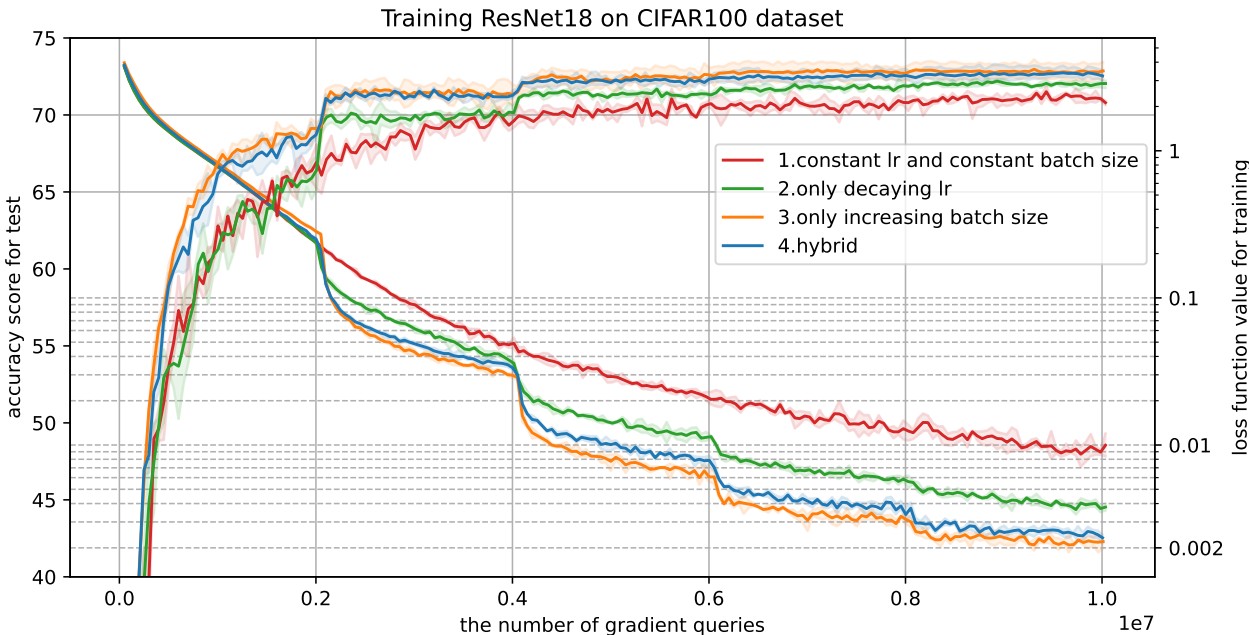

Figure 12: Accuracy score for testing and loss function value for training versus the number of gradient queries in training ResNet18 on the CIFAR100 dataset. The solid line represents the mean value, and the shaded area represents the maximum and minimum over three runs. In method 1, the learning rate and the batch size were fixed at 0.1 and 128, respectively. In method 2, the learning rate decreased every 40 epochs as in $\left[0.1, \frac{1}{10\sqrt{2}}, 0.05, \frac{1}{20\sqrt{2}}, 0.025\right]$ and the batch size was fixed at 128. In method 3, the learning rate was fixed at 0.1, and the batch size was increased as $[16, 32, 64, 128, 256]$. In method 4, the learning rate was decreased as $\left[0.1, \frac{\sqrt{3}}{20}, 0.075, \frac{3\sqrt{3}}{80}, 0.05625\right]$ and the batch size was increased as $[32, 48, 72, 108, 162]$. **This graph shows almost the same results as Figure 7.**

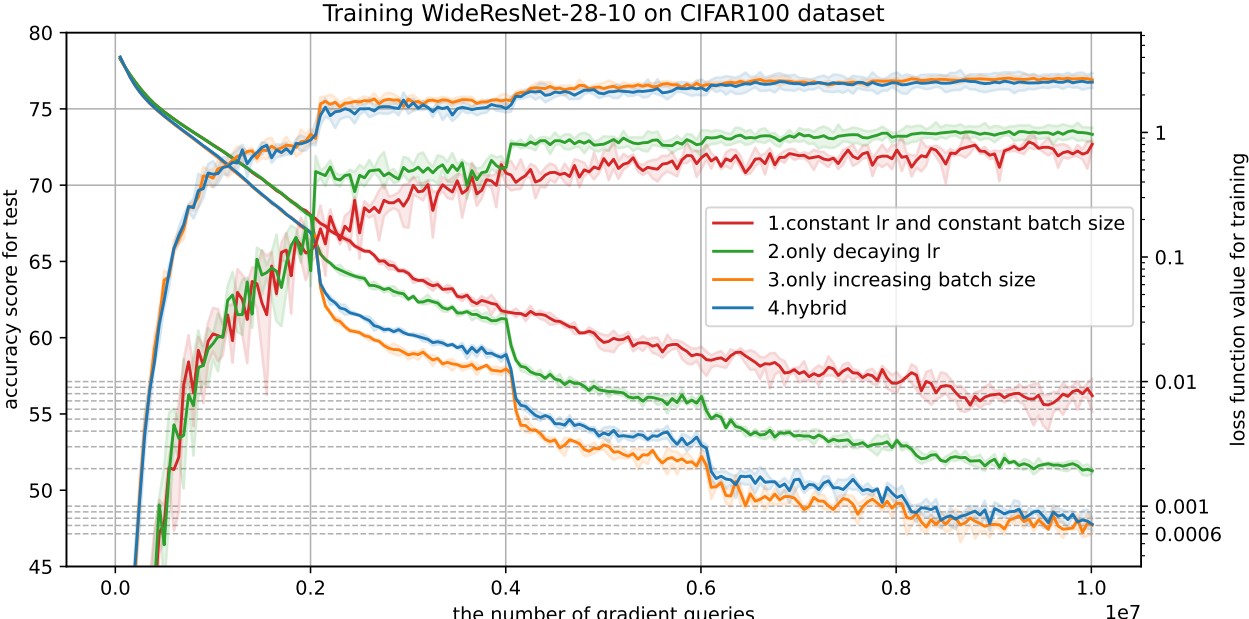

Figure 13: Accuracy score for testing and loss function value for training versus the number of gradient queries in training WideResNet-28-10 on the CIFAR100 dataset. The solid line represents the mean value, and the shaded area represents the maximum and minimum over three runs. In method 1, the learning rate and batch size were fixed at 0.1 and 128, respectively. In method 2, the learning rate was decreased every 40 epochs as $\left[0.1, \frac{1}{10\sqrt{2}}, 0.05, \frac{1}{20\sqrt{2}}, 0.025\right]$ and the batch size was fixed at 128. In method 3, the learning rate was fixed at 0.1, and the batch size increased as $[8, 16, 32, 64, 128]$. In method 4, the learning rate decreased as $\left[0.1, \frac{\sqrt{3}}{20}, 0.075, \frac{3\sqrt{3}}{80}, 0.05625\right]$ and the batch size increased as $[8, 12, 18, 27, 40]$. **This graph shows almost the same results as Figure 8.**

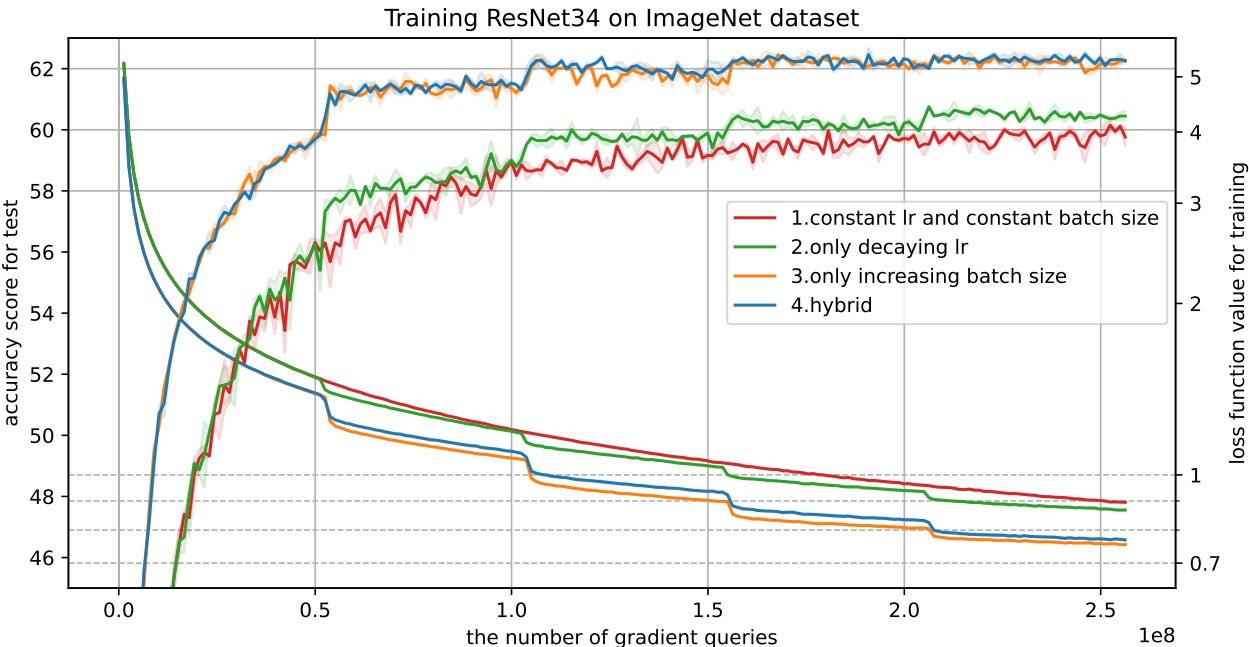

Figure 14: Accuracy score for testing and loss function value for training versus the number of gradient queries in training ResNet34 on the ImageNet dataset. The solid line represents the mean value, and the shaded area represents the maximum and minimum over three runs. In method 1, the learning rate and batch size were fixed at 0.1 and 256, respectively. In method 2, the learning rate was decreased every 40 epochs as $\left[0.1, \frac{1}{10\sqrt{2}}, 0.05, \frac{1}{20\sqrt{2}}, 0.025\right]$ and the batch size was fixed at 256. In method 3, the learning rate was fixed at 0.1, and the batch size was increased as $[32, 64, 128, 256, 512]$. In method 4, the learning rate was decreased as $\left[0.1, \frac{\sqrt{3}}{20}, 0.075, \frac{3\sqrt{3}}{80}, 0.05625\right]$ and the batch size was increased as $[32, 48, 72, 108, 162]$. **This graph shows almost the same results as Figure 9.**

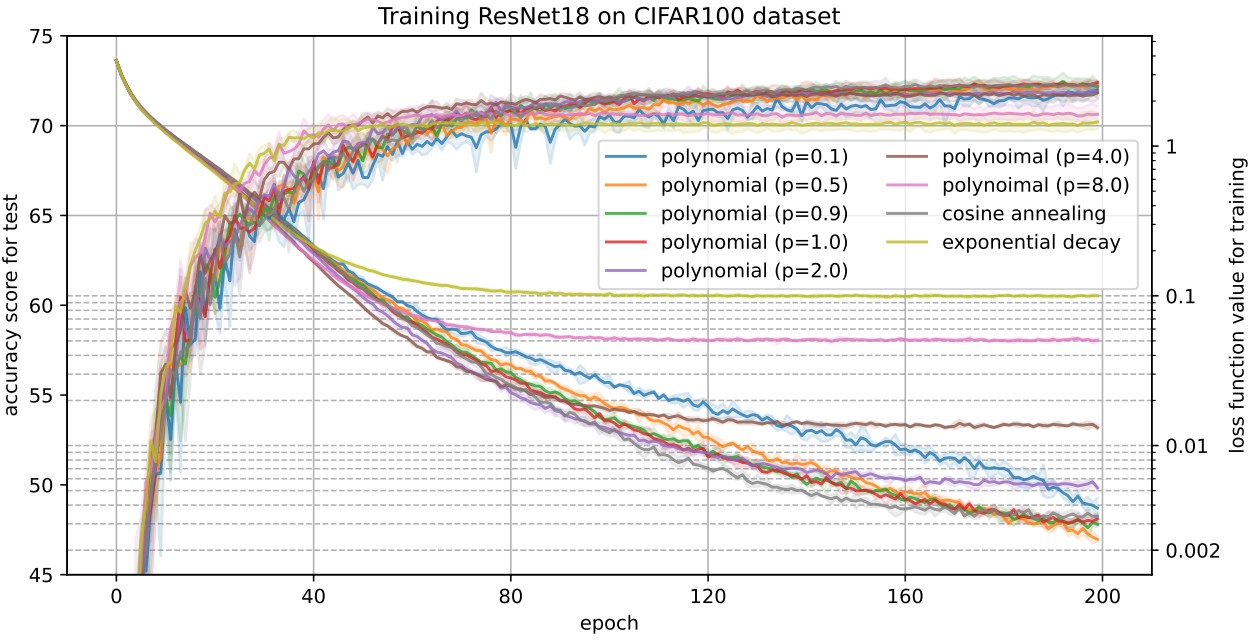

Figure 15: Accuracy score for testing and loss function value for training versus epochs in training of ResNet18 on the CIFAR100 dataset. The solid line represents the mean value, and the shaded area represents the maximum and minimum over three runs. This is the full version of Figure 10.

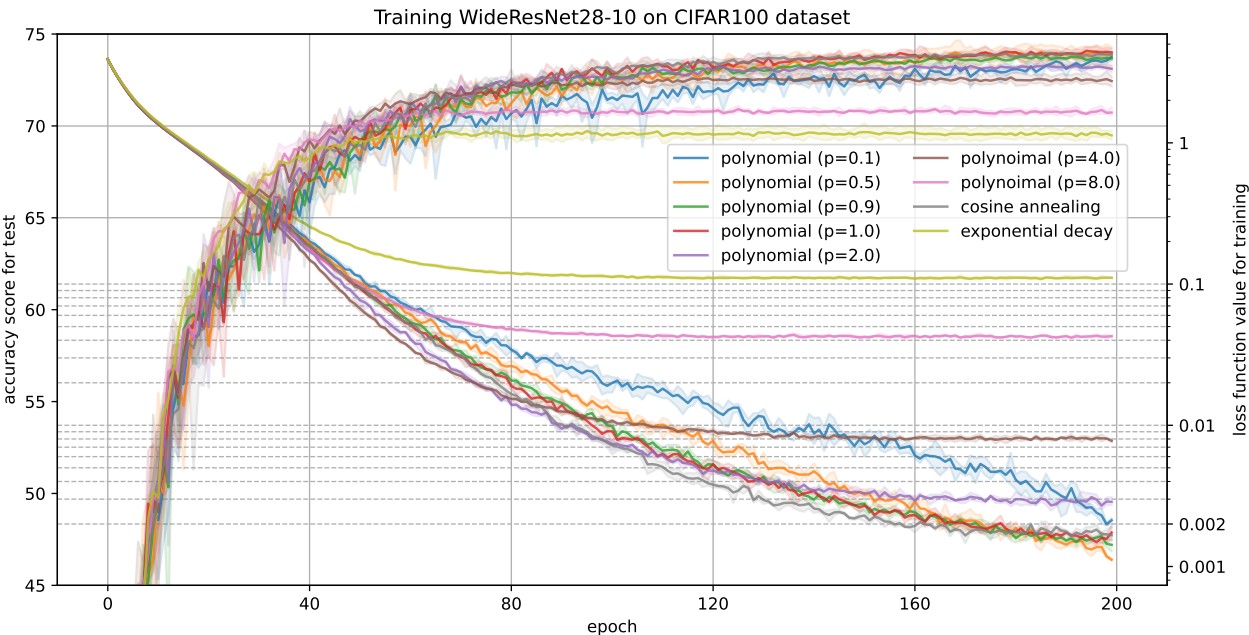

Figure 16: Accuracy score for testing and loss function value for training versus epochs in training of WideResNet-28-10 on the CIFAR100 dataset. The solid line represents the mean value, and the shaded area represents the maximum and minimum over three runs. This is the full version of Figure 11.

