# OpenReview forum: "Using Stochastic Gradient Descent to Smooth Nonconvex Functions: Analysis of Implicit Graduated Optimization with Optimal Noise Scheduling"
_TMLR — Rejected by TMLR_

### Review · Reviewer_9LjZ · 2024-04-03

**Summary Of Contributions:**

The manuscript studies the problem of training neural networks with graduated optimization. The authors generalize the notion of $\sigma$-nice functions by Hazan et al. (2016) and suggest approximating stochastic gradient noise as a uniform distribution over the ball. Then, this work proposes running SGD over a smoothed function with a gradually decreased smoothing level to find a global optimum supported by convergence analysis. The suggested framework is used to justify better learning rates and mini-batch size schedules, and it has also been validated via experiments on several computer vision problems and models.

**Audience:**

Yes

**Broader Impact Concerns:**

No concerns about this submission.

**Claims And Evidence:**

No

**Requested Changes:**

My concerns are listed in the Weaknesses part of the review. Next, I give a short overview of my major recommendations.

1. Revision of the related works section described in Weakness 1 by putting the findings into a modern context.

2. Adjustment of some claims based on Assumption 2.1.

**Strengths And Weaknesses:**

## Strengths
- The idea appears novel and not very well explored. The considered problem is important and deserves attention. The proposed approach may be of interest to the broader machine learning and optimization community from both theoretical and practical perspectives.
- Theoretical results are generally accurately stated and seem to be correct.
- The paper is well organized. The writing is detailed and mostly clear.
- Experiments include several practical problems of different scales with various models. The authors also include an implementation hosted via an anonymous repository.

## Weaknesses
1. __Background__.
While the literature overview describes many papers on Graduated optimization and various applications of this technique, it seems to completely miss the relevant works on smoothing (especially in optimization [1]) and analysis of SGD. This makes it harder to judge the theoretical contributions as they are not put into the proper context of optimization literature. Moreover, it creates an impression that the authors are unfamiliar with some classical results, such as SGD convergence [3, 4]. The choice of references on optimization methods in the second sentence of the 1.1 Background subsection is questionable.
     - It is unclear why the authors present convergence theory for SGD on smooth and strongly convex functions instead of reusing state-of-the-art results. The suggested analysis appears too long and suboptimal (e.g., constant $B_2$ can be arbitrarily large). That is why I recommend referring to Section 5.5 (Bibliographic notes) of a recent manuscript [2] for a historical overview and more recent techniques.
     - In addition, please compare the obtained theoretical and experimental results to a recent work [5] that performed extensive experiments on learning rate scheduling.

2. __Theoretical results__.
- I would like the authors to include more details about the optimization properties of the smoothed function (such as smoothness, Lipschitzness, and convexity).

 - I believe a more detailed discussion about Definition 2.2 is needed as it is not very common in optimization literature. For instance, how restrictive is it, and what is the intuition behind it?
 - Lower bound for $a_m > \sqrt{2}$ (in Proposition 3.1) seems strange as it means that smoothed function $\hat{f_{\delta_m}}$ can be strongly convex on an arbitrarily large ball around $x^\star$. At the same time, the "strong convexity area" of the same function is restricted to the ball of radius $d_m |\delta_m|$. Could you please clarify this point?
    - In addition, $d_m$ is inversely proportional to $|\delta^-|$ which can be equal to zero when $x = x^\star$.
- Theorem 3.2 needs discussion about its implications, proof technique, and innovations upon previous analysis.

- Assumption 2.1 (ii) is pretty restrictive as it does not hold even for subsampling of quadratic functions over an unbounded domain. While it is suitable for other situations, in the case of subsampling, it is less nuanced than existing approaches [6]. I find this issue significant as mini-batching is one of the focuses of this work.
    - Moreover, Assumption 2.1 (ii) serves as the basis for the central insights in Subsection 3.3. It is unclear how the transition from a general stochastic gradient error vector to a uniform distribution over a ball happens. If it is an approximation, it needs to be stated explicitly. Because of this issue, the paper makes the impression of suggesting a theory artificially tailored to explain some practical effects rather than developing a framework rigorously from first principles. One of the main statements
     > We proved that SGD with a mini-batch stochastic gradient has the effect of smoothing the function, and the degree of smoothing is greater with larger learning rates and smaller batch sizes.

    appears to be not supported enough. However, I am happy to change my opinion in case of misunderstanding.

3. __Writing__.
Notation is heavy in some places due to the many constants introduced. I had a hard time understanding some of the technical details. It would be nice if it could be simplified for the main part of the manuscript. In some cases, the paper seems to create confusion between optimization complexity and generalization performance (e.g., in subsection 1.3.2).

4. __Experimental results__.
    - What does "the number of parameter updates" mean on the horizontal axis of the plots? Do the figures take into account the fact that the per-iteration cost of methods with variable batch size changes?
    - There are too many curves in Figure 10, which makes it hard to distinguish the results.

### Minor
- I would like to ask the authors to comment on the parameters required to run the proposed Graduate optimization.
- I am also curious about how the constant $M$ is selected.

- > This noise is worthless in convex optimization

This claim is too bold or simply inaccurate.

___
[1] Duchi, John C., Peter L. Bartlett, and Martin J. Wainwright. "Randomized smoothing for stochastic optimization." SIAM Journal on Optimization 22.2 (2012): 674-701.

[2] Garrigos, Guillaume, and Robert M. Gower. "Handbook of convergence theorems for (stochastic) gradient methods." arXiv preprint arXiv:2301.11235 (2023).

[3] Moulines, Eric, and Francis Bach. "Non-asymptotic analysis of stochastic approximation algorithms for machine learning." Advances in neural information processing systems 24 (2011).

[4] Needell, Deanna, Rachel Ward, and Nati Srebro. "Stochastic gradient descent, weighted sampling, and the randomized Kaczmarz algorithm." Advances in neural information processing systems 27 (2014).


[5] Defazio, Aaron, et al. "When, Why and How Much? Adaptive Learning Rate Scheduling by Refinement." arXiv preprint arXiv:2310.07831 (2023).


[6] Gower, Robert Mansel, et al. "SGD: General analysis and improved rates." International conference on machine learning. PMLR, 2019

---

> ### Author Response · Authors · 2024-04-18
>
> **Comment 1:** While the literature overview describes many papers on Graduated optimization and various applications of this technique, it seems to completely miss the relevant works on smoothing (especially in optimization [1]) and analysis of SGD. This makes it harder to judge the theoretical contributions as they are not put into the proper context of optimization literature. Moreover, it creates an impression that the authors are unfamiliar with some classical results, such as SGD convergence [3, 4]. The choice of references on optimization methods in the second sentence of the 1.1 Background subsection is questionable.
>
> **Reply:** The literature [3, 4] is cited in the revised manuscript. Thanks for the helpful advice.
>
> **Comment 2:** It is unclear why the authors present convergence theory for SGD on smooth and strongly convex functions instead of reusing state-of-the-art results. The suggested analysis appears too long and suboptimal (e.g., constant can be arbitrarily large).
>
> **Reply:** We had to prepare our own proofs to match the proof techniques of the $\sigma$-nice paper [Hazan2016]. The evaluation metrics are different in our analysis from those in the literature [2] and elsewhere.
>
> **Comment 3:** In addition, I would like to ask the authors to compare their results to a recent work [5] which performed a very wide range of experiments on learning rate scheduling.
>
> **Reply:** The linear decay learning rate schedule, which is considered optimal by [5], is defined as follows:
> \begin{align*}
> \eta_t := \frac{C}{\sqrt{T}}\left( 1 - \frac{t}{T} \right)
> \end{align*}
> where $t \in [T]$, $T$ is the number of steps, and $C$ is an unknown constant that is not known in advance. Then, the learning rate is updated **per step**. We can implement SGD using this method. We trained ResNet18 on CIFAR100 and compared it with our results, for example with $C=\lbrace 1, 10, 100 \rbrace$ and $T = 78200$ which is the number of steps for 200 epochs with a batch size of 128. Some of the experimental results are shown at the following URL: \url{https://anonymous.4open.science/r/new-sigma-nice}.
>
> However, all of the schedulers in our paper decrease the learning rate **per epoch**. For example, a polynomial decay is defined as
> \begin{align*}
> \eta_t := 0.1 \cdot \left( 1 - \frac{t}{T} \right)^{0.9}
> \end{align*}
> where $t \in [T]$ and $T$ is the number of epochs. Therefore, it may not be worthwhile to compare the decay rates of the scheduler considered in our paper with the linear decay learning rate scheduler in the literature [5], as shown in Figures 4-6. For the same reason, we also exclude the often-theoretically used $\frac{1}{t}$ and $\frac{1}{\sqrt{t}}$ schedulers.
>
> **Comment 4:** I would like to authors to include more details about the optimization properties of the smoothed function (such as smoothness, Lipschitzness, and convexity).
>
> **Reply:** Thank you for your suggestion. To address this point, we have moved the discussion on smoothed functions from Appendix C to Section 2.2 of the main body in the revised manuscript.
>
> **Comment 5:** I believe a more detailed discussion about Definition 2.2 is needed For instance, how restrictive it is and what is the intuition behind it as it is not that common for optimization literature.
>
> **Reply:** A few explanations have been added near Definition 2.2 in the revised manuscript. Thank you for your suggestion.
>
> **Comment 6:** Lower bound for $a_m > \sqrt{2}$ (from Proposition 3.1) seems strange as it means that smoothed function $\hat{f}_{\delta_m}$ can be strongly convex on an arbitrary large ball around $x^\star$.
>
> **Reply:** Proposition 3.1 states that as long as $|\delta_m|$ satisfies the condition $|\delta_m|=|\delta^-|$, we can take a large $a_m$. However, when $a_m$ is large, the upper bound of the probability $p(a_m)$ of the existence of $\delta^-$ approaches 0, so $p(a_m)$ also approaches 0. Therefore, Proposition 3.1 is always true, but as you say, for infinitely large $a_m$, it is highly unlikely that $|\delta_m|=|\delta^-|$, so it may not hold that the function $\hat{f}_{\delta_m}$ is $\sigma$-strongly convex on $N(\boldsymbol{x}^\star; a_m r)$. We have put this discussion of probability $p(a_m)$ near the definition formula for $p(a_m)$ in Section 3.1 in the revised manuscript.
>
> **Comment 7:** At the same time “strong convexity area” of the same function is restricted to the ball of radius $d_m |\delta_m|$. Could you please clarify this point?
>
> **Reply:** We define $d_m$ by $d_m := a_m r/ |\delta^-|$. The function $\hat{f}_{\delta_m}$ is also strongly convex in $N(\boldsymbol{x}^\star; d_m |\delta_m|)$ if it is strongly convex in $N(\boldsymbol{x}^\star; a_m r)$. We clearly state this point in the revised manuscript.

---

> ### Author Response · Authors · 2024-04-18
>
> **Comment 8:** In addition, $d_m$ is inversely proportional to $|\delta|^-$ which can be equal to zero when $x=x^\star$.
>
> **Reply:** As you say, $|\delta^-|$ goes to 0 when $\boldsymbol{x}=\boldsymbol{x}^\star$, which is not what we intended. We have modified the definition of $\boldsymbol{x}$ in Proposition 3.1 from $\boldsymbol{x} \in N(\boldsymbol{x}^\star; a_m r)$ to $\boldsymbol{x} \in N(\boldsymbol{x}^\star; a_m r) \setminus \lbrace\boldsymbol{x}^\star \rbrace$. Thank you very much for your useful remarks.
>
> **Comment 9:** Theorem 3.2 needs discussion about the implications, proof technique, and innovations upon previous analysis.
>
> **Reply:** Our Theorems 3.2 and 3.4 were proved using the proof techniques of Theorem 5.1 of the previous result [Hazan et al. 2016]. Hazan et al.'s theorem was analyzed for $\sigma$-nice functions, whereas our theorem was analyzed for new $\sigma$-nice functions.
>
> Theorem 3.2 shows that convergence is faster when the power of the polynomial decay $p$ is large, and when $p=1$, it takes at least $\mathcal{O}\left( 1/\epsilon^3 \right)$ iterations. However, explicit graduated optimization, such as with our Algorithm 1 or Hazan et al.'s Algorithm 1, is not applicable to DNNs due to the impossibility of computing a smoothed function $\hat{f}_{\delta_m}$. On the other hand, implicit graduated optimization, such as with our Algorithm 2, can be applied to DNNs. This point was added to the revised manuscript after Theorems 3.2 and 3.4.
>
> **Comment 10:** Assumption 2.1 (ii) is pretty restrictive as it does not hold even for subsampling of quadratic functions over unbounded domain. While it is suitable for other situations, in case of subsampling it is less nuanced then existing approaches[6]. Due to the fact that one of the focus of this work is on the aspect of mini-batching I find this issue significant.
>
> **Reply:** When the function $f$ is unbounded, there is not even an optimal value $f(\boldsymbol{x}^\star)$, so indeed, Assumption A3(ii) does not hold for all $\boldsymbol{x} \in \mathbb{R}^d$, as you say. However, since A3(ii) is assumed for the SGD trajectory $(\boldsymbol{x}_t)$, there is no problem in that case either. This assumption also holds for mini-batching. See the following theorem.
>
> **Theorem 0.1**
> Let $f(\boldsymbol{x}) := \frac{1}{n} \sum_{i=1}^n f_i(\boldsymbol{x})$, where the function $f_i \colon \mathbb{R}^d \to \mathbb{R}$ is continuously differentiable and $L_i$-smooth, and $\xi$ be a random variable following a discrete uniform distribution. Then, the following holds for all $\boldsymbol{x} \in \mathbb{R}^d$:
>
> \begin{align*}
> \mathbb{E}\_{\xi} \left[ \Vert \nabla f\_{\xi}(\boldsymbol{x}) - \nabla f(\boldsymbol{x}) \Vert^2\_2 \right] \leq \frac{2}{n} \sum_{i=1}^{n} L_i M_i,
> \end{align*}
> where $M_i := \sup \lbrace f_i(\boldsymbol{x}) - f_i^\star \colon \boldsymbol{x} \in \mathbb{R}^d \rbrace \in [0, +\infty)$.
>
>
> **Proof of Theorem 0.1**
>
> From Assumption (A3)(i), we have
> \begin{align*}
> \mathbb{E}\_\xi \left[ \nabla f\_{\xi} (\boldsymbol{x}) \right] = \frac{1}{n} \sum_{i=1}^{n} \nabla f_i(\boldsymbol{x}) = \nabla \left( \frac{1}{n} \sum\_{i=1}^{n} f\_i \right)(\boldsymbol{x}) = \nabla f(\boldsymbol{x}).
> \end{align*}
> Let $\boldsymbol{\bar{x}} := \boldsymbol{x} - \frac{1}{L\_i} \nabla f\_i (\boldsymbol{x})$, from $L\_i$-smoothness of $f\_i$,
> \begin{align*}
> f\_i^\star
> &\leq f\_i (\bar{\boldsymbol{x}}) \leq f\_i(\boldsymbol{x}) + \langle \nabla f\_i(\boldsymbol{\boldsymbol{x}}), \underbrace{\boldsymbol{\bar{x}} - \boldsymbol{x}}\_{-\frac{1}{L_i} \nabla f_i(\boldsymbol{x})} \rangle + \frac{L_i}{2} \Vert \boldsymbol{\bar{\boldsymbol{x}}} - \boldsymbol{x} \Vert^2 \\\\
> &=f\_i(\boldsymbol{x}) - \frac{1}{L\_i}\Vert \nabla f\_i(\boldsymbol{x}) \Vert^2 + \frac{1}{2L\_i} \Vert \nabla f\_i(\boldsymbol{x}) \Vert^2 \\\\
> &=f_i(\boldsymbol{x}) - \frac{1}{2L_i} \Vert \nabla f_i(\boldsymbol{x}) \Vert^2.
> \end{align*}
> Therefore, for all $i \in [n]$ and all $\boldsymbol{x} \in \mathbb{R}^d$,
> \begin{align*}
> \Vert \nabla f_i(\boldsymbol{x}) \Vert^2 \leq 2L\_i (f\_i(\boldsymbol{x}) - f\_i^\star) \leq 2L\_iM\_i.
> \end{align*}
> Hence,
> \begin{align*}
> \mathbb{E}\_{\xi} \left[ \Vert \nabla f_{\xi}(\boldsymbol{x}) - \nabla f(\boldsymbol{x}) \Vert_2^2 \right]
> &=\mathbb{E}\_{\xi} \left[ \Vert \nabla f_\xi (\boldsymbol{x}) \Vert^2 \right] -2\mathbb{E}\_{\xi} \left[ \langle \nabla f_\xi(\boldsymbol{x}), \nabla f(\boldsymbol{x}) \rangle \right] + \mathbb{E}\_{\xi} \left[ \Vert \nabla f(\boldsymbol{x}) \Vert^2 \right] \\\\
> &= \mathbb{E}\_{\xi} \left[ \Vert \nabla f_{\xi} (\boldsymbol{x}) \Vert^2 \right] -2\Vert \nabla f(\boldsymbol{x}) \Vert^2 + \Vert \nabla f(\boldsymbol{x})\Vert^2 \\\\
> &=\frac{1}{n}\sum_{i=1}^{n} \Vert \nabla f\_i(\boldsymbol{x)} \Vert^2 - \Vert \nabla f(\boldsymbol{x}) \Vert^2 \\\\
> &\leq \frac{2}{n} \sum_{i=1}^{n} L\_iM\_i.
> \end{align*}
> This completes the proof.

---

> ### Author Response · Authors · 2024-04-18
>
> **Comment 11:** Moreover, Assumption 2.1 (ii) serves as the basis for of the main insights in Subsection 3.3. It is not clear for me how the transition from a general stochastic gradient error vector to a uniform distribution over a ball happens? If it is an approximation, it has to be stated explicitly. Because of this issue the paper creates an impression of suggesting a suitable theory for explaining some practical effects rather than proposing a framework rigorously from first principles.
>
> **Reply:** We have reconsidered the derivation of $\boldsymbol{\omega}_t$ in light of the reviewer's suggestion. We assume that the stochastic noise follows a normal distribution, based on [Zhang et al., 2020]. We can improve our results by using the fact that the standard normal distribution in high dimensions $d$ is close to a uniform distribution on a sphere of radius $\sqrt{d}$. The changes resulting from this amendment are minor. We have added this explanation to Section 3.3. Thank you for your careful reading.
>
> **Comment 12:** Writing. Notation is heavy and I found a hard time understanding the technical details. It would be nice if it can be simplified for the main part of the paper.
>
> **Reply:** Thank you for your comments. To address the reviewers' concerns, we have clarified the definitions and added explanations as appropriate, particularly in Section 3.1. Please check them out.
>
> **Comment 13:** In some cases paper seems to create a confusion between optimization complexity and generalization performance (such as in subsection 1.3.2).
>
> **Reply:** Our argument is based on the somewhat non-theoretical finding that flat local solutions around which the optimizer converges have better generalizability than sharp local solutions. Since the function optimized by the optimizer is constructed from a limited training sample, there should be some deviation from the function constructed with unknown data, including the test data. Therefore, the intuitive explanation is that the flatness around the local solution prevents the deviation from degrading the generalizability. As you said, it is not good for readability to keep this implicit, so we have added this to Section 3.3.1 of the revised manuscript. Thank you very much for your helpful remarks.
>
> **Comment 14:** What does “the number of parameter updates” means on the horizontal axis of the plots?
> Do the figures take into account the fact that the per-iteration cost of methods with variable batch size changes?
>
> **Reply:** "the number of parameter updates" means the number of steps, which is $T$ in the optimization step:
> \begin{align*}
> \boldsymbol{x}_{t+1} := \boldsymbol{x}_t + \eta \boldsymbol{d}_t \ (t \in [T]).
> \end{align*}
> Since the number of steps per epoch differs greatly for different batch sizes, it would be unfair for large batch sizes if only graphs with epochs on the horizontal axis are shown. Therefore, we tried to avoid this inequity by including a graph with a step on the horizontal axis.
>
> Several reviewers pointed this out, so we prepared a new graph with gradient query on the horizontal axis. Please see Appendix E for the graphs with gradient queries. Since the gradient query is defined by the number of steps $T$ and the batch size $b$, it takes into account the computational complexity per step and is less likely to be unfair across batch sizes. Thank you very much for your useful suggestions.
>
> **Comment 15:** There are too many curves in Figure 10 which makes it hard to distinguish the results.
>
> **Reply:** Thank you for your suggestion. In the revised manuscript, we have tried to address this issue by enlarging the figure.
>
> **Reference**
>
> [Hazan et al., 2016] Hazan, E., Yehuda, K., and Shalev-Shwartz, S. On graduated optimization for stochastic non-convex problems. ICML2016.
>
> [Zhang et al., 2020] J. Zhang, S. P. Karimireddy, A. Veit, S. Kim, S. Kumar, and S. Sra. Why are adaptive methods good for attention models? NeurIPS2020.

---

> > ### Comment · Reviewer_9LjZ · 2024-04-25
> > **Official Response**
> >
> > I would like to thank the authors for provided answers and clarifications. I would like to comment on some of the responses.
> >
> > - > **Comment 2.** The evaluation metrics are different in our analysis from those in the literature [2] and elsewhere.
> >
> > It is unclear to me how the *"evaluation metrics"* in your analysis (Theorem 3.1) are fundamentally different from the standard works analyzing SGD for (strongly convex) smooth functions.
> >
> > - The link (https://anonymous.4open.science/r/new-sigma-nice%7D) provided in response to **Comment 3.** is currently invalid as it says that *"The repository is not found."*.
> >
> > - Regarding **Comment 10** and Assumption 2.1 (ii) discussion. You use the constant
> > $M_i:=\sup \\{f_i(\mathbf{x}) - f^{\star}_i: \mathbf{x} \in \mathbb{R}^d\\}$
> > to upper-bound the variance of the stochastic gradient estimator, which can be unbounded, e.g., for quadratic functions $a x^2$. Moreover, this quantity can be arbitrarily large even for trajectory ($\mathbf{x}_t$) of SGD-type methods.
> >
> > - > **Comment 11.**  We assume that the stochastic noise follows a normal distribution, based on [Zhang et al., 2020].
> >
> > I do not find the current discussion justified and nuanced enough because Zhang et al. (2020) consider a case of particular deep learning models and datasets. In addition, they show that distribution for other cases (e.g., BERT training) is not Gaussian.
> >
> > - > **Comment 15.** In the revised manuscript, we have tried to address this issue by enlarging the figure.
> >
> > While the revised version brings improvement, I would suggest removing curves for some of the values $p$ as a smaller quantity, which will be sufficient to demonstrate the main effect.

---

> ### Author Response · Authors · 2024-04-25
>
> Thank you for your reply.
>
> >The link provided in response to Comment 3. is currently invalid as it says that "The repository is not found.".
>
> First, it appears that the link was changed because of the braces. The correct link is https://anonymous.4open.science/r/new-sigma-nice.
> We are sorry for the inconvenience.
>
> >Regarding Comment 10 and Assumption 2.1 (ii) discussion. You use the constant $M_i:=\sup \{f_i(\mathbf{x}) - f^{\star}_i: \mathbf{x} \in \mathbb{R}^d\}$ to upper-bound the variance of the stochastic gradient estimator, which can be unbounded, e.g., for quadratic functions $ax^2$. Moreover, this quantity can be arbitrarily large even for trajectory ($\boldsymbol{x}_t$) of SGD-type methods.
>
> See, for example, [N.Sato and H. Iiduka, 2024, Section 3.3]. This study estimates the variance of the stochastic gradient from an experiment and shows that $C^2$ is a finite value. For example, the variance of the stochastic gradient of SGD for training ResNet18 on the CIFAR100 dataset can be obtained as $C^2 < 1280$. This is precisely an example where Assumption 2.1 (ii) holds.
>
> >I do not find the current discussion justified and nuanced enough because Zhang et al. (2020) consider a case of particular deep learning models and datasets. In addition, they show that distribution for other cases (e.g., BERT training) is not Gaussian.
>
> You are right, we should have explained that sometimes stochastic noise does not follow a normal distribution.
> Section 3.3 has been appropriately corrected. Thank you for pointing this out.
>
> >While the revised version brings improvement, I would suggest removing curves for some of the values $p$ as a smaller quantity, which will be sufficient to demonstrate the main effect.
>
> In the revised manuscript, we reduced the number of graphs and moved the original results to Appendix F.
>
> >It is unclear to me how the "evaluation metrics" in your analysis (Theorem 3.1) are fundamentally different from the standard works analyzing SGD for (strongly convex) smooth functions.
>
> During the process of revising the manuscript, we modified the evaluation metric in Theorem 3.1 as $\min_{t \in [T]} (F(x_t) - F(x^\star))$ that is the same as in [Theorem 3.9, 2]. Hence, the metric used in the paper does not differ from the ones in the standard works analyzing SGD.
>
> **Reference**
>
> [N.Sato and H. Iiduka, 2024] Naoki Sato, Hideaki Iiduka, Role of Momentum in Smoothing Objective Function in Implicit Graduated Optimization, https://arxiv.org/abs/2402.02325, 2024.
>
> [2] Garrigos, Guillaume, and Robert M. Gower. "Handbook of convergence theorems for (stochastic) gradient methods." arXiv preprint arXiv:2301.11235 (2023).

---

### Review · Reviewer_9HcQ · 2024-04-05

**Summary Of Contributions:**

This submission connects SGD to graduated optimization, a global optimization
technique that works by iteratively minimizing smoothed versions of the
objective function. The authors argue that the noise in SGD smooths the
objective function and increasing the batch-size or decreasing the step-size
reduces the degree of smoothing, allowing for control of the graduated
optimizer. They establish a class of functions called "new $\sigma$-nice" for
which they prove graduated optimization with SGD has global convergence to a
minimizer of the objective. The authors then show that a polynomial decay
schedule for the step-size maximizes the rate of this global convergence
compared to other widely-used schedules. The paper concludes with experiments
on CIFAR-100 and ImageNet.

**Audience:**

Yes

**Broader Impact Concerns:**

None.

**Claims And Evidence:**

No

**Requested Changes:**

- The most important change is to fix the use of local strong convexity
    in Theorem 3.2 and 3.4. I'm not sure how the authors can argue that the
    iterates remain in the region where strong convexity holds almost
    surely, but something like this needs to hold to fix their proof.

- The experimental should be repeated for several random restarts to account
    for stochasticity.

- I strongly suggest improving the writing in Section 3.1. In particular, the
    large number of unexplained constants in Def. 3.1 and Prop. 3.1 (e.g.
    $d_{m}$, $\gamma_m$, $u_m$, $a_m$, $\delta^{-}$, etc.) should be addressed.

**Strengths And Weaknesses:**

This paper attempts to answer a widely studied question in machine learning:
why does the noise in SGD seem help find "good" solutions when training
non-convex models?
The authors use the graduated optimization framework to argue that SGD is
actually performing global optimization.
While I appreciate this fresh viewpoint, the paper is held back by the confused
writing and several important theoretical issues.
As it is, I cannot endorse accepting this work.
To summarize:

**Strengths**:

- Graduated optimization provides a fresh and interesting perspective on
    SGD in the non-convex setting.

- Experiments are provided for large-scale problems, including ImageNet.


**Weaknesses**:

- The global convergence theorems for graduated optimization with SGD
    implicitly require that the SGD iterates remain in a bounded neighborhood
    of the minimizer. Since this may not hold, I am skeptical that Theorems 3.2
    and 3.4 are correct.

- The writing is very difficult to understand. Moreover, since the various
    definitions and theorems are interconnected, it is difficult to track down
    the meanings and values of different variables.

- The argument that SGD smooths the objective is not rigorously developed and
    distributional assumptions on the gradient noise (see Eq. 8) are not
    justified.

- The experiments are given only for a single restart and plot epochs/iterations
    rather than the total number of gradient queries, leading to a biased
    presentation.

## Additional Details

Theorem 3.2:
- I believe the proof of this result has a serious issue.
    The proof of Theorem 3.1 requires strong convexity of $f$ to hold between
    between $\hat x_t$ and the minimizer $x^*$ for every $t$ (see Lemma D.1).
    This is true if $f$ is strongly-convex globally, such as assumed in Theorem
    3.1.  However, Theorem 3.2 uses Theorem 3.1 to control sub-optimality when
    minimizing $\hat f_{\delta_M}$, which is only strongly convex over a
    neighbourhood according to the definition of a new $\sigma$-nice function.
    Since there is no guarantee that that the iterates of SGD remain in this
    neighbourhood, it is not possible to apply Theorem 3.1 to minimization of
    $\hat f_{\delta_M}$.

- I am also very skeptical that you can guarantee the iterates
    generated by SGD remain in $\mathcal{N}(x^*, d_m \delta_m)$ with
    probability $1$. For general finite-sum functions, it is straightforward to
    argue that there is a sequence of stochastic gradients for which the
    iterates must exit any neighbourhood of the minimizer.

Equation 1:
- It took me a bit of effort to see this equivalence. Perhaps you should
    mention that $y_{t+1}$ is a GD step from $x_{t+1}$, so that it is a
    a random variable that depends on $w_t$.

- It does not immediately follow that
    $\mathbb{E}[\nabla f(y_t - \eta \omega_t)] = \nabla \mathbb{E}[f(y_t -
    \eta \omega_t)]$.
    Additional work is required to exchange the order of expectation and
    differentiation since this requires exchanging integrals and limits.
    As a result, I don't think your argument that SGD is smoothing $f$
    is fully rigorous.

Section 3.3:
- How do you argue that the gradient noise $\omega_t = \frac{C}{\sqrt{b}}
    u_t$, where $u_t$ is uniformly distributed on the norm ball? I see no
    reason why the gradient noise should be distributed so nicely,
    especially when $f$ is a finite sum function.

- The appearance of $u_m$ in Prop. 3.1 makes it seem like this distributional
    assumption on the noise is being used here as well? This needs to be
    clearly explained in the text.

Experiments:
- It's critical to average results for stochastic optimizers over
    multiple random restarts. Convergence guarantees for SGD and related
    methods are typically in-expectation and results for a single run may be
    misleading.  I suggest running $5$ trials and plotting both the median and
    interquartile range.

- I think it would be more fair to plot the total number of gradient queries
    rather than epochs or iterations. Plotting epochs biases the results
    towards small batch-sizes, while plotting iterations biases the results
    towards large batch-sizes.

- It seems like increasing the batch-size generally outperforms the hybrid
    approach in terms of total number of iterations on three out of four
    problems. Why do you claim that the hybrid approach is superior?

### Minor Issues:

Page 1:
- "and its variants, such as Adam kingma & Ba (2015)" --- Use `\citep` here, rather than
        `citet`. Also, "kingma" should be capitalized since it's a proper noun.


Page 2:
- "Equation (1) indicates that SGD smooths the function Kleinberg et al. (2018)" ---
        again, you want `citep` here instead of `\citet`.

Table 1:
- This is a very awkward way to introduce the notation for the paper.
        Since notation is not introduced when first used, I am forced to
        scroll back through the paper to reference this table whenever new
        notation is used.

Definition 3.1:
- Calling the condition "new $\sigma$-nice" is quite confusing
        For example, in Proposition 3.2 when you say "...sufficient condition for
        $f$ to be a new $\sigma$-nice function", it's not clear if $f$ is a
        new function which is $\sigma$-nice or if it is a new $\sigma$-nice function.
        I suggest coming up with a better name.

- In part (ii), do you require $\hat f_{\delta_m}$ to be strongly convex
        on $N(x^*, d_m \delta_m)$ or on $N(x^*_{\delta_m}, d_m \delta_m)$?
        It seems quite strong to require the smoothed function to be strongly
        convex on a neighbourhood of $x^*$ instead of the smoothed minimizer
        $x_{\delta_m}^8$ as required by the original $\sigma$-nice definition.

Proposition 3.1:
- It looks like $\delta^{-}$ is defined as a function of $x \in N(x^*, a_m r)$.
        Is $\delta^{-}$ supposed to be the supremum over this set? Otherwise
        I cannot see how it is a fixed value otherwise. The same goes for
        $u_m \sim B(0,1)$.

- What does it mean to say "if $d_m := a_m r / |\delta^{-}|$ holds"? This
        is the definition of $d_m$, so the expression always holds.

- What is the definition of $a_m$? This isn't written anywhere.

Proposition 3.2:
- Isn't the definition of new $\sigma$-nice supposed to hold
        for any choice of noise level $\delta_m \in \mathbb{R}$?
        That is, the definition of new $\sigma$-nice is given independently of the
        smoothing regime --- it's only a condition on $f$ --- so now it is strange
        to prove $f$ is new $\sigma$-nice for some specific noise levels.

- It's not at all clear what functions/noise levels satisfy Eq. (4).

Theorem 3.1:
- This is a standard result in the optimization literature and should not
        be presented as if new. See Rakhlin et al. [1] and the references
        therein.

All Figures:
- The font-sizes and line-widths are too small to be legible when
    printed. As a rule, the font-size of all figure text should be at least as
    large as the font-size of the text in the main paper.


### References:

[1] Rakhlin, Alexander, Ohad Shamir, and Karthik Sridharan. "Making gradient descent optimal for strongly convex stochastic optimization." Proceedings of the 29th International Coference on International Conference on Machine Learning. 2012.

---

> ### Author Response · Authors · 2024-04-18
>
> **Reply to comment 1 on weakness about Theorem 3.2:** Assuming that the function $f$ is a new $\sigma$-nice function, from condition (i) of Definition 3.1, $\boldsymbol{x}^\star_{\delta_m} \in N(\boldsymbol{x}^\star; d_m |\delta_m|)$ holds. This is shown in Proposition 3.3, as follows:
> \begin{align*}
> \Vert \boldsymbol{x}^\star_{\delta_m} - \boldsymbol{x}^\star \Vert
> &= \Vert \boldsymbol{x}^\star_{\delta_m} - \boldsymbol{x}^\star_{\delta_{m+1}} + \boldsymbol{x}^\star_{\delta_{m+1}} - \boldsymbol{x}^\star \Vert \\\\
> &\leq \Vert \boldsymbol{x}^\star_{\delta_m} - \boldsymbol{x}^\star_{\delta_{m+1}} \Vert + \Vert \boldsymbol{x}^\star_{\delta_{m+1}} - \boldsymbol{x}^\star \Vert \\\\
> &\leq \Vert \boldsymbol{x}^\star_{\delta_m} - \boldsymbol{x}^\star_{\delta_{m+1}} \Vert + \Vert \boldsymbol{x}^\star_{\delta_{m+1}} - \boldsymbol{x}^\star_{\delta_{m+2}} \Vert + \cdots + \Vert \boldsymbol{x}^\star_{\delta_M} - \boldsymbol{x}^\star_{\delta_{M+1}} \Vert + \Vert \boldsymbol{x}^\star_{\delta_{M+1}} - \boldsymbol{x}^\star \Vert \\\\
> &\leq (|\delta_m| - |\delta_{m+1}|) + (|\delta_{m+1}| - |\delta_{m+2}|) + \cdots + (|\delta_{M}| - |\delta_{M+1}|) + 0 \\\\
> &= |\delta_m| \\\\
> &< d_m |\delta_m|,
> \end{align*}
> where we have used $d_m > 1$. Therefore, the SGD or GD sequences never leave the strongly convex region because the optimal solution $\boldsymbol{x}^\star_{\delta_m}$ of the smoothed function $\hat f_{\delta_m} (m \in [M])$ is contained in the strongly convex region $N(\boldsymbol{x}^\star; d_m |\delta_m|)$.
>
> Now, in Proposition 3.2, which presents a sufficient condition for the function $f$ to be a new $\sigma$-nice function, we assumed $\boldsymbol{x}^\star_{\delta_m} \in N(\boldsymbol{x}^\star; d_m |\delta_m|)$. This may seem somewhat artificial, so we have revised Proposition 3.2 by assuming $\boldsymbol{x}^\star_{\delta_{m-1}} \in N(\boldsymbol{x}^\star; d_m |\delta_m|)$ instead. See the revised manuscript for details. This change has no effect on the paper as a whole. The theory is now more solid, thanks to your suggestions. Thank you very much.
>
> **Comment 2:** It took me a bit of effort to see this equivalence. Perhaps you should mention that $\boldsymbol{y}\_{t+1}$ is a GD step from $\boldsymbol{x}\_{t+1}$, so that it is a random variable that depends on $\boldsymbol{\omega}_t$.
>
> **Reply:** As you say, $\boldsymbol{y}_t$ depends on the random variable $\boldsymbol{\xi}\_{t-1}$, so we needed an expectation value for $\boldsymbol{y}_t$ in equation (1) as well. We have corrected equation (1) and everything related to it. Thank you for your suggestion.
>
> **Comment 3:** It does not immediately follow that $\mathbb{E}[\nabla f(y_t - \eta \omega_t)] = \nabla \mathbb{E}[f(y_t - \eta \omega_t)]$. Additional work is required to exchange the order of expectation and differentiation since this requires exchanging integrals and limits. As a result, I don't think your argument that SGD is smoothing $f$ is fully rigorous.
>
> **Reply:** According to the literature [Theorem 7.49; Shapiro et al., 2009], if the function $f$ is Lipschitz continuous and differentiable, it is possible to perform an operation that exchanges the derivative with the expectation value; we have added this explanation to the manuscript.
>
> **Comment 4:** How do you argue that the gradient noise $\boldsymbol{\omega}_t = \left( C/\sqrt{b} \right) \boldsymbol{u}_t$, where $\boldsymbol{u}_t$ is uniformly distributed on the norm ball? I see no reason why the gradient noise should be distributed so nicely, especially when $f$ is a finite sum function.
>
> **Reply:** We have reconsidered the derivation of $\boldsymbol{\omega}_t$ in light of the reviewer's suggestion. We assume that the stochastic noise follows a normal distribution, based on [Zhang et al., 2020]. We can improve our results by using the fact that the standard normal distribution in high dimensions $d$ is close to a uniform distribution on a sphere of radius $\sqrt{d}$. The changes resulting from this amendment are minor. We have added this explanation to Section 3.3. Thank you for your careful reading.
>
> **Reference**
>
> [Shapiro et al., 2009] A. Shapiro, D. Dentcheva, and A. Ruszczynski. Lectures on Stochastic Programming - Modeling and Theory, MOS-SIAM Series on Optimization, 2009.
>
> [Zhang et al., 2020] J. Zhang, S. P. Karimireddy, A. Veit, S. Kim, S. Kumar, and S. Sra. Why are adaptive methods good for attention models? NeurIPS2020.

---

> > ### Comment · Reviewer_9HcQ · 2024-04-18
> >
> > > "Therefore, the SGD or GD sequences never leave the strongly convex region..."
> >
> > I don't see what this has to do with the location of the minimizer of the smoothed function. Theorem 3.1 requires strong convexity at every iterate, not in a local region of the minimizer. Because you are running SGD, there is no guarantee that the iterates remain in the strongly convex region on which Theorem 3.1 can be used. That is, the sequence of stochastic gradients can allow the iterate sequence to escape the strongly-convex region with non-zero probability, at which point Theorem 3.1 cannot be used. This breaks Theorem 3.2 as far as I can tell.
> >
> > Since SGD is not a descent method in $f$ or in distances $\|x - x_{\delta}^*\|_2$, you must prove prove that the iterates remain in the strongly-convex region almost surely in order to use Theorem 3.1. Can you please direct me to this result in the manuscript?

---

> > > ### Author Response · Authors · 2024-04-19
> > >
> > > We have now submitted the revised manuscript again.
> > >
> > > As you say, in a finite number of iterations, it was possible for the SGD’s sequence to fall outside the strongly convex region, probabilistically.
> > > To address this problem, Algorithm 2 was changed from SGD to GD, and Theorem 3.1 and others were modified accordingly.
> > > Since these modifications and Proposition 3.3, sequences no longer leave the strongly convex region.
> > >
> > > In any case, explicit graduated optimization using Algorithm 1 and 2 is not feasible due to computational complexity issues, so our paper recommends implicit graduated optimization using Algorithm 3 and 4.
> > >
> > > Thank you for your careful reading and valuable comments.

---

> > > > ### Comment · Reviewer_9HcQ · 2024-04-25
> > > >
> > > > Doesn't it undermine the main idea of the paper to use deterministic gradient methods here? If I understood properly, the point is that the noise from SGD smoothes $f$ simultaneously with optimization. This smoothing means that SGD is, in effect, being run on a locally strong convex function and lets you get stronger guarantees. Or did I misunderstand and the smoothing was always coming from a secondary source of noise, i.e. from directly convolving $f$ with a random variable?
> > > >
> > > > In other words, where does the smoothing come from in Algorithm 3 now that GD is the main optimization routine? Are you just convolving $f$ is noise to obtain a smoothed version?

---

> ### Author Response · Authors · 2024-04-18
>
> **Comment 5:** The appearance of $\boldsymbol{u}_m$ in Prop. 3.1 makes it seem like this distributional assumption on the noise is being used here as well? This needs to be clearly explained in the text.
>
> **Reply:** For a general smoothing as in Definition 2.1, the distribution followed by the random variable $\boldsymbol{u}$ need not necessarily be uniform; it can be a normal distribution. In fact, several previous studies [Wu, 1996; and Iwakiri et al., 2022] assumed that $\boldsymbol{u}$ follows a normal distribution. We assumed that it follows a uniform distribution because this is necessary for the analysis of the new $\sigma$-nice function. This is also true for the analysis of the $\sigma$-nice function [Hazan et al., 2016]. The random variable $\boldsymbol{u}_m$ appearing in Proposition 3.1 follows a uniform distribution because, in our paper, the random variable used to smooth the function follows a uniform distribution.
>
> As you say, these points were vague and difficult to understand. We have added appropriate explanations to Definition 2.1, Proposition 3.1, and other relevant parts in Sections 3.1 and 3.3. Thank you for your suggestion.
>
> **Comment 6:** It's critical to average results for stochastic optimizers over multiple random restarts. Convergence guarantees for SGD and related methods are typically in-expectation and results for a single run may be misleading. I suggest running 5 trials and plotting both the median and interquartile range.
>
> **Reply:** All of our results were averages of three runs. In response to your comment, we have prepared a graph plotting the maximum, average, and minimum values of the three runs.
>
> **Comment 7:** I think it would be more fair to plot the total number of gradient queries rather than epochs or iterations. Plotting epochs biases the results towards small batch-sizes, while plotting iterations biases the results towards large batch-sizes.
>
> **Reply:** We showed both graphs with epochs and iterations on the horizontal axis to eliminate inequities between batch sizes. After receiving reviews, we also plotted a graph with gradient queries as the horizontal axis. Please see Appendix E for the graphs with gradient queries.
>
> **Comment 8:** It seems like increasing the batch-size generally outperforms the hybrid approach in terms of total number of iterations on three out of four problems. Why do you claim that the hybrid approach is superior?
>
> **Reply:** Our Algorithm 3 includes three methods: a method to reduce the learning rate only, a method to increase the batch size only, and a hybrid method. We do not claim that the hybrid method is the best, but observed in our experiments that the two methods that increase batch size, including the hybrid method, are superior.
>
> **Reply to minor issues on citation notation:** Thank you for your careful reading. We have fixed it as you pointed out.
>
> **Minor Comment 2:** It looks like $\delta^-$ is defined as a function of $\boldsymbol{x} \in N(\boldsymbol{x}^\star; a_m r)$. Is $\delta^-$ supposed to be the supremum over this set? Otherwise I cannot see how it is a fixed value otherwise. The same goes for $\boldsymbol{u}_m \sim B(\boldsymbol{0}; 1)$.
>
> **Reply:** As you say, $\delta^-$ is a fixed value, so the notation was incorrect. The revised manuscript has been corrected. Thank you for your suggestion.
>
> **Minor Comment 3:** What does it mean to say "if $d_m := a_m r / |\delta^-|$ holds"? This is the definition of $d_m$, so the expression always holds. What is the definition of $a_m$? This isn't written anywhere.
>
> **Reply:** You are right, it needed to be revised. We have corrected it; please check the revised manuscript. Thank you for your suggestion.
>
> **Minor Comment 4:** Isn't the definition of new $\sigma$-nice supposed to hold for any choice of noise level $\delta_m \in \mathbb{R}$? That is, the definition of new $\sigma$-nice is given independently of the smoothing regime --- it's only a condition on $f$ --- so now it is strange to prove $f$ is new $\sigma$-nice for some specific noise levels.
>
> **Reply:** We apologize for any misunderstanding. The new $\sigma$-nice function is defined for all $\delta_m \in \mathbb{R}^d$ satisfying $\delta_{m+1} := \gamma_m \delta_m$. We have specified this in Definition 3.1.
>
> **Minor Comment 5:** This is a standard result in the optimization literature and should not be presented as if new. See Rakhlin et al. [1] and the references therein.
>
> **Reply:** Rakhlin et al. assume that the gradient norm is bounded, whereas we do not. Therefore, our results differ from those of Ralkin et al.

---

> ### Author Response · Authors · 2024-04-18
>
> **Reference**
>
> [Wu, 1996] Z. Wu. The effective energy transformation scheme as a special continuation approach to global optimization with application to molecular conformation. SIAM Journal on Optimization, 1996.
>
> [Iwakiri et al., 2022] H. Iwakiri, Y. Wang, S. Ito, and A. Takeda. Single loop gaussian homotopy method for non-convex optimization. NeurIPS2022.
>
> [Hazan et al., 2016] Hazan, E., Yehuda, K., and Shalev-Shwartz, S. On graduated optimization for stochastic non-convex problems. ICML2016.

---

> ### Author Response · Authors · 2024-04-25
>
> Thank you for your reply.
>
> Your following comment:
> >if I understood properly, the point is that the noise from SGD smoothes f simultaneously with optimization. This smoothing means that SGD is, in in effect, being run on a locally strong convex function and lets you get stronger guarantees.
>
> is absolutely right. The smoothing in Algorithm 3 is achieved **implicitly** by the stochastic noise in SGD.
>
> >Are you just convolving f is noise to obtain a smoothed version?
>
> No. Our proposed algorithms 3 and 4 are algorithms that can achieve graduated optimization while avoiding the **explicit** smoothing as Definition 2.1.
>
> Algorithms 1 and 2 represent **explicit** graduated optimization algorithms. Function smoothing is accomplished **explicitly** by convolving random variables as in Definition 2.1, and the smoothed strongly convex function is optimized by the gradient descent (Algorithm 2). However, in general, the integral operation of the function $f$ is not possible, so optimization by Algorithms 1 and 2 is not feasible.
>
> Algorithms 3 and 4 represent **implicit** graduated optimization algorithms. Function smoothing is accomplished **implicitly** by the stochastic noise in SGD.
> From Section 3.3, since the smoothed version of $f$ can be considered to be optimized by gradient descent, Algorithm 4 is a gradient descent.
> **In other words, SGD is actually performed in the experiment, but the theoretical analysis for the smoothed function needs to be done for GD.**
>
> If smoothing of the function $f$ by Definition 2.1 is possible and $\hat{f}_{\delta}$ is accessible, then Algorithm 2 can be GD, SGD, or even Adam, as long as there is no problem with the convergence guarantees you pointed out. This is **explicit** graduated optimization.  But Algorithm 4 must be gradient descent.
>
> >where does the smoothing come from in Algorithm 3 now that GD is the main optimization routine?
>
> $\hat{f}\_{\delta\_m}$ in Algorithm 3 has informations on the stochastic noise of SGD using the batch size $b_m$, a learning rate $\eta_m$, and the upper bound of the variance $C^2$ since $\delta_m$ is defined by $\delta_m = \eta_m C/\sqrt{b_m}$ (please see (8)).

---

> > ### Comment · Reviewer_9HcQ · 2024-04-25
> >
> > Thank you for the clarification.
> >
> > I am deeply concerned by the mismatch theory between theory and practice. As I said in my previous comment, I think it undermines the main idea of this paper to base the analysis on deterministic optimization with GD. Moreover, without the convergence guarantee for SGD on the un-smoothed function, I am skeptical about the central claim that SGD is the same graduated optimization. This simply isn't justified theoretically anymore.
> >
> > Rather than changing the paper to use GD, I suggest the authors consider ways to resolve the central convergence issue with SGD. Perhaps a more detailed analysis (or algorithmic modification) can guarantee that SGD remains within the strongly convex region with high probability. At least then you would be able to show a high-probability guarantee for global optimization, which would restore the central message of the paper.

---

> ### Author Response · Authors · 2024-04-27
>
> In Algorithms 1 and 2, we assume that the computation of $\hat{f}_{\delta_m}$ by Definition 2.1 and the access to its full gradient $\nabla \hat{f}\_{\delta_m}$ are possible. Hence, Algorithm 2 can be any optimizer as long as there is no problem with the convergence guarantees you pointed out. Therefore, Algorithm 2 can be GD.
>
> Algorithm 4 was not changed to GD to address your point. It has been GD since it was first posted. From (8), Algorithm 4 can be GD.
> In Algorithm 3, the function $\hat{f}\_{\delta\_m}$ that is optimized by GD has informations on the stochastic noise of SGD using the batch size $b_m$, a learning rate $\eta\_m$, and the upper bound of the variance $C^2$ since $\delta_m$ is defined by $\delta_m = \eta\_m C/\sqrt{b\_m}$.
> Thus, the purpose of the paper is not undermined.
>
> On the other hands, one might feel that there is a mismatch between theory and practice, since the experiment Section 4 uses SGD. In fact, if assuming $\hat{f}_{\frac{\eta C}{\sqrt{b}}} \approx \frac{1}{b}\sum\_{i=1}^b f\_{\xi\_i}$, then Algorithm 4 can be SGD.
>
>
> We can modify Algorithm 2 or 4 as the projected SGD to guarantee that it remains within the strongly convex region with high probability. In fact, the projected SGD generated by the sequence $(\hat{x}_t)$ with $\hat{x}\_{t+1} = P_m (\hat{x}\_t - \eta\_t \nabla F\_{S_t} (\hat{x}\_t))$, where $P\_m$ is the projection onto $B(\boldsymbol{x}^*; \gamma_m \delta_m)$, remains within  the strongly convex region $B(\boldsymbol{x}^*; \gamma_m \delta_m)$.
> Using the proof of Theorem 3.1 and the nonexpansivity of $P_m$ (i.e., $\Vert P\_m (\boldsymbol{x}) - P\_m (\boldsymbol{y}) \Vert \leq \Vert \boldsymbol{x} - \boldsymbol{y}\Vert$), we can show that the projected SGD satisfies
> \begin{align*}
> \min\_{t \in [T]} \mathbb{E} \left[ F(\hat{\boldsymbol{x}}\_t) - F(\boldsymbol{x}^\star) \right] \leq \frac{2 D\_2}{\sigma T},
> \end{align*}
> where $D\_1 := \sup\_{t \in \mathbb{N}} \mathbb{E} \left[\Vert \nabla F (\hat{\boldsymbol{x}}\_t)\Vert^2 \right]$ and $D_2 := C^2 + D_1$.
>
> These explanations have been added as Remarks near Theorems 3.1 and 3.3.

---

### Review · Reviewer_Kbek · 2024-04-12

**Summary Of Contributions:**

The paper attempts to offer insights and guidelines for training by presenting an analysis of graduated optimization combined with SGD. Specifically, it introduces a relaxed notion of a $\sigma$-nice function and demonstrates the convergence of the graduated optimization method with specific step sizes and noise scheduling. The paper recommends increasing the batch size and decreasing the learning rate.

**Audience:**

Yes

**Claims And Evidence:**

No

**Requested Changes:**

1. In some sections of the paper, it is unclear whether the discussion pertains to SGD alone or to graduated optimization combined with SGD. This should be clarified.

2. The clarity of the writing in the paper needs significant improvement. For instance, in Definition 3.1, it is ambiguous whether "$\left(\gamma_m \in\left(\frac{1}{d_{m+1}}, 1\right)\right)$" refers to "for all" $\gamma$ or "exists one". Also, the inputs for Algorithm 4 and Algorithm 2 differ; why is Algorithm 4 termed GD rather than SGD? The authors should verify the clarity of every theorem, proposition, and lemma.

3. I suggest that the authors also provide the total complexity of Algorithm 1, integrating Theorem 3.2 and Theorem 3.1, as well as for Algorithm 4.

**Strengths And Weaknesses:**

Strength:

The paper aims at the fundamental question of selecting the learning rate and batch size during training. Some discoveries of the paper coincide with practical observations.

Weaknesses:

1. The sufficient condition for this new $\sigma$-nice function is not clear to me. I didn't understand the role of random variable $u_m \sim B(0,1)$ in Proposition 3.1. The definition of $ |\delta^-|$ include $||u_m||$, x, and, the angle between $u_m$ and $x^* - x$ . Does $||u_m||$ denote the norm of the random variable or the norm of a realization of this random variable? If it is the norm of a random variable, which norm is applied here? If it is a realization, I found it hard to understand why being "\sigma"-nice would depend on such realization. Also, does different $x$ give different $|\delta^-|$ values?

2. In Section 3.3 Equation (8),  how does noise $\omega_t$ reduce to the uniform distribution over a ball $u_t$? $\omega_t$ is the noise due to minibatch, so this noise can follow different kinds of distribution. Also, I don't think the arguments in Sections 3.3.1 and 3.3.2 are rigorous.

3. In Algorithm 1, it is required to compute the stochastic gradient $\hat f_{\delta}$. It should be explained how to compute that.

4. In Section A, it should be mentioned when the expectation and gradient can be exchanged.

---

> ### Author Response · Authors · 2024-04-18
>
> **Weakness 1:** The sufficient condition for this new $\sigma$-nice function is not clear to me. I didn't understand the role of random variable $\boldsymbol{u}_m \sim B(\boldsymbol{0},1)$ in Proposition 3.1. The definition of $\delta^-$ include $\Vert \boldsymbol{u}_m \Vert$, $\boldsymbol{x}$, and, the angle between $\boldsymbol{u}_m$ and $\boldsymbol{x}^\star-\boldsymbol{x}$. Does $\Vert \boldsymbol{u}_m \Vert$ denote the norm of the random variable or the norm of a realization of this random variable? If it is the norm of a random variable, which norm is applied here? If it is a realization, I found it hard to understand why being $\sigma$-nice would depend on such realization. Also, does different $\boldsymbol{x}$ give different $\delta^-$ values?
>
> **Reply:** $\Vert \boldsymbol{u}_m \Vert$ is the norm of the random variable and $\Vert \cdot \Vert$ is the norm induced from the inner product $\langle \cdot, \cdot \rangle$ of $\mathbb{R}^d$. For example, the 2-norm is possible.
>
> As you say, since $\delta^-$ is a constant, this definition is incorrect. We have thus refined the definition of $\delta^-$ in Proposition 3.1 in the revised manuscript. Please check it out.
>
> **Weakness 2:** In Section 3.3 Equation (8), how does noise $\boldsymbol{\omega}_t$ reduce to the uniform distribution over a ball $\boldsymbol{u}_t$? $\boldsymbol{\omega}_t$ is the noise due to minibatch, so this noise can follow different kinds of distribution. Also, I don't think the arguments in Sections 3.3.1 and 3.3.2 are rigorous.
>
> **Reply:** Thank you for pointing this out. We have reconsidered the derivation of $\boldsymbol{\omega}_t$ in accordance with the reviewer's suggestion. We assume that the stochastic noise follows a normal distribution, based on [Zhang et al., 2020]. We can improve our results by using the fact that the standard normal distribution in high dimensions $d$ is close to a uniform distribution on a sphere of radius $\sqrt{d}$. The changes resulting from this amendment are minor.
>
> We have also clarified our argument by adding an explanation of the generalization and optimization, which was missing from Section 3.3.1. Please see Section 3.3 of the revised manuscript.
>
> The derivation of equation (8) has been corrected. Therefore, we hope that we have resolved the reviewer's concerns regarding Section 3.3.2.
>
> **Weakness 3:** In Algorithm 1, it is required to compute the stochastic gradient $\hat{f}_{\delta}$. It should be explained how to compute that.
>
> **Reply:** Following [Hazan2016], the gradient of $\hat{f}_{\delta}(\boldsymbol{x})$ is approximated by $\nabla f(\boldsymbol{x} - \delta \boldsymbol{u})$, where $\boldsymbol{u} \sim B(\boldsymbol{0}; 1)$. This is sampling and can be used in stochastic algorithms such as SGD [Hazan et al., 2016]. We have added this explanation to the revised manuscript.
>
> However, this is not practical when the full gradient of $f$ cannot be computed. This issue is described near Theorem 3.2 in the revised manuscript.
>
> **Weakness 4:** In Section A, it should be mentioned when the expectation and gradient can be exchanged.
>
> **Reply:** As you say, the necessary explanation was lacking. In order to exchange the expectation and the gradient, the function must be Lipschitz continuous and differentiable. This has been added to Appendix A of the revised manuscript. Thank you for your careful reading.
>
> **Reference**
>
> [Hazan et al., 2016] Hazan, E., Yehuda, K., and Shalev-Shwartz, S. On graduated optimization for stochastic non-convex problems. ICML2016.
>
> [Zhang et al., 2020] J. Zhang, S. P. Karimireddy, A. Veit, S. Kim, S. Kumar, and S. Sra. Why are adaptive methods good for attention models? NeurIPS2020.

---

> ### Author Response · Authors · 2024-04-18
>
> **Request Change 1:** In some sections of the paper, it is unclear whether the discussion pertains to SGD alone or to graduated optimization combined with SGD. This should be clarified.
>
> **Reply:** We have made modifications to Section 3.3 to address Weakness 2. This should address the reviewers' concerns. Please let us know if you still have any further concerns or questions.
>
> **Request Change 3:** I suggest that the authors also provide the total complexity of Algorithm 1, integrating Theorem 3.2 and Theorem 3.1, as well as for Algorithm 4.
>
> **Reply:** Theorem 3.2 already provides a complexity that integrates Algorithm 1 and Algorithm 2 because Algorithm 1 contains Algorithm 2. In the proof of Theorem 3.2, we use the result of Theorem 3.1 and $T_{\text{total}}$ means the total complexity. This is also true for Theorem 3.4. As you say, we didn't word it well enough, so we added this explanation near Theorem 3.2 and 3.4 in the revised manuscript.
>
> **Request Change 2:** The clarity of the writing in the paper needs significant improvement. For instance, in Definition 3.1, it is ambiguous whether "$\gamma_m \in (1/d_{m+1}, 1)$" refers to "for all" $\gamma$ or "exists one". Also, the inputs for Algorithm 4 and Algorithm 2 differ; why is Algorithm 4 termed GD rather than SGD? The authors should verify the clarity of every theorem, proposition, and lemma.
>
> **Reply:** Thank you for pointing out the problem. In the revised manuscript, we have improved the notation so that our results are clear.
>
> From equation (8), for implicit graduated optimization with SGD using a decreasing learning rate or increasing batch size, the function after each smoothing can be considered to be optimized by gradient descent. That is why we call the Algorithm 4 GD.
>
> We have corrected the algorithm input for Algorithm 1 and 2. Thank you for your careful reading.
>
> In response to Requested change 3 and other comments, the notation of theorems and complements has been standardized and the results made clearer. We believe that this has eliminated any concerns of the reviewers. Please let us know if you have any further concerns or questions.

---

> > ### Comment · Reviewer_Kbek · 2024-04-26
> >
> > Thank the authors for the response.
> >
> > I am still looking for clarifications for the following questions
> >
> > 1. I have some concerns about Proposition 3.1 and 3.2.
> >
> > a.  If I understand correctly, I think it is not rigorous to call $u_m$ a random variable. In the definition of $|\delta_m^-|$, the authors take the expectation of $|| u_m ||$ over distribution $B(0,1)$. Should $u_m$ be considered a sample from a random variable following the distribution?
> >
> > b. In the definition of $|\delta_m^-|$,  I am concerned about the scenario where the term inside the square root becomes negative. For instance, if $u_m = 0$, the term inside the square root would simplify to $-r^2(a_m^2 - 1)$, which is negative. How is the expectation handled in this case?
> >
> > c. Definition 3.1 (i) appears to require that the inequality holds for all $\delta_m$. However, do the sufficient conditions outlined in Propositions 3.1 and 3.2 only apply when $| \delta_m| = |\delta_m^-|$?
> >
> > 2. Regarding the use of $\nabla f(\boldsymbol{x}-\delta \boldsymbol{u})$ with $\boldsymbol{u} \sim B(\mathbf{0} ; 1)$ to approximate the gradient, is there a concern that it may be challenging to ensure that the iterates remain within the strongly-convex regime?

---

> > > ### Author Response · Authors · 2024-04-27
> > >
> > > **Reply to Comment a and b:** We consider that $\boldsymbol{u}_m$ is a random variable following the uniform distribution. The term $|\delta_m^-|$ can only be defined if the term in its square root is positive; besides $\boldsymbol{u}\_m=\boldsymbol{0}$, the square root term is negative unless $\left| \cos \theta \right| \geq \left| \frac{r \sqrt{a\_m^2-1}}{\Vert \boldsymbol{x}^\star - \boldsymbol{x} \Vert \Vert \boldsymbol{u}\_m \Vert} \right|$ (14). Therefore, $|\delta\_m^-|$ is not always definable, and the probability $p(a_m)$ that it can be defined can be expressed in the  equation immediately below (4). In other words, the probability $p(a_m)$ is the probability that the term in the square root will be positive.
> > >
> > > **Reply to Comment c:** Indeed, there was something inappropriate in the definition of the new $\sigma$-nice function. From Proposition 3.2, the condition that Definition 3.1 holds for any $δ_m$ may be strict. Therefore, we have modified Definition 3.1 as follows:
> > > Definition 3.1
> > >
> > > (i) For all $m \in [M]$ and all $\gamma\_m \in (0,1)$, there exist $\delta\_m \in \mathbb{R}$ with $|\delta\_{m+1}| := \gamma\_m|\delta\_m|$ and $\boldsymbol{x}\_{\delta\_m}^\star$ such that
> > > \begin{align*}
> > > \left\Vert \boldsymbol{x}_{\delta_m}^\star - \boldsymbol{x}\_{\delta\_{m+1}}^\star \right\Vert \leq |\delta\_m| - |\delta\_{m+1}|.
> > > \end{align*}
> > >
> > > (ii) For all $m \in [M]$ and all $\gamma_m \in (0,1)$, there exist $\delta_m \in \mathbb{R}$ with $|\delta\_{m+1}| := \gamma\_m|\delta\_m|$ and $d\_m > 1$ such that the function $\hat{f}\_{\delta\_m} (\boldsymbol{x})$ is $\sigma$-strongly convex on $N(\boldsymbol{x}^\star; d\_m\delta\_m)$.
> > >
> > > We deeply appreciate your careful review of our manuscript.
> > >
> > > **Reply to Comment 2:** You are right, there is a concern that if the gradient $\nabla \hat{f}\_{\delta\_m}$ is approximated by $\nabla f(\boldsymbol{x}-\delta \boldsymbol{u})$ in Algorithm 1, it may leave the strongly convex region. This is more pronounced the larger $\delta_m$ is. Therefore, we would like to assume in Algorithm 1 and Theorem 3.2 that the gradient $\nabla \hat{f}\_{\delta\_m}$ is directly accessible. Thus, our explicit graduated optimization by Algorithms 1 and 2 is only valid for functions $f$ for which the computation of $\hat{f}\_{\delta\_m}$ by Definition 2.1 and access to its full gradient $\nabla \hat{f}\_{\delta\_m}$ are possible. Algorithm 1 and 2 remain inapplicable to DNNs.
> > >
> > > We apologize for the confusion caused by the repeated corrections.

---

### Author Response · Authors · 2024-04-18

We would like to express our gratitude to the Action Editor and the three reviewers for their valuable comments on our manuscript. We appreciate their detailed assessments and helpful feedback. We have revised the manuscript to incorporate all of the recommendations, which has resulted in an improved presentation of our work. The revised parts of the manuscript are marked in red.

---

### Decision · Action_Editor_32c8 · 2024-05-21

**Recommendation:** Reject

**Comment:**

This submission received a thorough review process thanks to the three reviewers who provided insightful comments. The reviewers also all engaged with the authors during discussion period for further clarifications.

Please see also the comments in the section "Claims and Evidence". Unfortunately, applicability of both explicit and implicit graduated optimization approaches proposed in the paper and the relationship between SGD and Algorithm 3 is not clear.

To summarize the final recommendations of reviewers:

- All the reviewers and myself agree that the paper underwent a significant change between the initial version and the final version. In particular, the reviewers found many technical issues and inaccuracies which are acknowledged by the authors. Since these lead to critical parts of the paper to change significantly (in addition, some of the reviewers are still not convinced by the correctness of some parts, for example, Propositions 3.1 and 3.2), the reviewers do not have sufficient trust in the current version of the manuscript.

- All the reviewers and myself do not find the discussions and explanations about the connection between SGD and the algorithms that are proposed and analyzed in the paper clear or convincing enough.

- **Reviewer 9HcQ** argues that the main theme of the algorithm in establishing the connection between SGD and graduated optimization is undermined due to the lack of a clear connection and the fact that the new Algorithm 1&2 are not implementable in practice, due to Alg 2 being GD. I also agree with this view.

- **Reviewer Kbek** still has concerns with the accuracy of Propositions 3.1 and 3.2, the reviewer already provided pointers for their argument. I agree with the reviewer's concerns.

- In summary, all the reviewers and myself agree on the importance of the topic of the paper. We also agree that the approach the authors take is interesting. However, unfortunately the current version is not suitable for acceptance. I recommend the authors to incorporate the constructive feedback from the reviewers to improve readability and clarity of the submission.

**Audience:**

This paper considers graduated optimization, a global optimization approach that is popular in practice. The paper promises to improve the theoretical understanding of this method and shed light to its success in practice by establishing connections with stochastic gradient descent (SGD). Given the empirical interest in graduated optimization and SGD, the topic of the paper is relevant to the TMLR audience.

**Claims And Evidence:**

The paper's main claim, as can be seen from the abstract is *"SGD with mini-batch stochastic gradients has the effect of smoothing the function, the degree of which is determined by batch size and learning rate"* and the authors argue to provide further implications of this. Even though the paper promises to bridge a gap between theory and practice, unfortunately it ends up leading to more gaps.

One source of this is due to a mistake in the original version that is pointed out by **Reviewer 9HcQ**. As a result, the authors changed SGD in Algorithm 2 to gradient descent in their manuscript and their results (this is the result for **explicit graduation optimization**). The authors mention both in their response and their manuscript that GD need to be used for theoretical purposes whereas SGD is used in experiments. (The alternative solution by the authors propose to use a projected SGD but this is not feasible since one does not have access to the feasible set that they want iterates to stay in) This is not consistent with the original promise of the paper.

Another limitation is that many of the main claims are based on assumptions that are not theoretically justified. For example, Section 3.3 contains a result that the authors emphasize often. However, this relies on a practical observation (as explained before Eq. (8)) that the error between the stochastic gradient and the full gradient follows a normal distribution for *"for some deep learning models and datasets"*. This unfortunately does not read like a principled theoretical analysis. Another example is Section 3.3.1, which, as the authors state *"is based on the somewhat non-theoretical finding that flat local solutions have better generalizability than sharp local solutions"*. These limit the soundness of the theoretical analysis, which is the main contribution of the paper.

The connection between SGD's smoothing effect and implicit graduated optimization approach proposed in this paper is not made clear. Remark before Algorithm 3 is not clear, the connection of Algorithm 3 and the *"SGD running behind the scenes"* is unfortunately unclear.

In addition, both **Reviewer 9HcQ** and **Reviewer Kbek** pointed out many inaccuracies and confusing mathematical statements in the paper, including the final version, particularly in Propositions 3.1 and 3.2. Authors did not manage to explain these points to convince the reviewers. I also agree with the concerns raised by the reviewers and recommend the authors to significantly improve the readability and precision of the mathematical statements in these propositions and overall in the paper.

Due to concerns with the inconsistency between the main claims and the final manuscript and accuracy of the mathematical content, the paper does not satisfy the criterion of TMLR for accuracy between the claims and the results.